# Basal freeze-on generates complex ice-sheet stratigraphy

G.J.-M.C. Leysinger Vieli [1], C. Martín[2], R.C.A. Hindmarsh [2] & M.P. Lüthi[1]

Large, plume-like internal ice-layer-structures have been observed in radar images from both Antarctica and Greenland, rising from the ice-sheet base to up to half of the ice thickness. Their origins are not yet understood. Here, we simulate their genesis by basal freeze-on using numerical ice-flow modelling and analyse the transient evolution of the emerging ice-plume and the surrounding ice-layer structure as a function of both freeze-on rate and ice flux. We find good agreement between radar observations, modelled ice-plume geometry and internal layer structure, and further show that plume height relates primarily to ice-flux and only secondarily to freeze-on. An in-depth analysis, performed for Northern Greenland of observed spatial plume distribution related to ice flow, basal topography and water availability supports our findings regarding ice flux and suggests freeze-on is controlled by ascending subglacial water flow. Our results imply that widespread basal freeze-on strongly affects ice stratigraphy and consequently ice-core interpretations.

[1] Glaciology and Geomorphodynamics Group, Department of Geography, University of Zurich, Winterthurerstrasse 190, 8057 Zurich, Switzerland. [2] British Antarctic Survey, Natural Environment Research Council, Madingley Road, Cambridge CB3 0ET, UK. Correspondence and requests for materials should be addressed to G.J.-M.C.L.V. (email: gwendolyn.leysinger@geo.uzh.ch)

Radio-echo sounding (RES) enables the detection of the bed and internal ice layers of an ice sheet, with the internal reflections, assumed to be isochrones, that correspond to former ice-sheet surfaces[1–3]. Over the past decade, the relevance of RES data has been widely recognised and spatial coverage over Antarctica and Greenland has been substantially increased by NASA's Operation IceBridge[4]. Recent RES data provide internal ice layers at an improved vertical resolution over the whole ice-sheet thickness, permitting observations of the deepest layers[5,6]. What had been identified as mountains in previous RES profiles[7] has now been recognised to be large-scale plume-like internal ice-layer structures, observed for example in large numbers in the Gamburtsev mountains, East Antarctica, as well as in Northern Greenland[8,9] (Fig. 1a). Such ice structures extend from the bed to up to as much as half of the ice thickness, and horizontally over several ice thicknesses (Supplementary Fig. 1); the formation mechanism remains unclear[10]. Current explanations for the genesis of these large and complex plume-like structures involve: basal freeze-on[8,9,11], changes in near-basal ice rheology[12,13] in combination with convergent flow[14], and transient changes in basal friction[15,16] or combinations of the above[10].

Only two of the suggested processes have been tested by means of numerical modelling[14–16]. However, none of these model results reproduce the observed RES layer architecture over the whole depth of the ice sheet, concentrating on the dominant near-basal structures. Bons et al.[14] have obtained some fold structure using a two-dimensional (2-D) structural model at crystal scale, and Wolovick et al.[15] and Wolovick and Creyts[16] produced overturned folds that resemble in parts the observed near-basal fold structures by applying transient changes in slipperiness. However, as previously shown by Leysinger Vieli et al.[17], the layer structure over the whole ice column is of importance, since a local change in boundary condition or flow mode deflects the isochrones over the column, and leads to patterns that can look distinctively different for transects along or oblique to ice flow. This suggests that in order to evaluate a process by means of layer stratigraphy, both modelled and observed ice stratigraphy need to be compared over the whole ice column along transects of similar angle to ice flow.

In Antarctica the occurrence of plume-like structures has been found to be strongly related to basal topography[8,11], and favours basal freeze-on of ascending water at the base of the ice as an explanation. Under the assumption of uniform, constant and

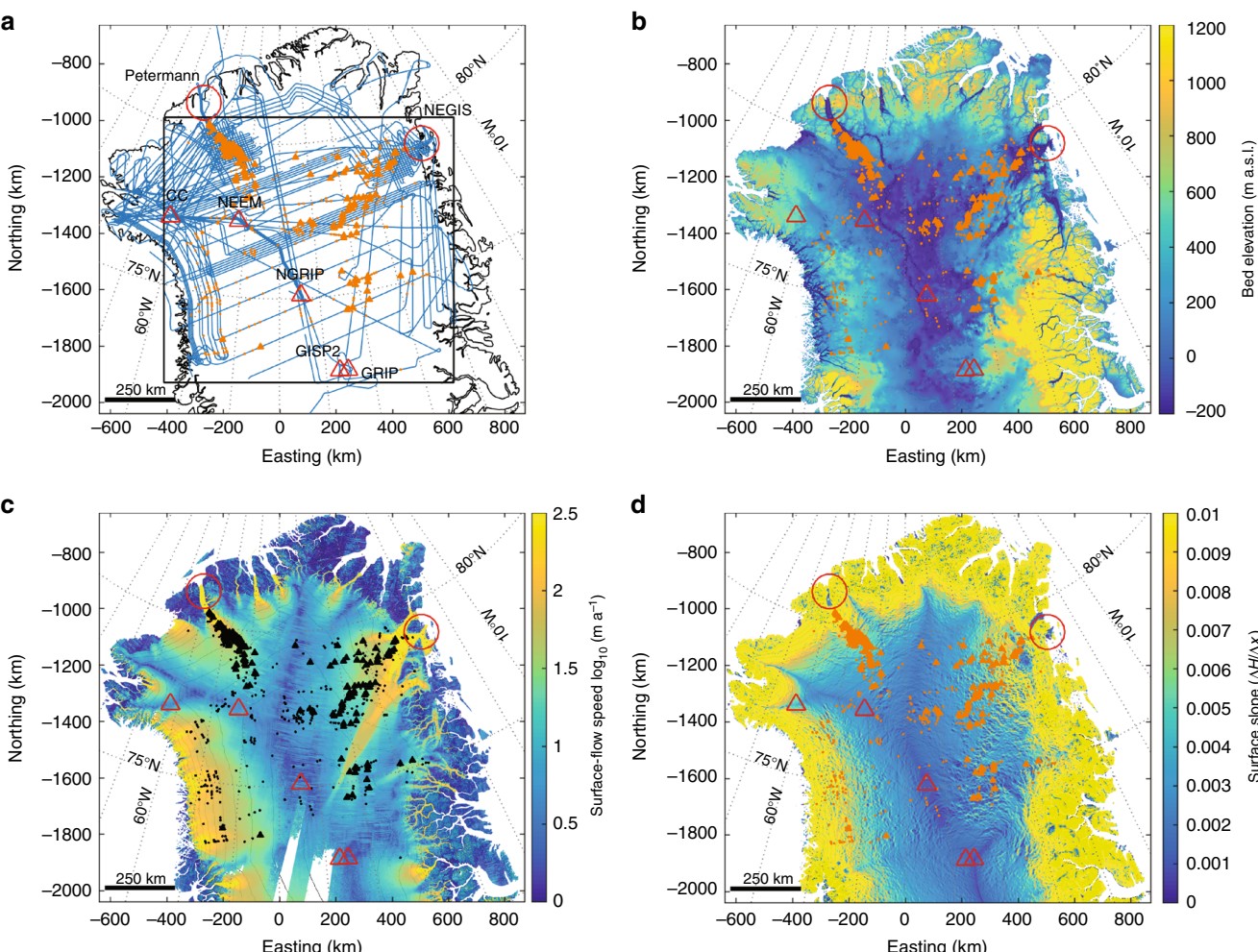

**Fig. 1** Overview of mapped plume locations in North Greenland. Mapped locations of plumes ≥H/3 (filled triangles) and <H/3 (filled circles), from NASA's 2010–2014 Operation IceBridge flights[4], and ice-core locations (red triangles; from NW to SE: Camp Century, NEEM, NGRIP, GISP2 and GRIP) together with Petermann glacier and Northeast Greenland Ice Stream (NEGIS; large red circle) are shown for all subplots. **a** Outline of North Greenland[46] together with 2010–2014 flight lines[4] (blue). Outlined area (black box) shows section used in Fig. 2 and Supplementary Figs. 2 and 3. **b** Bed topography[23,26] (additional contours at −500, −250 and 0 m.a.s.l.). **c** Surface topography[26] (250 m contours) superimposed by surface-flow speed[43]. **d** Surface-slope ΔH/Δx[32]

low effective pressure $P_{w}$ (basal water pressure $P_{w}$ near or at the ice overburden pressure; see Methods section 'Basal heat balance'), the mechanism behind glaciohydraulic freeze-on is the pressure-induced depression of the melting point (termed here PiDMP), stemming from the Clausius–Clapeyron relation, which describes the relationship between pressure and melting point. As water flows uphill along the hydraulic gradient (this is possible for bed inclines no steeper than 11 times the surface slope[18]), reduction in ice overburden leads to an increase in its melting point temperature[19,20]. In consequence, water will freeze onto the base with the released latent heat keeping the temperature of the uphill flowing water close to its melting temperature.

The basal temperature of the Greenland ice sheet is only poorly constrained, as we do not know the geothermal heat flux or how it varies on a kilometre to 100 km scale. Analysis combining independent data are needed to obtain reconstructions for the basal temperature[21,22]. Neither do we know the basal topography well enough to evaluate if there are warm-based deeper parts surrounded by frozen areas[23,24]. In addition, does the community's limited understanding of basal hydrology prevent us from predicting its thermodynamics consequences, in particular answering whether it is possible for meltwater channels to advance through freeze-on areas so as to connect isolated melt patches.

Here, we examine the hypothesis of basal freeze-on and present the consequent isochrone layer architecture obtained using mathematical models of various complexities and spatial dimensions; the results closely resemble the layer architecture observed in RES profiles. Observed plume-like features map well with locations that are favourable in topography for the PiDMP

process and match areas where water flow is likely, additionally corroborating the hypothesis. We further establish that low ice flux and contrasts in ice rheology are conducive to larger plumes. Our findings highlight important implications of the basal freeze-on process for ice dynamics and ice-core drilling.

## Results

**Applying basal freeze-on.** Although in Greenland the relief of the subglacial topography is less pronounced than in Antarctica, we find that plume-like structures (position of apex in RES profiles; see Methods section 'RES data') are predominantly located on the flanks of the main subglacial basin in regions of moderate ice flow and surface slope (Fig. 1b–d). To verify the relationship with a rising bed, we define an index for freeze-on ($\Phi$) as a purely geometrical quantity (see Methods section 'Basal freeze-on index' Eq. (25)) that accounts for the effect of the bed incline (relative to surface slope) on freeze-on triggered by PiDMP. Analysing the spatial distribution of internal ice-layer structures with regard to $\Phi$ and water flow path, while considering the likelihood of a frozen bed, suggests that in general plumes are located along the main subglacial water paths crossing areas of elevated $\Phi$ (Fig. 2, Supplementary Figs. 2 and 3 and Supplementary Tables 1 and 2; see Methods sections 'Estimate of water paths' and 'Estimating correlation between freeze-on index and plumes'; note, uncertainties are largest in regions of sparse radar data[23], which affects both $\Phi$ and the computed water path). According to theory, PiDMP depends dominantly on basal water flux and topography, whereas the thermal state of the ice is of secondary importance.

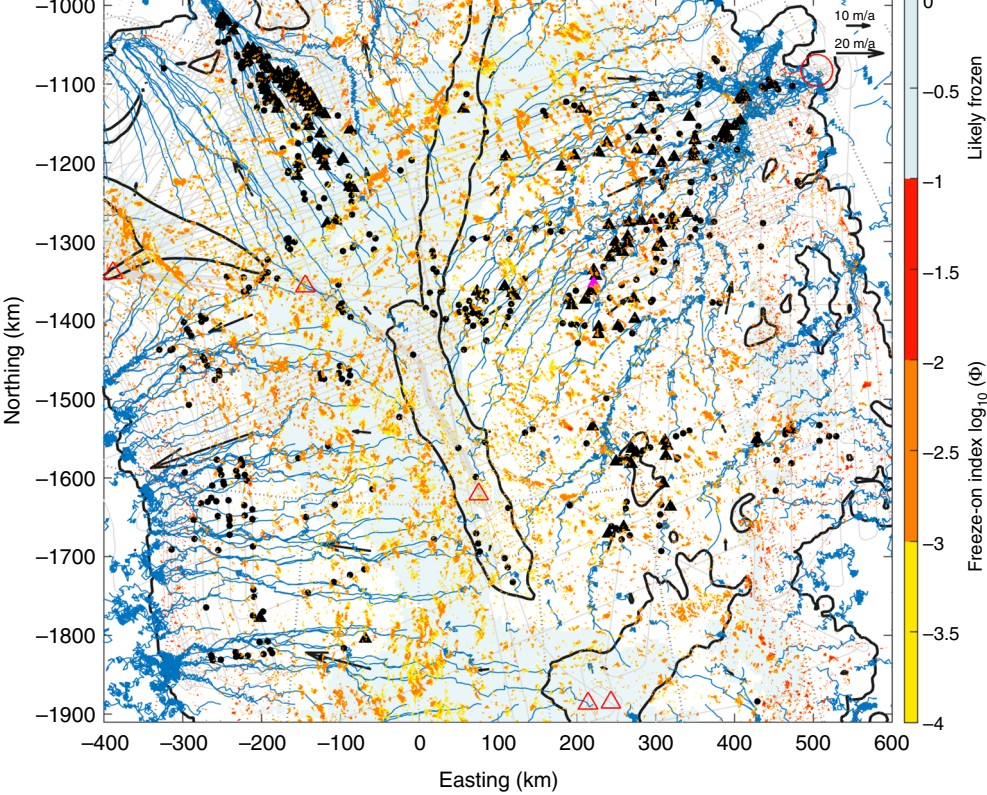

**Fig. 2** Plumes in relation to freeze-on index $\Phi$ and water paths. Calculated freeze-on index $\Phi$ (yellow-orange-red; grid 1050 m; note the logarithmic scale for $\Phi$) superimposed on the modelled basal water flow path (blue; seeds on 45 km grid with 5 km step size along streamline; see Methods section 'Estimate of water paths') along the hydrostatic-potential gradient. Map section as shown in Fig. 1a using same labelling, with 2010–2014 radar flight lines[4] (grey thin lines). Direction and magnitude of ice-surface velocity[43] on 150 km grid (dark-grey arrows) together with likely frozen bed after MacGregor et al.[22] (light blue) and balance velocity contours[47] at 4 m a$^{-1}$ (dark grey) are shown in background. Highlighted plume (magenta-filled triangle) is shown in Fig. 3a and Supplementary Fig. 1

Our spatial analysis thus supports the hypothesis of basal freeze-on due to ascending water for Greenland.

Here we test by numerical modelling the hypothesis of basal freeze-on causing plume formation and investigate how such accreted ice at the ice-sheet base influences the internal layer architecture. We use a forward, 3-D, time-dependent, numerical ice-flow model conceived as a stream-tube model (bed and surface are fixed in time; see Methods section '3-D stream-tube model') using finite differences. It uses simplified mechanics based on the shallow-ice approximation (SIA), which ignores all horizontal deviatoric stress gradients and is therefore based on the assumption of small horizontal deviatoric stress gradients. The model is capable of continuously tracking tracers and isochrone layers within the ice[17,25] (see Methods section 'Numerical models'). We simulate local freeze-on by adding mass at the ice-sheet base. The geometrical setting for a typical plume (Fig. 3a, b) is taken from observations (ice thickness,

surface slope and distance to ice divide)[4,23,26] using simplified topography (uniformly inclined surface and a flat bed) while assuming no basal sliding.

First, we ran our model to reach the steady-state solution for the undisturbed case of uniform surface accumulation without basal accretion, which results in surface and bed parallel internal layers and a non-linear age-depth relationship (Fig. 3c–e). Following this, we simulated local freeze-on by adding mass using a freeze-on rate $\dot{f} = 0.8 \, \mathrm{m \, a^{-1}}$—of similar magnitude to the surface accumulation ($\dot{a} = 0.1 \, \mathrm{m \, a^{-1}}$)—over a basal-accretion area 6 km along flow and 7 km across flow. We observe two main phases of plume growth in an along profile: initially, after the start of basal freeze-on, the plume grows vertically, adding mass to the column consisting of the meteoric ice above, and warping isochrone layers upwards. The oldest meteoric ice is located immediately above the plume (Fig. 3f–h and Supplementary Fig. 4a, b). Subsequently, once the plume has reached its

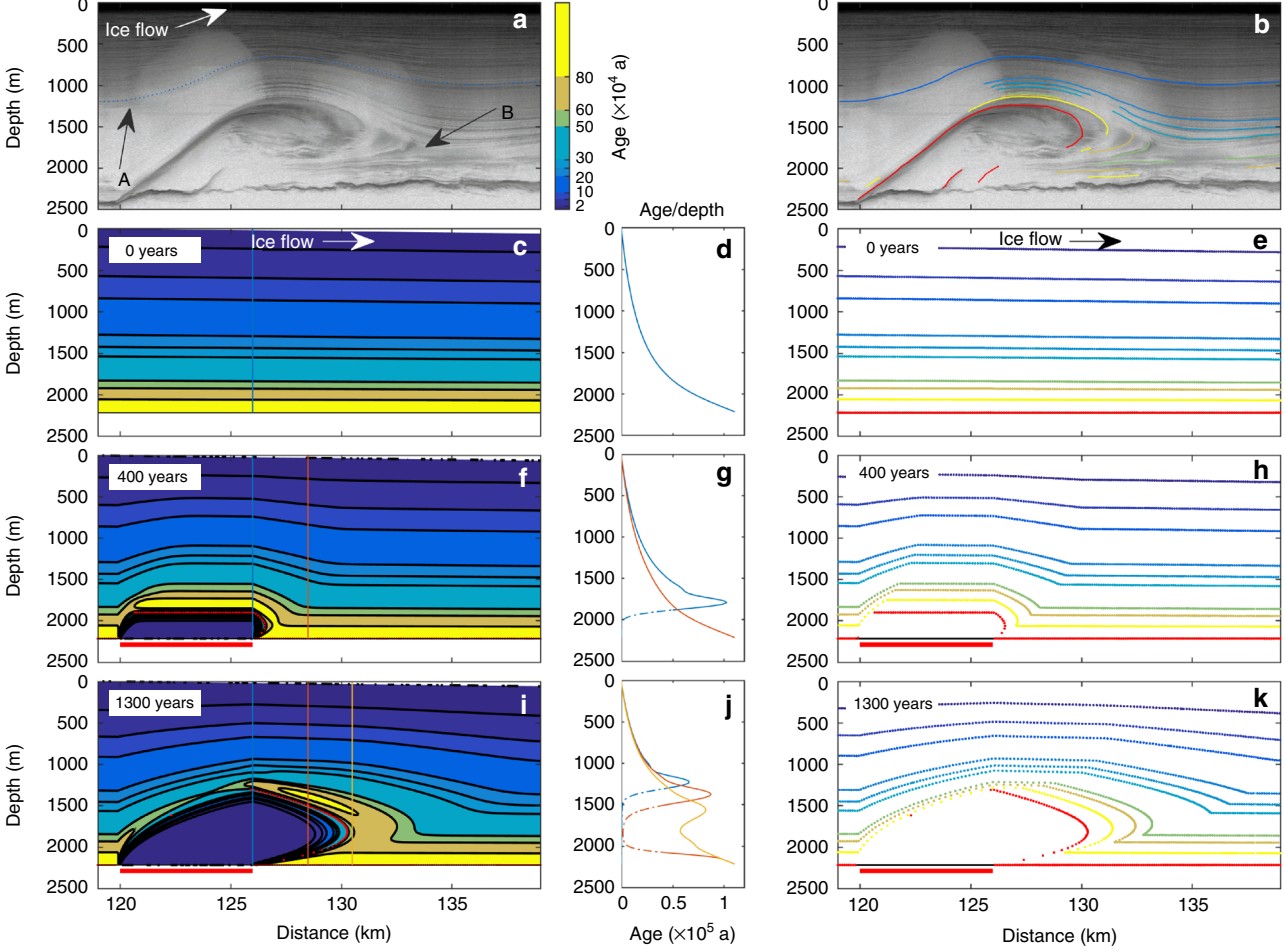

**Fig. 3** Observed and modelled plume showing internal layering. **a** RES profile (20120507_07_001,002)[4] of a plume-like feature (magenta-filled triangle in Fig. 2) ≈ 25° oblique to flow (see Supplementary Fig. 5), showing characteristic internal layering with upward bending (arrow A) and downstream folding (arrow B). **b** Same RES profile with interpretation of meteoric layers (solid lines; colour legend for age as for modelled layers) and outlined freeze-on plume (red solid line). **c** Modelled internal layer structure purely by internal deformation for a surface-accumulation rate $\dot{a} = 0.1 \, \mathrm{m \, a^{-1}}$, zero accretion and a surface-slope $\alpha_s = 0.003$. Age contours (at 0, 2, 6, 10, 20, 25, 30, ..., 80 × 10⁴ years) matching inferred layers (right margin in **b**). **d** Undisturbed age-depth profile (blue line in **c**). **e** Particles (coloured dots) taken along age contours in **c**. **f, i** Modelled internal layer structure 400 and 1300 years after switch on of freeze-on, respectively, with freeze-on rate $\dot{f} = 0.8 \, \mathrm{m \, a^{-1}}$ over 6 km accretion area (red horizontal bar). Tracked interface between meteoric and accreted ice (red dots from **h** and **k**, respectively). **g** Disturbed and undisturbed age-depth profile for vertical lines in **f**, above (blue line) and in front (red line) of the accretion area, respectively. **h, k** Particles tracked over time using modelled velocity field (from **f** and **i**, respectively) showing the layer architecture in detail, as it is not affected by diffusion. Interface between meteoric and accreted ice (red dots). **j** Disturbed age-depth profiles above (blue line) and in front of the accretion area (red and orange lines). Note, age-depth profile representing accreted ice (dashed-dotted line) and age diffusion in the vicinity of the interface is a numerical artefact (see Methods section 'Considerations of numerical age modelling')

maximum height, the accreted mass is advected downstream, with the plume stretching horizontally, intruding into the already distorted meteoric ice isochrones (meteoric ice originates from surface accumulation), and causing folding (Fig. 3i–k, Supplementary Fig. 4c, d and Supplementary Movie 1). The same comments apply to profiles oblique to ice flow (Supplementary Fig. 5).

The internal meteoric ice-layer architecture follows the plume shape. Typically, near-basal internal layers are bent sharply upwards at the upper end of the plume, and folded at the downstream end of the plume, resembling the observed RES structure (arrows A and B, respectively, Fig. 3a, b, i, k). This plume growth results in a strong age-depth disturbance over the accretion area, with the lowest, oldest meteoric ice being pushed upwards by accretion of fresh basal ice (Fig. 3g, j (blue line)), and exhibiting an age-depth inversion for both accreted and meteoric ice at the downstream end of the plume (Fig. 3j).

Applying accretion locally at the ice-sheet base results in large horizontal gradients in basal vertical velocity, which might lead to incompletely correct calculated velocities by the SIA-based model, affecting the isochrone structure. To check on the calculated layer structure, we perform additional experiments with a 2-D plane-flow full-system (FS) model solving the equations describing the mechanics of ice flow (full-Stokes equations) and a freely evolving surface (see Methods section '2-D FS ice-flow model'). By solving all stress gradients, the FS model avoids the difficulties associated with the simplified mechanics of the SIA model. As the FS model comes at a high computational cost, we use it here in plane flow only, for validation purposes. These additional experiments confirm the plume dynamics and isochrones patterning obtained by the SIA-based model (Fig. 4a).

A further complication is the jump in age that would occur in nature between the fresh accreted ice and the oldest meteoric ice. For both SIA and FS models this jump leads to numerical diffusion of age. In order to follow the age discontinuity and to test independently the isochrone structure obtained by solving the age advection equation (see Methods section 'Governing equations', Eq. (7)), we track particles for the SIA-based model taken along the initial layer structure over time using the modelled velocity field (Fig. 3e, h, k; see Methods section 'Considerations of numerical age modelling' and 'Particle tracking').

In order to investigate the effect of ice flow on the plume shape, we apply an uniform amount of basal freeze-on, using a freeze-on rate three times larger than the surface accumulation along an extent of 5 km, to produce distinctive plumes at three different locations along the flow line. Inserting three separate accretion locations allows us to examine the influence of varying the proportions of meteoric and accreted ice flux. This numerical experiment results in plumes of different shapes becoming increasingly shallow and elongated for locations further downstream (Fig. 5a). This difference in height can be explained using mass-conservation principles and ice kinematics arising from its mechanics. In steady-state the total ice flux ($Q$) can be expressed by the balance flux ($B$), which is the sum of the accumulation rate ($\dot{a}$) and the freeze-on rate ($\dot{f}$) integrated from the divide to the current location downstream, where for surface or basal melt $\dot{a}, \dot{f} < 0$ (see Methods sections 'Governing equations' and 'Ice-flux relationship with basal freeze-on', Eqs. (2)–(5), (6) and (8)).

In summary, the total ice flux $Q$ consists of ice flux arising from meteoric accumulation ($M$) and from freeze-on ($F$). In a vertical ice column in plane flow, the total ice flux $Q$ is the sum of the meteoric ice flux $M$, which comes from the surface, and the freeze-on (or accreted) ice flux $F$, which originates at the bed (Fig. 5c). The relative height of each flux component is expressed by the partial flux through the respective ice column (see

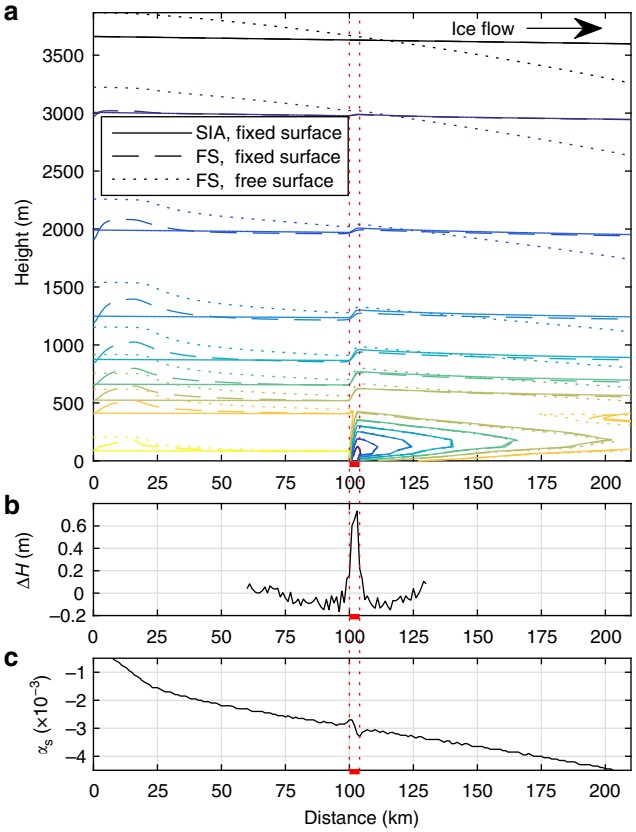

**Fig. 4** Full-system modelling of freeze-on and comparison of internal layer structure resulting from simplified flow physics. **a** Comparison of basal accretion ($\dot{f} = \dot{a} = 0.25 \, \text{m a}^{-1}$) between SIA (solid line) and two FS calculations, with (1) fixed surface (dashed line)—as for the SIA case—and (2) for a free surface (dotted line), where all three were calculated with the finite-element model Elmer/Ice[38]. All show characteristic sharp upward bending of isochrones over the accretion extent (red line) and closely matching plume layer structure. **b** Difference between detrended (polynomial of degree 2) and original surface from FS free-surface run, showing the surface expression of the plume. **c** Surface-slope $\alpha_s$ over 1 km grid. Note, surface expression of plume leads to a change in surface-slope $\alpha_s$, with increasing $\alpha_s$ over the accretion area and decreasing $\alpha_s$ upstream and downstream of the accretion area. The boundary of the accretion area (red horizontal bar) is shown for all three subplots (red dotted line)

Methods section 'Ice-flux relationship with basal freeze-on', Eq. (11)), and is a function of the relative contribution of $M$ and $B$ to the total ice flux $Q$ and to the horizontal velocity distribution with depth (see Methods section Ice-flux relationship with basal freeze-on', Eqs. (9) and (12)). (The relative contribution to total ice flux $Q$ for each flux component is $1 = M/Q + F/Q$.) We define the fraction of the freeze-on flux component as the ice-flux ratio

$$R \equiv \frac{F}{Q}, \tag{1}$$

which equals the normalised partial flux for an ice column of normalised height ($h$; see Methods section 'Ice-flux relationship with basal freeze-on', Eq. (17)).

Along flow, the freeze-on flux $F$ increases only over the accretion area for both the transient and the steady-state case, while the meteoric ice flux $M$ is steadily increasing (Fig. 5c). This along-flow relationship between $F$ and $M$ leads, apart from above accretion areas, to a decreasing flux ratio $R$ (Fig. 5d). We find that

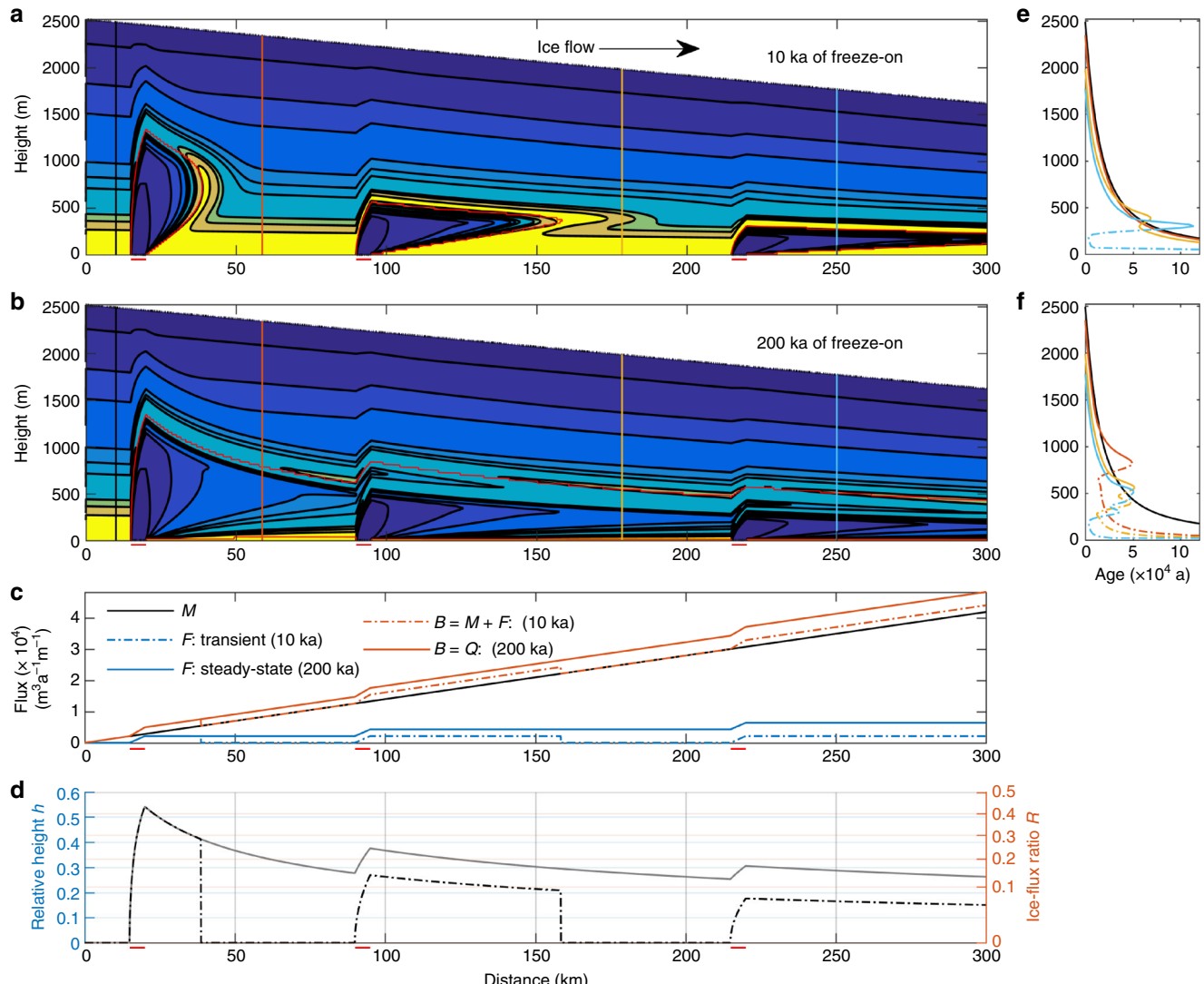

**Fig. 5** Plume size, internal layer structure and accretionary-meteoric ice-flux contributions along ice-sheet flow. Modelled internal layers along flow (flow purely by ice deformation) with meteoric-to-accreted ice interface (red outline) after application of a freeze-on rate of ($\dot{f} = 0.42\,\mathrm{m\,a^{-1}}$) over three accretion regions of 5 km extent. Surface accumulation ($\dot{a} = 0.14\,\mathrm{m\,a^{-1}}$), surface slope and background melt ($10^{-6}\,\mathrm{m\,a^{-1}}$) are constant over time. **a** Transient run after 10,000 years. **b** Steady-state run (200,000 years). **c** Meteoric ice flux $M$ shown together with freeze-on flux $F$ and balance flux $B$. **d** Ice-flux ratio $R$ (right $y$-axis) and relative plume height $h$ (left $y$-axis) for ice flow by internal ice deformation only, shown for both 10 ka (black dashed-dotted line) and 200 ka (grey solid line) of freeze-on. **e, f** Age-depth relationship for vertical profiles in **a**, **b** for meteoric (solid line) and plume ice (dashed line) compared to an undisturbed profile (black solid line). Note, age contours chosen as in Fig. 3

the factor limiting the plume height is primarily given by the ice-flux ratio $R$.

Of secondary importance for the plume height is the distribution of the horizontal speed with depth (see Methods section 'Ice-flux relationship with basal freeze-on', Eqs. (13) and (14). For the case of plug flow (no internal deformation) the maximum plume-height relative to the total ice thickness ($h$) is proportional to $R$ (Fig. 6). For the case of flow purely by internal deformation (as used in our numerical experiment), the plume height further increases for the same $R$. This is due to the presence of internal shear, which causes the horizontal velocity to decrease non-linearly with depth (Fig. 6). This plume-height effect is strongest for small $R$ (e.g. tripling of $h$ for $R = 5\%$).

**Ice-flux dependency.** As illustrated in Fig. 5c, the main contribution to the balance flux $B$ (total flux $Q$) comes from the meteoric ice flux $M$ inducing in general a decrease in the ice-flux

ratio $R$ along flow (Fig. 5d), and hence plume height (Fig. 5a, b). These calculations show that as the steady-state isochrone geometry is approached, multiple accretion patches aligned along flow lead to plumes with different sources being stacked on top of each other, with thin layers of meteoric ice lying in between (Fig. 5b). This also implies that accreted ice extending downstream is subsequently lifted by growing plumes that have originated downstream, producing complex vertical layer-pattern and age-depth distributions (Fig. 5e, f and e.g. Supplementary Fig. 1b, f, g). The complex, modelled result can produce the distinct morphological structures observed in RES profiles from Antarctica and Greenland[14,27].

The flux dependence illustrated has crucial implications for areas of low surface-accumulation rates (e.g. East Antarctica or Northeast Greenland), since where meteoric ice flux $M$ is small, even a low freeze-on flux $F$ is able to produce relatively large plumes (see Methods section 'Calculating freeze-on plume

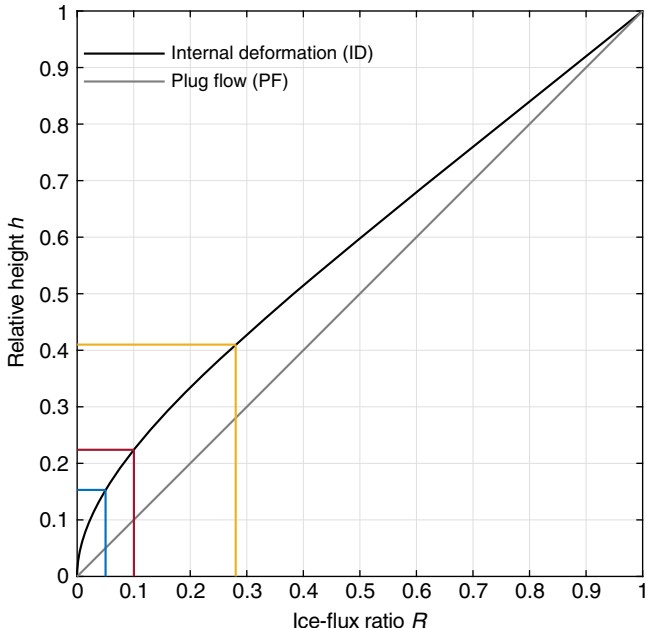

**Fig. 6** Relationship of relative plume height $h$ to the ice-flux ratio $R$ of basally derived ice flux to the total ice flux. Vertical flux shape function for internal deformation (solid line) and plug flow (grey line) obtained from $R$ as a function of $h$. $h$ for 5, 10 and 28% (blue, red and yellow lines) of total ice flux $Q$ are highlighted. A $R$ of 1 is equivalent to the total ice flux $Q$

parameters'). This effect is amplified in regions with small surface-accumulation areas, which further reduce the meteoric ice flux $M$; such regions exist for instance near ice divides or where flow is divergent, typically upstream of rising beds[28,29]. Downstream, plume heights generally decrease owing to the increasing meteoric ice flux $M$, which is a consequence of both contributing accumulation area and flow convergence. In addition, basal sliding becomes more important downstream, affecting the distribution of horizontal velocity with depth, thus further decreasing the plume height (Fig. 6).

**Relation to basal freeze-on rates $\dot{f}$.** In general, subglacial hydrology at the base of the ice sheet is poorly constrained as a consequence of the logistic difficulties of achieving sufficient observations. However, recent modelling studies suggest the existence of major meltwater pathways, subglacial lakes and, related to the mechanism described here as PiDMP, widely occurring basal freeze-on[30,31]. Predicted freeze-on rates are in general very low (<1 mm a$^{-1}$) and only locally larger rates (a few centimetres to metres per year) are obtained[31]. Substantial rates (tens of centimetres per year) have been previously suggested as a result of PiDMP, controlled by the bed slope ($\alpha_b$), surface slope ($\alpha_s$), and the water flux ($Q_w$) when $\alpha_b$ has opposite sign to $\alpha_s$[20] (see Methods section 'Basal heat balance'). The PiDMP process leads to an increase in freeze-on rate $\dot{f}$ both for increases in water flux $Q_w$ and bed slope $\alpha_b$, and to a decrease in $\dot{f}$ for an increase in surface slope $\alpha_s$[18–20].

Since water flux $Q_w$ is unmeasured, we use the relationship between the bed and surface slope to quantify the effect of PiDMP on the freeze-on rate $\dot{f}$. Thus, we define the slope ratio $S = \alpha_b/\alpha_s$, and find that for slope ratios ranging between $-2 \geq S \geq -11$, the freeze-on rate $\dot{f}$ not only increases for increasing $Q_w$ but also for increasing $\alpha_s$ (Fig. 7 and see Methods section 'Basal freeze-on rates for plane flow', Eq. (20)). The effect of the slope ratio $S$ on the freeze-on rate $\dot{f}$, as shown in Fig. 7a, b, ranges between a few centimetres (for both small $\alpha_s$ and $Q_w$) to tens of centimetres (for

large $Q_w$) and even metres per year (for both large $\alpha_s$ and $Q_w$). As the calculated bed and surface slope depend heavily on the grid size used, it is readily conceivable that locally conditions arise in $\Phi$ and $Q_w$, which lead to freeze-on rates comparable to the ones used in our experiments (Fig. 7).

We term a bed slope of opposite sign to the surface slope an 'adverse slope'. The dependence of PiDMP on $S$ has the consequence that water can flow up steeper adverse bed slopes $\alpha_b > 0$ for a steeper inclined surface slope $\alpha_s < 0$ (or vice versa), leading to a greater PiDMP effect owing to larger spatial gradients in overburden and thus melting point. This dependence on slope is of importance in the interior of Greenland, where the ice-sheet surface in general exhibits larger slopes ($\alpha_s \leq -0.4°$ or $\alpha_s \leq -0.007$) than in Antarctica ($\alpha_s \leq -0.2°$ or $\alpha_s \leq -0.0035$)[32], allowing for larger freeze-on rates $\dot{f}$ for the same water flux $Q_w$ (Fig. 7a, b). PiDMP can further be affected by feedback from plume growth on the local surface topography above. If we assume that water flows along the same path the freeze-on rate $\dot{f}$ becomes smaller for steeper surface slopes and vice versa.

Our model experiment including full-flow physics and a freely evolving surface results in a reduced surface slope $\alpha_s$ by $2 \times 10^{-4}$ upstream and downstream of the freeze-on area and steadily increasing from 2.7 to $3.3 \times 10^{-3}$ over the freeze-on area (Fig. 4b, c), leading to a drop in slope ratio of nearly $2S$ along the bed slope $\alpha_b$, and hence leading to freeze-on rates $\dot{f}$ that are slightly larger upstream than downstream assuming a constant water flux $Q_w$ (Fig. 7a, b, black line). The difference in $\dot{f}$ between the onset and end of the freeze-on area is a few percent and negligible over the whole area as they counterbalance each other. However, of significance is the effect of both spatial and temporal surface change[28] on the hydraulic gradient, possibly redirecting the water upstream or allowing water flow over steeper adverse slopes downstream. This complex feedback mechanism is expected to amplify and extend plume growth over initially unfavourable $\alpha_b$, while changing the hydraulic gradient. Furthermore, a distributed water system over an adverse slope $\alpha_b$ steep enough for freeze-on to occur is likely to contribute to plume formation over a broader area[20].

**Analysis of spatial distribution of plumes.** A detailed observational analysis for Greenland locates large plumes along possible water paths that cross areas of high freeze-on indices ($\Phi > 10^{-3}$; Fig. 2 and Supplementary Fig. 2 and Supplementary Table 1; see Methods section, 'Estimating correlation between freeze-on index and plumes') in regions of predominantly low surface accumulation[32] (permitting a larger ice-flux ratio $R$) and moderate ice velocities ($10 - 100$ m a$^{-1}$; Fig. 1c). Near the ice divide the observed plumes are usually smaller than $H/10$, in the range of a few 100 m. For these relative small plume heights, the flux ratio $R$ too is small when assuming flow purely by internal deformation. As the ice flux is low, only low water fluxes are needed to produce the required freeze-on rates (a few millimetres to centimetres per year; see Methods section, 'Calculating freeze-on plume parameters'). Since the surface slope is nearly flat, $\Phi$ is also low (<10$^{-3}$), commonly leading to bed slopes that are too steep for water to flow uphill, and restricting plume formation. Away from the divide, the first large plumes (>$H/3$) start to form in regions where the basal meltwater is likely convergent, leading to an increasing basal water flux $Q_w$, with the total ice flux $Q$ and freeze-on index $\Phi$ still being generally small. Further along, towards the margin both $\Phi$ and $Q_w$ tend to increase (while $Q$ is still moderate) and allow for larger plume heights. In regions of fast flow (large $Q$), the available basal water flux and $\Phi$ are in general too low to produce freeze-on rates $\dot{f}$ that would sustain the large ice-flux ratio $R$ required for large plumes. These

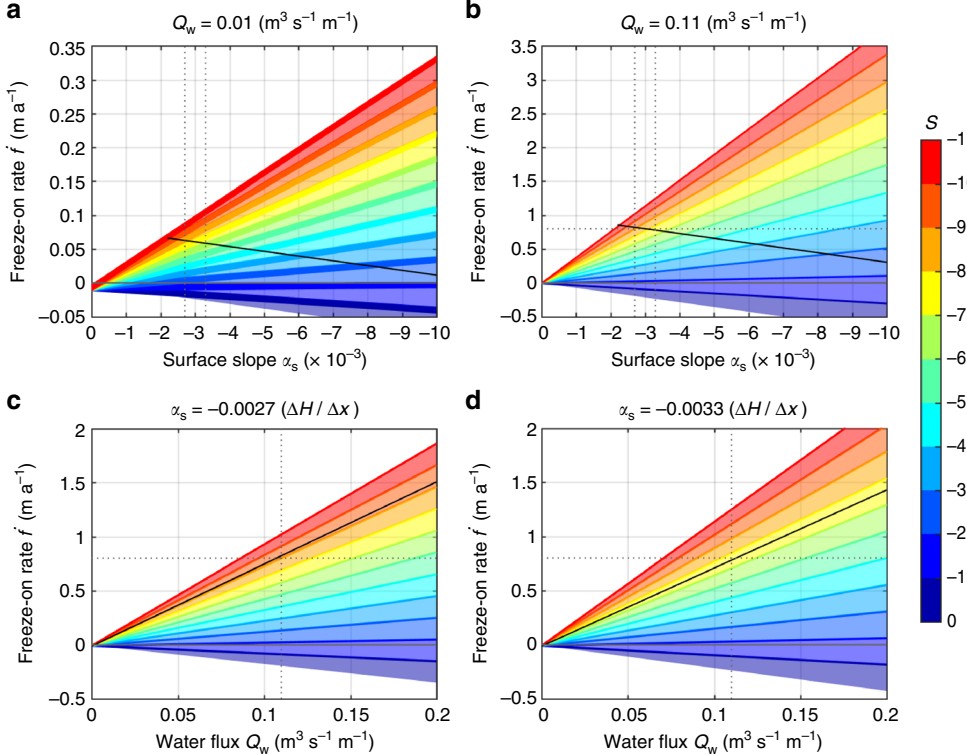

**Fig. 7** Calculated freeze-on rates. Freeze-on rates calculated for a range of geothermal heat fluxes ($0.03 \leq G_g \leq 0.11$ W m$^{-2}$; effect of range reflected in the thickness of the coloured solid lines) using an ice velocity $u_s = 10$ m a$^{-1}$ and an ice thickness $H = 2200$ m. **a**, **b** Freeze-on rate $\dot{f}$ for a constant water flux $Q_w$ and various bed-to-surface-slope ratios $S$ (colour range) over a range of surface slopes $\alpha_s$ (grey vertical dashed line for both $\alpha_s = -2.7 \times 10^{-3}$ and $\alpha_s = -3.3 \times 10^{-3}$ as used in **c**, **d**). **c**, **d** $\dot{f}$ for a constant $\alpha_s$ and various $S$ over a range of $Q_w$ (grey vertical dashed line for $Q_w = 0.11$ m$^3$ s$^{-1}$ m$^{-1}$ as used in **b**). Freeze-on rate $\dot{f} = 0.8$ m a$^{-1}$ (grey horizontal dotted line) and fixed bed slope $\alpha_b = 0.025$ (black solid line), as calculated for the plume in Fig. 3a, b (see Methods sections 'Basal freeze-on rates for plane flow', 'Estimate of basal meltwater' and 'Estimate of freeze-on rate and water flux'). Note the different scales for $\dot{f}$, and how the surface slope affects $\dot{f}$ along a fixed bed in **c**, **d**

observations fit well with the modelling results by Dow et al.[31] producing larger freeze-on rates along important water flow paths and towards the margin.

The enhanced availability of basal meltwater generated from basal sliding and water flow, combined with adverse bed slopes, is key to the formation of large plumes in regions of moderate ice flow. Meanwhile, the along-flow reduction of water flux $Q_w$ by basal freeze-on is minimal, as the freeze-on flux $F$ is orders of magnitude smaller than $Q_w$ (see Methods section, 'Along-flow reduction in water flux'). Towards the onset of fast flow, the existing plume-like structures concentrate due to convergent flow (i.e. Petermann Glacier, Northeast Greenland Ice Stream (NEGIS); Figs. 1c and 2 and Supplementary Fig. 2), merging the advected plume accretions and adding to layer complexity[14,33].

In fast-flow areas, the large ice flux together with increased basal sliding (leading to reduced vertical gradients of horizontal velocity that resemble the plug flow case) prevent in general high-rising plumes (Fig. 5). An exception occurs for very large water fluxes and along-flow series of freeze-on areas, which leads to stacking of the advected plumes (Fig. 5b). Further along, in the ablation area, where surface melt reduces both the meteoric ice flux $M$ and, as a consequence, the total ice flux $Q$ (hence increasing $R$) and high availability of surface meltwater reaching the bed[9,34], larger plumes are again possible.

**Contrast in ice rheology.** Plume-ice rheology is expected to differ from meteoric ice rheology[9], reflecting different ice temperature and fabric, and consequently affecting the relationship between

ice-flux ratio $R$ and relative height $h$ (see Methods section 'Ice-flux relationship with basal freeze-on', Eq. (17)) for cases with internal deformation rather than plug flow. Large differences in rheological properties in near-basal ice has been observed in Greenland[12]. A recent study by Wrona et al.[27] describing the geometry and morphology of near-basal anomalous layer structures in RES profiles over the Gamburtsev mountains, East Antarctica, concludes that mechanical mixing between meteoric and accreted ice is triggered by rheology contrasts, producing complex structures.

To investigate the effect of a rheology contrast between the meteoric and accreted ice on plume growth, we perform a simple experiment, using the FS model. We use the same experimental set-up as described earlier, with the exception of a modified accreted ice viscosity. By changing the rate factor ($A$) for ice flow[35,36] (see Methods section 'Ice-flux relationship with basal freeze-on') by factors of 0.1, 0.5, 2 and 10 in four separate experiments we increase or reduce the stiffness of the accreted ice respectively, in order to compare it with the base (accretion and meteoric ice rate factors equal) case, which uses an uniform viscosity throughout the ice sheet. For the same amount of accreted ice with increased stiffness (factors 0.1 and 0.5) we find that plumes as expected rise higher than in the base case (Fig. 8; red dashed line), since stiffer ice cannot shear so easily. A reduction in accreted ice stiffness leads however to a complex and not entirely expected result. Over the accretion area the softer accreted ice rises not as high as in the base case (Fig. 8a, d), which might be explained by the fact that it shears more easily. However, downstream of the accretion area the softer ice

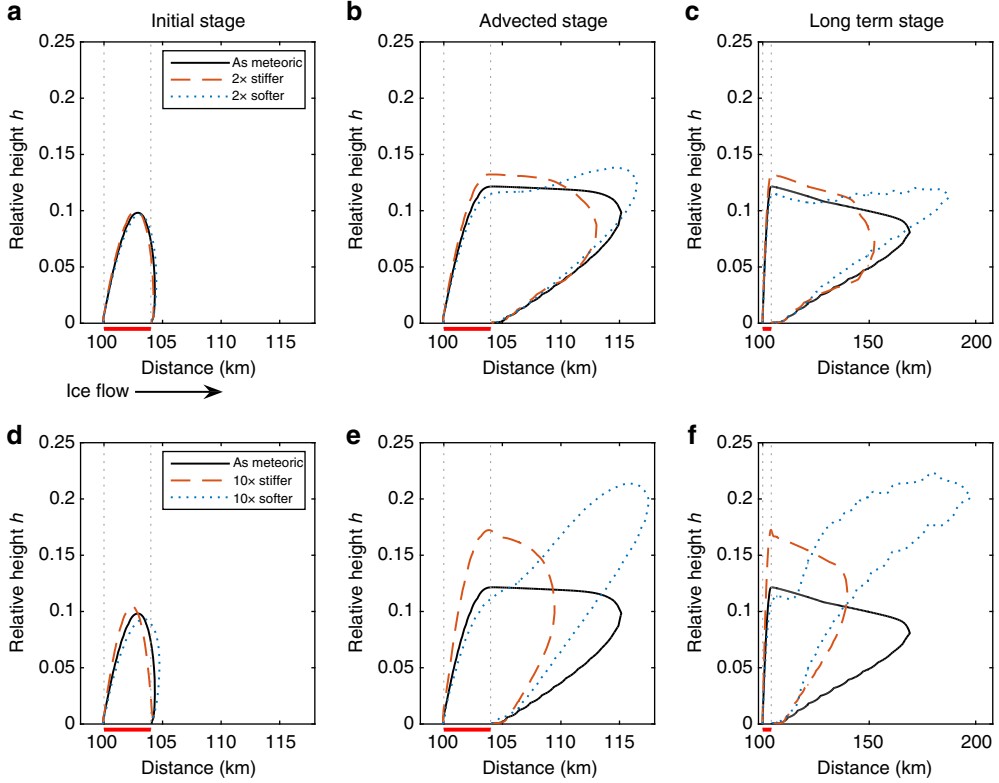

**Fig. 8** Plume-height dependency on rheology contrast between accreted and meteoric ice. Modelled interface between accreted and meteoric ice using the FS model (Elmer/Ice[38]) as used in Fig. 4, but now with modified accreted ice viscosity. Comparison between stiffer (red dashed line) and softer (blue dotted line) ice with ice of uniform viscosity (black solid line). **a–c** Twofold (2×) viscosity contrast between accreted and meteoric ice. **d–f** Tenfold (10×) viscosity contrast between accreted and meteoric ice. The snapshots in time show the different growth and advection stages: **a**, **d** vertical plume growth for all three cases; **b**, **e** only horizontal advection, except for the soft case, which is still growing; **c**, **f** only horizontal advection for all cases. All three time snapshots show accretion area (red horizontal bar) and margin (light-grey dotted line). Note different horizontal scale for **c**, **f**

eventually rises higher than the original plume, which is simply advected downstream (Fig. 8b, e). A possible explanation could be that downstream of the accretion area the soft ice has to override the stiffer meteoric ice. Accretion of softer ice over a long time scale leads to instabilities along the advected plume margin (Fig. 8c, f) resembling the 'fingering' structures described by Wrona et al [27]. A fluid infiltrating a higher viscosity fluid is a complex fluid dynamical problem, meaning that there may not be a simple explanation. Nevertheless, we can state that the plume height highly depends on the viscosity contrast, requiring smaller freeze-on rates to achieve the same plume height than those stated for the base case. We find that, whatever the rheological contrast (within reason), plume structures can be formed by the accretion process.

## Discussion

In summary, our arguments regarding the interplay between ice flux and PiDMP through the freeze-on rate $\dot{f}$, which is controlled by the basal water flux $Q_w$, the freeze-on index $\Phi$, and the location of the observed plumes, support the hypothesis of basal freeze-on. Modelled isochrone structures closely resemble features observed in internal layer structures of RES profiles[14,27] and the mapped plumes are located in regions favourable to the freeze-on process and broadly coincide with modelled freeze-on regions[31]. In the interior, the relatively low ice fluxes require small freeze-on fluxes $F$ and the small surface slope $\alpha_s$[32] (Fig. 1d), which readily leads to large negative slope ratios $S$ allowing for larger freeze-on rates $\dot{f}$ (Fig. 7a). On the other hand, towards the

margin, both $|\alpha_s|$[32] and water flux $Q_w$ increase (Figs. 1d and 2 and Supplementary Fig. 2), leading to water being able to flow up significant steeper adverse slopes $\alpha_b$ and consequently leading to substantially larger $\dot{f}$ than in the interior (see Methods section 'Basal freeze-on rates for plane flow', Eq. (20) and Fig. 7b–d and Supplementary Fig. 6). Based on our analysis and in agreement with observations, it seems likely that basal freeze-on plumes are a common feature in ice sheets. Discrepancies in freeze-on rates between models and reality can be explained by differences in topographic resolution, affecting both the water pathways and the range in bed and surface slopes, with the undersampled coarser grid levelling out the extremes in slope.

As accreted ice is to be expected to show a viscosity contrast, thus allowing easier plume growth for both stiffer (over accretion area) and softer ice (downstream of accretion area), freeze-on rates $\dot{f}$ required to generate observed plume heights become significant smaller compared to $\dot{f}$ calculated for no viscosity contrast. Therefore, statements about the mismatch between plume size and freeze-on rates[31] are questionable without also considering the effect on the relationship between plume-height and freeze-on rate both from the flux relationship and rheology contrast. Moreover, plume growth alters the age-depth relationship, as bottom layers are lifted and compressed, while protected from basal melt, with important implications for ice-core drilling, interpretations, and the search for a site to core at least 1.5-million-year-old ice[37]. A further consequence of basal freeze-on is its influence on ice-sheet mass balance and dynamics, which is currently not accounted for in traditional mass-change assessments and ice-sheet models. Ultimately, drilling into and

sampling of such structures is needed to improve our process understanding.

## Methods

**RES data**. Based on RES data of NASA's Operation IceBridge flights[4] from 2010 to 2014 over northern Greenland (Fig. 1a), we mapped the apex of 674 plume-like structures that seem to rise from the bed (Figs. 1 and 2 and Supplementary Figs. 2 and 3). Such structures with relative heights $h$ larger than one-third of the total ice thickness $H$ are referred to as large plumes in this paper and marked as filled triangles, where filled dots refer to structures <$1/3H$ (see Supplementary Fig. 1, for examples, see Methods section 'Estimating correlation between freeze-on index and plumes' and Supplementary Table 1). A consequence of mapping the apex of the 2-D plume-like structure as seen in a RES profile and not the apex of the 3-D structure is that a single plume shows up as a plume on multiple flight lines, leading to along-flow alignment for advected structures when plotted on a map. Further, the accuracy for larger plumes is expected to be better, since their scale leads to distinct layer structures in RES profiles.

The spatial accuracy of the RES profiles used has an along-track resolution of ≈55 m (30 m sampling) and a cross-track resolution ranging between 420 and 740 m[4–6], constrained by the pulse-limited footprint of the chosen antenna structure. The error in vertical thickness is ≈10 m for an ice thickness of 2000 m[4,5].

**Numerical models**. In this paper, the model used in general to investigate the ice-sheet isochrone architecture is a forward, 3-D, time-dependent, finite-difference stream-tube model (surface and bed are fixed in time), using simplified flow physics, capable of calculating tracers and isochrone layers within the ice[17,25]. As it uses the SIA, which ignores all horizontal deviatoric stress gradients, we use an FS model, which solves all stress gradients (full-Stokes equations) and therefore gets around the difficulties of the SIA model. Owing to its high computational cost, we use the FS model in this paper only in plane flow and the SIA-based model mainly in two dimensions.

**3-D stream-tube model**. As an input, the model requires surface and basal topography, accumulation and melt (freeze-on) rate to calculate the velocities using balance fluxes, prescribing the velocity distribution in the vertical by a shape function. Here we use a surface that is fixed in time, and a shape function for velocity and ice flux that arise from the SIA and prescribe balance fluxes at the inlet marginal boundaries (Dirichlet boundary condition)[17,25].

**2-D FS ice-flow model**. As a check on our simplified model results, we use the Elmer/Ice model[38] solving the Stokes equations describing ice-flow mechanics (full ice-flow equations, FS). We use this computationally much more expensive flow model in two dimensions only to compare the internal layers qualitatively with the ones obtained by the simplified model. To avoid the comparison of code dependencies, we ran the finite-element Elmer/Ice code both by solving for the SIA and the FS using a fixed surface—as for the case of the simplified model, applying Dirichlet boundary conditions as before. Additionally, we ran the FS with a freely adjustable surface. The differences in the modelled 2-D isochrone structures are small for both FS experiments, with largest differences at the ice divide and, for the free-surface case, towards the surface (Fig. 4a). The characteristic sharp upward bend of the internal layers over the basal-accretion zone (decreasing towards the surface) and the folded meteoric ice layers in front of the plume is seen in all model runs. The good agreement between the SIA model and fixed-surface FS model can be explained by large length-to-thickness ratio, which is the underlying assumption for the SIA, and the simple bed topography. Compared with the free-surface FS experiment, the increasing misfit towards the surface stems from the differences due to the freely adjusting surface. The expression of the plume at the surface in the form of a bump over the accretion area (Fig. 4b) leads to a reduction in surface slope $\alpha_s$ upstream of the accretion area and an increase in $\alpha_s$ over the accretion area with its maximum at the downstream end (Fig. 4c).

**Governing equations**. The ice flow is calculated using a Cartesian coordinate system $\mathbf{r} = (x, z)$ for the 2-D vertical plane-flow model or $\mathbf{r} = (x, y, z)$ in the 3-D space, with $z$ pointing upwards, and the time coordinate $t$. The horizontal coordinates are denoted by $\mu = (x)$ and $\mu = (x, y)$, respectively. The surface is given by $z = s(\mu, t)$, the bed by $z = b(\mu, t)$, and the total ice thickness by $H(\mu,t) = s(\mu, t) - b(\mu, t)$.

The relevant equations for the steady-state stream-tube model are:

$$\mathbf{u}(\mathbf{r}) \cdot \nabla X(\mathbf{r}) = 1 \, (\text{age equation}), \tag{2}$$

$$\nabla \cdot \mathbf{Q}(\mu) = \dot{a}(\mu) + \dot{f}(\mu) = \nabla \cdot \mathbf{B} \, (\text{global mass balance}), \tag{3}$$

$$\nabla \cdot \mathbf{M}(\mu) = \dot{a}(\mu) \, (\text{meteoric mass balance}), \tag{4}$$

$$\nabla \cdot \mathbf{F}(\mu) = \dot{f}(\mu) \, (\text{accretion ice mass balance}), \tag{5}$$

where $\mathbf{u}$ is the ice-flow velocity vector, $X$ the age, $\mathbf{Q} = [Q_x, Q_y]$ the total horizontal ice flux, which is the same as the balance flux $\mathbf{B}$ in steady-state, $\mathbf{M} = [M_x, M_y]$ the ice flux arising from meteoric accumulation, $\mathbf{F} = [F_x, F_y]$ the ice flux arising from freeze-on, $\dot{a}$ the meteoric-accumulation rate and $\dot{f}$ the freeze-on rate, where for melt, $\dot{f} < 0$. By steady state, we refer to the ice thickness being unchanging in time; as will be seen, we can evolve the isochrones architecture within this fixed geometry.

By adding Eqs. (4) and (5) and comparing the result with Eq. (3) we can deduce

$$\nabla \cdot \mathbf{Q} = \nabla \cdot \mathbf{M} + \nabla \cdot \mathbf{F} = \nabla \cdot \mathbf{B}, \tag{6}$$

where $\nabla \cdot$ is the divergence operator.

Transient internal layers of age $X$ are obtained by solving the time-dependent age equation:

$$\partial_t X + \mathbf{u} \cdot \nabla X = 1. \tag{7}$$

To avoid arithmetical issues that arise in steady state from predicted infinite ages at the ice-sheet base, we prescribe a small basal melt rate (here 5 mm a$^{-1}$ [39]) as background melt rate, except along freeze-on patches.

**Considerations of numerical age modelling**. In our ice-flow models, internal layers represent isochrones and are displayed by plotting the contours of the time since deposition of the ice. They are a product of the spatial and temporal varying surface-accumulation and basal freeze-on rates as well as of the velocity distribution within the ice sheet (Figs. 3 and 5 and Supplementary Figs. 4 and 5). The boundary condition for the age equals zero years for both accumulation at the surface and freeze-on at the bed. The age of the accreted water is not known, since the water arises from melt of ice, but presumably mixes under the ice sheet with other waters of different age. In numerical models, this age discontinuity gets smoothed out, where accreted ice towards the meteoric ice boundary (Figs. 3g, j and 5e, f) becomes older than the duration of the freeze-on process and the oldest meteoric ice becomes younger. However, as the method of particle tracking uses the velocity field instead of solving the age advection equation, its result is hardly affected by numerical diffusion, which allows us to reproduce the plume shape and accurately define the boundary of accreted ice (red line around plume in Figs. 3f, i and 5a, b).

**Particle tracking**. We employed particle tracking in order to test independently the isochrone structure obtained from the age advection Eq. (2). For each model time step the velocity field is interpolated using bilinear finite-element shape functions. Particle tracks were calculated with a forward Euler scheme using substeps for each model time step. By backtracing the tracks we tested the suitability of the step size. We tracked layers of individual starting points through the evolving velocity field to determine the shapes of the deformed age layers.

**Ice-flux relationship with basal freeze-on**. As in this paper the relationship of ice flux and basal freeze-on is considered mainly in the direction of the steepest surface gradient, we continue by writing the equations for the vertical plane-flow model. Integrating Eq. (6) from the divide downstream, we obtain the 1-D balance flux

$$Q(x) = M(x) + F(x) = \int_0^x (\dot{a} + \dot{f}) \, \mathrm{d}x = B(x). \tag{8}$$

At any position $x$ the ice flux

$$Q(x) = \bar{u}(x) H(x) \tag{9}$$

can be calculated from the product of the depth-averaged flow velocity $\bar{u}$ and the ice thickness $H$, where $Q(x) = B(x)$ in steady state.

The surface velocity $u_s$ can be split into the contribution coming from basal sliding $u_b$ and ice deformation $u_d$. For the SIA (ice flow given by local surface slope $\alpha_s$ and ice thickness $H$), the surface velocity is

$$u_s = u_b + u_d = u_b + \frac{2A}{n+1} \tau_b^n H, \tag{10}$$

where $\tau_b = \rho g H \alpha_s$ is the basal shear stress, $A$ is the rate factor and $n$ the flow-law exponent (used here as $n = 3$)[35,36]. It follows that the partial flux $q(z) = \int_0^z u(z') \mathrm{d}z'$ through a vertical ice column between the bed and the height $z$ is

$$q(z) = u_b(z - b) - u_d H \left[ \left( \left( 1 - \frac{z-b}{H} \right) - \frac{\left( 1 - \frac{z-b}{H} \right)^{n+2}}{n+2} \right) - \left( \frac{n+1}{n+2} \right) \right]. \tag{11}$$

The total flux $Q$ between the bed and the surface $z = H$ is therefore

$$Q \equiv q(z = H) = u_b H + u_d H \frac{n+1}{n+2}. \quad (12)$$

For the specific cases of uniform plug flow, PF (horizontal ice flow solely due to basal sliding, $u_d = 0$), and internal deformation, ID (ice flow solely due to internal ice deformation, $u_b = 0$), the depth-averaged velocity $\bar{u}$ is obtained from Eqs. (9) and (12), so that

$$PF : \bar{u} = u_b, \quad (13)$$

$$ID : \bar{u} = u_s \frac{n+1}{n+2}. \quad (14)$$

To obtain the flux shape function, it is convenient to work in normalised $z$-coordinates $\zeta = (z - b)/H$ and the normalised partial flux $\omega = q(\zeta)/Q$[25,40], obtained by dividing Eq. (11) by the total flux (Eq. (12)). For uniform plug flow and for internal ice deformation, the profile shape is

$$PF : \omega(\zeta) = \zeta, \quad (15)$$

$$ID : \omega(\zeta) = \frac{(1-\zeta)^{n+2} + (n+2)\zeta - 1}{(n+1)}; \quad (16)$$

the second equation only holds when the SIA is being applied. At the normalised plume height, $h$, the normalised partial flux, $\omega$, equals the balance flux component $F$ (see Eq. (8)) originating from freeze-on relative to the total ice flux

$$\omega(h) \equiv \frac{F}{Q} \equiv R. \quad (17)$$

We can now numerically solve Eq. (17) (combined with Eqs. (15) or (16), respectively) for the normalised plume height $h$, given the flux ratio, $R$, from the balance flux components originating from freeze-on $F$ and $M$ (Fig. 6) and vice versa.

**Calculating freeze-on plume parameters.** For a plume near the ice divide of 115 m height ($H = 3000$ m, $h = 0.038$, $\bar{u} = 2$ m a$^{-1}$, with $Q = 6000$ m$^3$ a$^{-1}$ m$^{-1}$ using Eq. (9)), we obtain an ice-flux ratio $R = 0.0035$ (solving $\omega(h)$ in Eq. (16); see Fig. 6), and a freeze-on flux $F = 21$ m$^3$ a$^{-1}$ m$^{-1}$, which is obtained by the multiplication $R \times Q$ (Eq. (17)). Dividing $F$ by either the accretion extent or the freeze-on rate $\dot{f}$, results in $\dot{f}$ or the accretion extent, respectively. Not considering the effect of ice flux Dow et al.[31] made a simple plume-height calculation by multiplying the freeze-on rate of $\dot{f} = 1.5$ mm a$^{-1}$ with time, resulting in a height of only 15 m over 10 ka. Whereas including ice flux, a plume height of 115 m with a freeze-on rate $\dot{f} = 1.5$ mm a$^{-1}$ requires an accretion extent of 14 km.

**Basal heat balance.** It is beyond the scope of the paper to calculate the ice-sheet heat balance at the base including the effects of surface accumulation, but the calculation or reliable estimation of $\dot{f}$ is pertinent to the aims of the paper. This is done in glaciology by a detailed thermodynamic calculation of the ice-bed interface, with the energy conservation statement

$$G_w + G_s + G_g + G_p + G_f + G_c = 0, \quad (18)$$

where $G_w$ is the heat generated by subglacial water flow, $G_s$ is the heat generated by sliding, $G_g$ is the geothermal flux, $G_p$ is the heat required to maintain the flowing water at pressure melting point, and $G_f$ is the heat released by freezing or used by melting; the heat $G_c$ conducted away in the ice is ignored in this paper. The signs of all the $G$ components are consistent, that is, a positive sign is heat being added to the system, a negative sign is removal of heat. By ignoring $G_c$ we underestimate the freeze-on rate.

Rearranging Eq. (18) using the terms corresponding to the $G$ components (labelled with corresponding subscript), we obtain a general expression for the freeze-on rate $\dot{f}$

$$\dot{f} = -\frac{1}{L} \left[ \left( \overbrace{-\mathbf{Q}_w \cdot \nabla\psi}^{w} + \overbrace{\mathbf{T} \cdot \mathbf{u}}^{s} + \overbrace{-k_r \partial_z \theta|_{z=b-}}^{g} \right) \right.$$
$$\left. + \left( \overbrace{Q_w JC\rho_i g \partial_x H}^{p} + \overbrace{k_i \partial_z \theta|_{z=b+}}^{c} \right) \right], \quad (19)$$

where $L$ is the volumetric latent heat of freezing, $\mathbf{Q}_w$ is the water flux, $\nabla\psi$ is the hydrostatic-potential gradient, $\mathbf{T}$ the basal tangential traction, $\mathbf{u}$ is the ice velocity, $k_r$ and $k_i$ is the conductivity of the rock and ice material respectively, $\partial_z \theta$ is the

vertical temperature gradient, and $b-$, $b+$ refer to elevations an infinitesimally small distance below and above the bed respectively. Further, $Q_w = \mathbf{Q}_w \cdot \nabla\psi/|\nabla\psi|$ is the water flux taken along the steepest gradient of the hydrostatic potential, $J$ is the volumetric heat capacity of water, $C$ is the Clapeyron constant for water, $\rho_i$ is the ice density, $g$ is the gravitational acceleration and $\partial_x H = (\alpha_s - \alpha_b)$, with $\alpha_s$ and $\alpha_b$ as the surface and bed gradient, respectively, is defined along the steepest surface gradient. Note the relationship of the water flux $\mathbf{Q}_w = -\kappa \nabla\psi$ with the hydraulic conductivity $\kappa$ along the hydrostatic potential, ensuring that $G_w > 0$. However, in this paper we do not need to know $\kappa$, since we assume a basal water pressure $P_w$ that is near or at the ice overburden pressure $P_i = \rho_i g H$ leading to a spatial uniformity of the effective pressure (defined as $P_{eff} = P_i - P_w$).

The terms in Eq. (19) in the first bracket on the right-hand side with labels w, s, g are always positive, and cause $\dot{f}$ to be negative (i.e. cause melting). The terms in the second bracket, with labels p, c can either be negative or positive—in fact, $G_c$ is generally negative; terms p and c can, if sufficiently large, permit $\dot{f} > 0$. Therefore, by ignoring $G_c$ we underestimate the freeze-on rate.

**Basal freeze-on rates for plane flow.** Following Alley et al.[20], we express the freeze-on rate $\dot{f}$ as a function of basal water flux $Q_w$, bed and surface slope $\alpha_s$ and $\alpha_b$, respectively, along the steepest gradient. Further, we assume the ice temperature at the contact to the bed to be at the pressure melting point and that the released latent heat from freeze-on is warming the basal water, maintaining it at the new pressure melting point, and thus allowing it to flow up the slope. Using Eq. (19) on an adverse slope ($G_p$ becomes negative) and neglecting $G_c$, $\dot{f}$ is obtained from

$$\dot{f} = \frac{(G_p - G_w - G_g - G_s)\cos(\text{atan}\,\alpha_b)}{L}, \quad (20)$$

where $L = 3.06 \times 10^8$ J m$^{-3}$ and $\cos(\text{atan}\,\alpha_b)$ transforms the result $\dot{f}$, normal to $\alpha_b$, to the $z$-coordinate. As $\alpha_b$ in our calculations is so small ($\cos(\text{atan}\,\alpha_b)$ is nearly one) it can be omitted. (Strictly speaking $\alpha_b$ should be taken along the gradient of the hydrostatic potential for $G_p$ and $G_w$ and along the surface gradient for $G_g$ and $G_s$.) The geothermal heat flux $G_g$ used ranges from 0.03 to 0.11 W m$^{-2}$ and the heat of sliding $G_s$ is calculated using the surface velocity $u_s$ to describe the sliding velocity at the base of the ice sheet, thus overestimating $G_s$, so that

$$G_s = \tau_b u_s = \rho_i g H \alpha_s u_s, \quad (21)$$

where $\tau_b$ is the basal drag, the ice density $\rho_i = 916$ kg m$^{-3}$ and the gravitational acceleration $g = 9.8$ m s$^{-2}$.

Expressing the heat terms $G_w$ and $G_p$ in Eq. (19) for plane flow and the change in ice thickness $\partial_x H$ as $(\alpha_s - \alpha_b)$, followed by replacing bed slope with $\alpha_b = S\alpha_s$, we obtain for the water-flow heat source

$$G_w = -Q_w(\rho_i g \alpha_s + (\rho_w - \rho_i)g\alpha_b)$$
$$= Q_w \rho_i g \overbrace{(-1)\alpha_s(1 + (\rho_w/\rho_i - 1)S)}^{\bar{G}_w}, \quad (22)$$

and for the heat term required to maintain the water at the pressure melting point

$$G_p = Q_w JC\rho_i g(\alpha_s - \alpha_b)$$
$$= Q_w \rho_i g \overbrace{\alpha_s(1 - S)JC}^{\bar{G}_p}, \quad (23)$$

where $\rho_w = 1000$ kg m$^{-3}$ is the water density, the volumetric heat capacity of water $J = 4.2 \times 10^6$ J m$^{-3}$ K$^{-1}$, the Clapeyron constant $C = -7.4 \times 10^{-8}$ K Pa$^{-1}$, and $\bar{G}_w$ and $\bar{G}_p$ the respective, non-dimensional heat term component.

**Basal freeze-on index.** In order to spatially compare on an ice-sheet-wide scale the potential freeze-on purely related to the PiDMP mechanism, we define the basal freeze-on index $\Phi$ as an entirely geometrical quantity, which reflects the basal topography relative to the surface topography. Since the mechanism of PiDMP requires water flow along an adverse bed slope, we only take the heat terms $G_p$ and $G_w$ in Eq. (20) that depend on the water flux $Q_w$ and the bed-to-surface-slope ratio $S$ to calculate $\Phi$. The heat from the sum of the neglected heat sources $G_g + G_s$ is comparable to the difference of $G_p - G_w$ calculated for small water fluxes (i.e. $Q_w = 0.01$ is m$^3$ s$^{-1}$ m$^{-1}$ width) and ratios $S \geq -5$, even for increased ice flow (i.e. 100 m a$^{-1}$). For large water fluxes (i.e. $Q_w = 0.11$ m$^3$ s$^{-1}$ m$^{-1}$ width); however, the difference $G_p - G_w$ is one to two orders of magnitude larger (Supplementary Fig. 6a). By neglecting the two heat sources $G_g$ and $G_s$, we implicitly assume that they are being balanced by heat conduction towards the cooler ice-sheet surface as well as advection of colder surface-ice along flow.

To obtain the freeze-on index $\Phi$ as a non-dimensional quantity, we rewrite Eq. (20) using the terms $G_p$ and $G_w$ only (ignoring $G_g$ and $G_s$) and introducing Eqs. (23) and (22), respectively, to obtain

$$\dot{f} = \frac{(G_p - G_w)\cos\alpha_b}{L}$$
$$= \frac{Q_w \rho_i g}{L}(\bar{G}_p - \bar{G}_w)\cos\alpha_b, \quad (24)$$

and define $\Phi$ as

$$
\begin{aligned}
\Phi &= (\tilde{G}_p - \tilde{G}_w)\cos\alpha_b \\
&= \alpha_s[(1-S)JC + 1 + (\rho_w/\rho_i - 1)S]\cos\alpha_b,
\end{aligned} \qquad (25)
$$

where $\Phi$ now depends largely on bed and surface slope and scales with $Q_w$ (Supplementary Fig. 6). The quantity $G_w$ is the heat generated by subglacial water flow, while $G_p$ is the heat needed to maintain the water at pressure melting point. For a given water flux $Q_w$, the freeze-on rate is the result of a simple multiplication between $Q_w$, the ice density $\rho_i$, the gravitational acceleration $g$, the inverse of the volumetric latent heat of freezing $1/L$, and $\Phi$

$$
\dot{f} = \frac{Q_w\rho_i g}{L}\Phi. \qquad (26)
$$

Comparing the non-dimensional heat terms $\tilde{G}_w$ and $\tilde{G}_p$ (Supplementary Fig. 6b; Eqs. (22) and (23), respectively) for a range of $-2 \geq S \geq -11$ shows that the heat taken up by $\tilde{G}_p$ to warm the water increases with decreasing $S$, while the heat emitted by $\tilde{G}_w$, due to water flow, decreases towards zero. As the increase in $\tilde{G}_p$ is greater than the decrease in $\tilde{G}_w$ for decreasing $S$, the heat difference $\tilde{G}_p - \tilde{G}_w$ is increasing, leading to increased freeze-on in order to balance the heat terms (Supplementary Fig. 6b).

The calculation of the freeze-on index $\Phi$ depends highly on the bed and surface slope $\alpha_b$ and $\alpha_s$. However, the interpolation of the RES data to obtain the bed topography leads to large vertical errors in bed elevation exceeding 600 m in regions of sparse radar data (see Supplementary Fig. 3 in Morlighem et al.[23]). A minimum error of 200 m allows us to nearly continuously visualise all used radar transects leading to a minimum transect width of $\approx$4 km in regions of sparse data. Increasing the minimum transect width to 5 and 10 km leads to a maximum error of 225 and 300 m, respectively. Note that $\Phi$ and the water path are consistent as the same data set for bed and surface is used, so that with each improved topography data set the $\Phi$ and water paths become more reliable.

**Estimate of water paths**. To visualise the direction in which water tends to flow in North Greenland, we use a simple approach by assuming that water is available everywhere and that it freely follows the steepest gradient of the hydrostatic potential. This is done by using the MATLAB streamline function[41] with a 3 or 5 km step size and starting points (seeds) at a regular spacing (e.g. every 12 km) on a 1050 m gridded Greenland ice-sheet surface and bed topography[23,26] (see Fig. 2 and Supplementary Figs. 2 and 3). The uncertainties in water paths reflect the uncertainties in the bed and surface topography, which are largest for the bed topography in areas of sparse radar data[23] (see also Methods section 'Basal freeze-on index').

The majority of the generated water paths (streamlines), transporting the subglacial water, have their source in the central region of the ice sheet (Fig. 2 and Supplementary Fig. 3a), the main subglacial basin (Fig. 1b), in a zone of elevated geothermal heat flux ($G_g$) ranging from 70 to 106 mW m$^{-2}$ [21]. Within this zone, high basal melt rates have been inferred from internal layering along the ice ridge (around the NGRIP ice-core drill site), ranging between 0.005 and 0.02 m a$^{-1}$ [39], and towards the onset of NEGIS on local spots leading to rates exceeding 0.1 m a$^{-1}$ [42]. Comparing our mapped plumes with a composition of areas with a temperate bed[22], we find that most plumes are within areas of a 'likely thawed' or 'uncertain' bed, with water paths seeded from both areas reaching most plumes (Supplementary Fig. 3b, c).

**Estimate of basal meltwater**. A rough estimate of the basal meltwater catchment area feeding a series of large plumes (Fig. 3a and Supplementary Fig. 1; area outlined in magenta in Supplementary Fig. 2) is shown in Supplementary Fig. 3a–c (black outlined area of $10^3$ km$^3$). We assume an elevated basal melt rate ranging between 0.01 and 0.02 m a$^{-1}$ over this catchment area, as it lies within the inferred zone of elevated geothermal heat flux[21] and is affected by the high melt rates around NGRIP (west margin)[39]. To obtain the source area, we seed at 1050 m along the 6 km accretion length (observed in RES profile) and 15 km plume upstream along ice flow, and calculate streamlines along the reversed (upwards) hydraulic gradient using steps of 5 km (Supplementary Fig. 3d).

For the outlined catchment area and the relevant basal melt rates, we obtain a total water flux of $10^4$ km$^2 \times (0.01/1000)$ km a$^{-1}$ = 0.1 km$^3$ a$^{-1}$ = 3.2 m$^3$ s$^{-1}$ and 0.2 km$^3$ a$^{-1}$ = 6.3 m$^3$ s$^{-1}$, respectively. This quantity does not include water flux from melt production ($G_w$ and $G_s$) along the flow path.

**Estimate of freeze-on rate and water flux**. Considering ice-flux dependencies, we can deduce the freeze-on rate ($\dot{f} \approx 0.8$ m a$^{-1}$) over a given accretion extent (6 km), required to produce the observed plume height (Fig. 3a). From the relative plume height ($h \approx 0.4$, Fig. 3a), we obtain an ice-flux ratio ($R = 0.27$, using internal deformation, Eq. (16)) from its relationship with the normalised partial flux $\omega(h) = R = F/Q$ (Eq. (17); see Fig. 6). Further, we obtain the basal freeze-on ice flux ($F = 4752$ m$^3$ a$^{-1}$ m$^{-1}$) from the total ice flux ($Q = 17,600$ m$^3$ a$^{-1}$ m$^{-1}$), which results from Eq. (9) using the surface velocity[43] ($u_s = 10$ m a$^{-1}$) and total ice thickness[4] ($H = 2200$ m) at the plume location.

Using the relationship of the freeze-on rate $\dot{f}$ with the water flux $Q_w$ and the slope ratio $S = \alpha_b/\alpha_s$ (Eq. 20), we calculate the required water flux $Q_w$. We estimate the local surface and bed slopes extending over the plume as $\alpha_s = -0.003$ and $\alpha_b = 0.025$[23,26]. Along the corresponding bed-slope ratio $S = -8.33$, a water flux of $Q_w = 0.11$ m$^3$ s$^{-1}$ m$^{-1}$ (Eq. 20 and Fig. 7b, black line) is needed to produce the estimated freeze-on rate $\dot{f} = 0.8$ m a$^{-1}$.

Finally, assuming a water body height of 1 m, we obtain maximum channel widths ranging between 29 and 58 m for the available basal meltwater flux (3.2 to 6.3 m$^3$ s$^{-1}$) passing at a flux of $Q_w = 0.11$ m$^3$ s$^{-1}$ (water velocities of 0.11 m s$^{-1}$). Observations inferred from RES data of Thwaites Glacier in Antarctica suggest that there are both channels and distributed conduits[44]. However, a distributed hydraulic system seems more likely, as on an adverse slope, steep enough for freeze-on to occur, a channelised system is not sustainable when modelling the subglacial hydrology of an overdeepening, and it therefore shuts down[45].

**Along-flow reduction in water flux**. To freeze-on water at a rate of 0.8 m a$^{-1}$, the corresponding basal freeze-on flux is $F \sim 2.54 \times 10^{-8}$ m$^3$ s$^{-1}$ m$^{-1}$ width, which is 7 orders of magnitude smaller than the required water flux $Q_w = 0.11$ m$^3$ s$^{-1}$ m$^{-1}$ width. Over a length of 6 km this leads to a reduction of $Q_w$ by $1.52 \times 10^{-4}$ m$^3$ s$^{-1}$ m$^{-1}$ width, a negligible amount, allowing for sustained large water fluxes downstream.

**Estimating correlation between freeze-on index and plumes**. To quantify the relation between plumes and areas with freeze-on index, we follow streamlines along the reversed (surface) ice-flow gradient starting at each plume, and determine if they cross areas of freeze-on index $\Phi$. Further, we compare the results between the mapped plume sets ($\geq H/3$ and $< H/3$) with the averaged results for 10,000 randomly chosen locations. For the reversed streamlines calculated on a 1050 m grid, as used in this paper, we find the best match between plumes and $\Phi$ areas for the large plumes, followed by the small plumes and the random set, which are reduced by up to 8% and a further 9%, respectively (see Supplementary Tables 1 and 2). By comparing this difference between the observed and random data set relative to the total random mismatch (the percentage of plumes that are predicted to have no freeze-on area), we find that the observed data results in a distinctively better match than the random data set (Supplementary Table 2; e.g. the mismatch is reduced by >35 and >17% for plumes with heights $\geq H/3$ and $\geq H/3$, respectively). Consequently, we argue that the plume distribution is related to the pattern in $\Phi$, since if in the observational data set the plumes are randomly distributed we would expect a similar mismatch between the data sets.

Calculating reverse streamlines on a 150 m grid[26] along a 5 km stretch, for seeds within 1.5 km radius from the plume, about 98 and 94% of the large and small plumes, respectively, and <85% of the random locations are reached by ice flowing over areas of $\Phi$. This result depends on the grid, the seed area and the streamline distance. Reverse streamlines calculated on the 1050 m grid, seeded over an area of 3 km along 10 km, result in comparable values as for the fine grid (1.5 and 5 km). Larger RES mapping errors for small plumes (see Methods section 'RES data') partly explains the better result for plumes >$H/3$.

## Data availability

The freeze-on index $\Phi$ and the plume location data together with the model code that support the findings of this study have been deposited in the zenodo repository with the identifier: https://doi.org/10.5281/zenodo.1435749. All radio-echo sounding data are publicly available under CReSIS data products (https://data.cresis.ku.edu/data/rds/) or under IceBridge data on the National Snow and Ice Data Center (https://nsidc.org/data/icebridge/data_summaries.html). All other observational data sets are publicly available (see corresponding references for details). A reporting summary for this Article is available as a Supplementary Information file.

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

## Acknowledgements

G.J.-M.C.L.V. and M.P.L. were funded by the Faculty of Science, University of Zurich, Switzerland. While at Department of Geography, Durham University, UK, G.J.-M.C.L.V. was funded by a Royal Society BP Dorothy Hodgkin Fellowship. C.M. and R.C.A.H. were partly supported as part of the Polar Science for Planet Earth funding from the Natural Environment Research Council (NERC) to the British Antarctic Survey, UK. We acknowledge the use of data and/or data products from CReSIS generated with support from the University of Kansas, NASA Operation IceBridge grant NNX16AH54G, and NSF grant ACI-1443054. We thank Andreas Vieli for many constructive discussions and comments.

## Author contributions

G.J.-M.C.L.V. conceived the idea and designed the study, extended and applied the existing models, performed all analyses, carried out the plume mapping and wrote the manuscript with suggestions from all authors. C.M. extended the FS model using finite-element software Elmer/Ice and performed the FS modelling; R.C.A.H. developed the stream-tube model; M.P.L. developed and applied the tracking model.

## Additional information

**Competing interests:** The authors declare no competing interests.

