## [Peer Review File · Nature Communications]

Reviewers' comments:

Reviewer #1 (Remarks to the Author):

The manuscript by Leysinger Vieli and al. present advances in the understanding of the formation of large plume-like features within ice layers, and previously identified from radar images both in Greenland and Antarctica. Although apparently common, the origin of these features remains unknown, and thus, their impact on ice dynamics and mass balance.

This is a very interesting manuscript, with the most significant new result being that a plume height (several hundreds of meters) relates first to ice flux and only secondarily to freeze-on. This allows the authors to provide a physical explanation for large plume formations, even where basal freeze-on is low.

The manuscript is generally well written, but needs additional details in place. This would significantly improve clarity – and thus help strengthen their results at times (see first comment below in particular).

Overall, this is great work that merits publication. My comments below mainly aim at improving clarity on the work done.

- L87-90: These lines introduce a key output from the study, yet there is not enough information for the new processes to be clear. I would suggest replacing “can be explained using mass-conservation principles and ice mechanics” with a paragraph providing more details on how these come into play. This would also help clarify the link to the ratio R , which is not immediately obvious, and needs to be expanded upon (in other words, you show on Fig. 3c/d that plume height and ratio R are connected, but can you better explain why this is the case?).

- From Figure 1c / Supp Figure 1, it is difficult to see the good correspondence between the plume location and the places of crossing between subglacial water paths and elevated PHI, as it seems that quite a few do not fulfill both conditions. Would it be possible to specify in the text the proportion of plumes that are found to meet both criteria?

- Overall, the method section is very long, which makes the reference to “Methods” not very helpful. Suggest adding numbering to the method subsections, and referring to specific method subsections throughout the manuscript.

-L 59/60: You should state here that you are using the SIA approximation, and why this is appropriate in your study. This could also be a good place to introduce the fact that you have additional runs to test the sensitivity of the model to the assumed simplified flow physics.

- L.67 / L.85: specify the freeze-on rates and area used for the example. Actually, if there is space, you could provide a table that summarizes the key model parameters / boundary conditions used for the result presented. Currently, the info is a bit scattered in the method section.

- This is a bit confusing: are you assuming in all your simulations that the ice moves purely by internal deformation? Or are you using eq. 6 from the methods section, which takes into account both basal sliding and internal deformation? If the former, this could be mentioned earlier on, e.g., as part of a model set-up description.

-L. 110-111: You would expect that basal sliding increases for ice flow downstream. Shouldn't this also contribute to the smaller relative height of the plumes, according to Fig. 6? If that is the case, even if it is a secondary effect, you should include it in your interpretation.

- L113-119: this paragraph seems a bit out of place. It needs better integration (to the following paragraph, presumably?).
- L126-132: Quantifying the effects presented in this paragraph would strengthen its statement. Is the feedback mentioned important or not?
- L. 142: similar comment to L. 110-111; what about the effect of increased basal sliding?
- L.145-146: This first sentence isn't clear.
- Finally, there is a lack of discussion on how the new findings fit / relate to existing work on the formation of these plumes. I would suggest adding details either in the intro and/or conclusion.

Figures:

Overall, I find the figures to be of good quality. They are however showing a lot of information, and the captions are not sufficiently clear to easily understand what is shown. Below are a few suggestions to improve on this.

-Fig. 1:

Panel c, it is hard to see the ice core locations. Consider using a different color, or larger symbol?
Caption: The radar flight lines can only be seen on panel c. The mention of "grey thin lines" should be moved to that particular section.

-Fig. 2:

Caption: consider grouping the information per panels a bit more. For example, most panels are listed 3 times, which doesn't facilitate following what is happening on each panels. In particular, I find the last three lines confusing (after "f and h"): are the blue / red / and orange lines mentioned corresponding to the vertical profiles? (if so, orange line is on panel I only)? Which solid and dashed-dotted lines are being referred to?
Also, I would specify more clearly which is the accretion area: maybe use "red horizontal bar" – as there are other red lines in the panels.

-Fig. 3:

You could make clearer from the beginning, the fact that the 10ka run is a transient run, and the 200ka run is a steady state run (for example, you could state this in you're a and b captions). Furthermore, I suggest removing the references to a and b in your c caption, which I find confusing. Instead, I would replace references to a and b in the panel c legend with 10ka / 200 ka, accordingly (so, the legend for the dashed blue line would read F: transient (10 ka), etc...)
Similarly, I would remove the reference to a and b in your d caption, and modify so that it reads something like "..., shown for both 10 ka (black dashed-dotted line), and 200 ka (grey solid line)."

Reviewer #2 (Remarks to the Author):

Review of Basal freeze-on at the origin of complex internal ice-layer stratigraphy.

Title: Might consider clearer title: Extent of basal freeze-on in Northern Greenland

Overview:

This paper works to address systematically a very intriguing question --- how much of the ice at the base of the Greenland Ice Sheet is refrozen basal water and how does the refreezing work. This is a nice integration of data analysis and modeling. The authors examine the pressure-induced

depression of the melting point (PiDMP). Their major conclusion is that the features may be comprised of freeze-on and that the ice flux is the primary control on their height. The major claims of this paper are novel as several groups have been trying to isolate the source of these structures in the ice sheet. This work will be of interest to the ice rheology, ice core and eventually the ice sheet modeling communities. I have noted my concerns with the paper framing and the figures.

I have concerns about Figure 1 have some suggestions on how to improve it. They should address why their hydrologic pathways differ from other published pathways. Others have indicated these features are along pathways --- it curious that theirs differ.

"Thus, our spatial analysis supports the hypothesis of basal freeze-on due to ascending water for Greenland."

In the summary section the authors note the presence of plumes in the regions of fast flow but do not address the distinct difference between Petermann and NEGIS. In Petermann there are plumes in the onset region while in NEGIS there are not. (line 140).

Other Comments

The authors omit mentions one of other primary difference between Antarctica and Greenland – the well defined presence of water. In Antarctica there is a clear association with the basal plume and the subglacial water network. In Greenland the ability to map subglacial water is only recently been resolved. Currently, we do not know the distribution of subglacial water in Greenland and its relationship to plumes.

Figures:

Figure 1 is terribly complicated and difficult to follow. I could recommend several revisions. (a) to many lines in 1c. I suggest moving to two separate panels first showing plumes and heat flow anomalies (maybe) on flightlines and ice surface then another panel with plumes on top of hydrology, velocity arrows and contours, (b) to many circles ---- using x as symbol for the plumes would link to labeling on radar cross sections better and simplify somewhat (c) not clear what the red circles are referring to... neither Fahnestock et al (#35) nor Rogozhina et al (#34) seem to have this pattern??? (d) the important new observation to present is the freeze-on index that is almost lost in these plots... if should be a bolder color or on top or a more transparent hydrology. Please label with lat, lon somewhere. I know this is not very trendy but at some point someone will try and figure out where these points are and northing and easting just will not be appropriate. Is there an offset between the blue box and the location of the modeled structures ???

I am concerned about the drainage pathways presented. These are very different than the drainage pathways for Greenland published by Livingston et al 2013 Cryosphere or Bell et al 2014 in Nature Geoscience

Figure 2 is lovely and captures the essence of this work demonstrating the potential for the process to modify the age-depth relationship in the ice sheet and to deform the stratigraphy.

Methods

Supplemental Figures

Linkage to supplementary figures weak and non-sequential. Supplementary Figure 2 is first to be mentioned.

Sup. Figure 1: similar issues to Figure 1. Magenta arrow confuses me... To make the case that there is a relationship between plume and freeze-on index how about a plot of plume heath versus freeze-on index to compliment the map???

Sup. Figure 2: where is this focused region?? Again is it the magenta arrow on previous figure... if so why are there a different number of flight lines and circles. For completeness I would like to see the two radar profiles upstream of 2a where there is supposed to be a plume and one without.

Reviewer #3 (Remarks to the Author):

Review:

Leysinger Vieli et al., "Basal freeze-on at the origin of complex internal ice-sheet layer stratigraphy"

The paper presents results from two ice-flow models, demonstrating that with sufficient material added to the base of an ice sheet it is possible to form structures similar to those observed in radar data from the Greenland Ice Sheet. The authors further argue that this process may influence mass balance estimates and flow properties of the ice sheet.

The results are interesting, in particular the use of a 3D model allowing for different viewing geometries, and the model run with a freely evolving surface. Having said that, I have several reservations outlined below regarding the interpretation and conclusions of the paper. I would like to preface this by saying that I agree that the modelled plumes look very similar to the observed plumes but other studies (Wolowick et al. 2014, Bons et al. 2016) have also obtained similar plume structures invoking different processes. Therefore, similarity with observations is not sufficient to imply causation.

Correlation between freeze-on index and plumes

The correlation in Fig. 1 (and supplementary Fig. 1) between the areas where streamlines cross a high "freeze-on index" and the location of a "plume" is not convincing. There are numerous areas where the streamlines cross a high freeze-on index but no plume is observed in the radar data (e.g. east of Petermann glacier) or where plumes are observed but the freeze-on index is low (the plumes west of the magenta arrow in Fig.1). Other studies (Panton and Karlsson, 2015) have mapped significantly more plumes and if these locations were included, the correlation might be clearer. At least, all plumes mapped for this study should be included in Fig. 1, at the moment it is not clear why plumes smaller than H/3 are not shown. Are they expected to result from a different process?

Recommendation: Show that there is a correlation between freeze-on index (or rate) and plumes, and include all plumes on the map.

Freeze-on rates and subglacial water discharge

The freeze-on rates necessary to create the plumes require a significant amount of subglacial meltwater. The authors justify the high melt rates by considering different heat fluxes and sources in the basal environment, and then calculate the subglacial water routes from the hydropotential. This seems like a simplified approach considering that several hydrological models have been developed for ice-flow models including Elmer/Ice (de Fleurian et al., 2014). Even so, the simple approach does provide an estimate of freeze-on rates. It is not clear why this estimate of freeze-on rates is not shown in Fig. 1 instead of the freeze-on index. Admittedly, the estimate is uncertain but by prescribing a high melt rate for the whole domain, the authors would get an upper boundary for the freeze-on rates. For the reader the freeze-on index is confusing and not easily translatable into freeze-on rate.

The catchment area for the plume described and modelled in Fig. 2 is outlined in Fig. 1 as a large area close to the ice divide. In Fig. 1, however, only a few of the streamlines that initiate within the catchment area terminates or passes the plume. This suggests (to the reader) that the catchment area for the plume is in fact substantially smaller and by extension so is the amount of available water and thereby the freeze-on rates.

Finally, studies have indicated that at least in the western part of North Greenland, the bed is likely to be frozen (MacGregor et al., 2016) thus it seems unlikely that large amounts of meltwater is passing through the subglacial system in this region. In contrast, in the eastern part the bed is likely thawed. This poses a problem for the freeze-on hypothesis since the plumes to the east of the ice divide would then be a result of a different process than the plumes west of the ice divide.

Recommendation: Show (an upper bound for) freeze-on rates on the map, show that the outlined area is a catchment area for the plume in Fig. 2, and address the problem regarding the different

basal thermal states of the ice sheet.

Ice-flow models

The authors mention that the rheology of the plumes is different compared to meteoric ice. Unfortunately, this is not investigated further. The Elmer/Ice model does have the capability to resolve ice with different rheological properties and it would be extremely interesting to see the impact on ice-flow when a plume builds up with a different viscosity. Would the rheology contrast in itself be enough to sustain the plume (as claimed in other studies (Bons et al., 2016))? By including different ice rheologies the authors would be able to convincingly refute (or demonstrate) that rheology contrasts are important for the plume formation.

Recommendation: Simulate plume formation and ice-flow invoking different ice rheologies.

Minor comments

Fig. 1 is difficult to interpret. The contrast between orange, magenta and red (and blue) is not big enough. Overall, the figure is too busy. A legend would also help the reader.

In Fig. 3d, it is not clear which line belongs to which axis (dotted line is h?).

Line 23: assumed to be isochrones rather than "termed isochrones"

Line 72: Distored = distorted?

Line 109: Missing a space after full stop.

Lines 121 and 122: The use of the >< symbol is not very helpful to the reader and could be phrased differently.

Line 160: Presumably, "oldest ice" refers to the 1.5 million year old ice-core project but this should be stated more explicitly.

References

- Bons, P. D., D. Jansen, F. Mundel, C. C. Bauer, T. Binder, O. Eisen, M. W. Jessell, M.-G. Llorens, F. Steinbach, D. Steinhage, et al. (2016), Converging flow and anisotropy cause large-scale folding in Greenland's ice sheet, *Nature communications*, 7, 11,427. doi:10.1038/ncomms11427
- Wolovick, M. J., T. T. Creyts, W. R. Buck, and R. E. Bell (2014), Traveling slippery patches produce thickness-scale folds in ice sheets, *Geophys. Res. Lett.*, 41, 8895–8901, doi:10.1002/2014GL062248.
- de Fleurian, B., Gagliardini, O., Zwinger, T., Durand, G., Le Meur, E., Mair, D., and Råback, P. (2014): A double continuum hydrological model for glacier applications, *The Cryosphere*, 8, 137-153, <https://doi.org/10.5194/tc-8-137-2014>.
- Panton, C. and Karlsson, N. B. (2015), Automated mapping of near bed radio-echo layer disruptions in the Greenland Ice Sheet, In *Earth and Planetary Science Letters*, Volume 432. <https://doi.org/10.1016/j.epsl.2015.10.024>.
- MacGregor, J. A., et al. (2016), A synthesis of the basal thermal state of the Greenland Ice Sheet, *J. Geophys. Res. Earth Surf.*, 121, 1328–1350, doi:10.1002/2015JF003803.

Response to the Reviewer's comments

We would like to thank the Reviewers for their thorough and excellent comments and suggestions made. We took the revision very seriously and believe that the findings in our manuscript are now stronger and clearer.

Notes to the following response: 'Lxx' refers to the Lines of the originally submitted manuscript (otherwise noted as 'new Lxx'), *slanted black text is the reviewers comments*, **our comments are in orange**, original text in black, **and in blue is the changed or added text as found in the current manuscript**.

Reviewers' comments:

Reviewer #1 (Remarks to the Author):

The manuscript by Leysinger Vieli and al. present advances in the understanding of the formation of large plume-like features within ice layers, and previously identified from radar images both in Greenland and Antarctica. Although apparently common, the origin of these features remains unknown, and thus, their impact on ice dynamics and mass balance.

This is a very interesting manuscript, with the most significant new result being that a plume height (several hundreds of meters) relates first to ice flux and only secondarily to freeze-on. This allows the authors to provide a physical explanation for large plume formations, even where basal freeze-on is low.

The manuscript is generally well written, but needs additional details in place. This would significantly improve clarity and thus help strengthen their results at times (see first comment below in particular).

Overall, this is great work that merits publication. My comments below mainly aim at improving clarity on the work done.

- L87-90: These lines introduce a key output from the study, yet there is not enough information for the new processes to be clear. I would suggest replacing "can be explained using mass-conservation principles and ice mechanics" with a paragraph providing more details on how these come into play. This would also help clarify the link to the ratio R, which is not immediately obvious, and needs to be expanded upon (in other words, you show on Fig. 3c/d that plume height and ratio R are connected, but can you better explain why this is the case?).

Instead of referring only to the Methods we explain in more detail the principal of the flux relationship with regards to mass-conservation and ice mechanics. We continue from L88 (new L106) with the following:

In steady-state the total ice flux (Q) can be expressed by the balance flux (B), which is the sum of the accumulation rate (\dot{a}) and the freeze-on rate (\dot{f}) integrated from the divide to the current location downstream, where for surface or basal melt $\dot{a}, \dot{f} < 0$ (see Methods subsections 2.3 and 3, equations (2),(3) and (5)).

In summary, the total ice flux Q consists of ice flux arising from meteoric accumulation (M) and from freeze-on (F). In a vertical ice column in plane flow, the total ice flux Q is the sum of the meteoric ice flux M , which comes from the surface, and the freeze-on (or accreted) ice flux F , which originates at the bed (Fig. 5c). The relative height of each flux component is expressed by the partial flux through the respective ice column (see Methods section 3, equation (8)), and is a function of the relative contribution of M and B to the total ice flux Q and to the horizontal velocity distribution with depth (Methods equations (6) and (9)). (The relative contribution to total ice flux Q for each flux component is $1 = M/Q + F/Q$.) We define the fraction of the freeze-on flux component as ice-flux ratio

$$R \equiv \frac{F}{Q}, \quad (1)$$

which equals the normalised partial flux for an ice column of normalised height (h ; see Methods section 3, equation (12)).

Along flow, the freeze-on flux F increases only over the accretion area for both the transient and the steady-state case, while the meteoric ice flux M is steadily increasing (Fig. 5c). This along-flow relationship between F and M leads, apart above accretion areas, to a decreasing flux ratio R (Fig. 5d). We find that the factor limiting the plume height is primarily given by the ice-flux ratio R .

- From Figure 1c / Supp Figure 1, it is difficult to see the good correspondence between the plume location and the places of crossing between subglacial water paths and elevated PHI, as it seems that quite a few do not fulfill both conditions. Would it be possible to specify in the text the proportion of plumes that are found to meet both criteria?

The analysis between Φ and the water path would indeed be nice if we knew where and at what rate the water is flowing. However, without this information this analysis adds not much information. This data set is complex and not very suitable for statistics, as plume information is available along radar transects only and all other information is calculated from surface and bed data, interpolated from RES data^{11,12}). For freeze-on to occur we need the water flowing over a freeze-on area (elevated Φ) but the plume seen in the radar might be a bit further away along ice flow. Therefore, what we can do is to check if the ice at the plume comes from a nearby region crossing an accretion area. An argument that the observed plume relates to freeze-on, as the ice comes from an area with elevated Φ . We did that for the plumes $\geq 1/3H$,

< $1/3H$ and a random selection and added a new subsection 5.5 in the Methods section. The table is added to the Supplementary Material (see also answer to Reviewer 3 on correlation between freeze-on index and plumes).

- Overall, the method section is very long, which makes the reference to “Methods” not very helpful. Suggest adding numbering to the method subsections, and referring to specific method subsections throughout the manuscript.

We added numbering to the Methods subsections and now refer specifically to them throughout the text. Furthermore, parts of the method section ‘numerical models’ have now been moved into the main text to improve clarity. However, the Methods section is still long.

-L 59/60: You should state here that you are using the SIA approximation, and why this is appropriate in your study. This could also be a good place to introduce the fact that you have additional runs to test the sensitivity of the model to the assumed simplified flow physics.

As mentioned above the sections removed from the numerical modelling section in Methods have been moved to that paragraph. We replaced L59/60 and added a sentence describing the model:

We use a forward, three-dimensional, time-dependent, numerical ice-flow model conceived as a stream-tube model (bed and surface are fixed in time; see Methods section 2.1) using finite differences. It uses simplified mechanics based on the shallow ice approximation (SIA), which ignores all horizontal stress gradients and is based on the assumption of small spatial stress gradients.

We replace L81-83 to continue with the explanation why the used models are appropriate and introduce the FS model (new L85):

Applying accretion locally at the ice-sheet base results in large horizontal gradients in basal vertical velocity, which might lead to not completely correctly calculated velocities by the SIA-based model, affecting the isochrone structure. To check on the calculated layer structure, we perform additional experiments with a two-dimensional full-system (FS) model solving the equations describing the mechanics of ice flow (full Stokes equations) and a freely-evolving surface (see Methods section 2.2). By solving all stress gradients the FS model avoids the difficulties associated with the simplified mechanics of the SIA model. As the FS model comes at a high computational cost, we use it here in plane-flow only, for validation purposes. These additional experiments confirm the plume dynamics and isochrones patterning obtained by the SIA-based model (Fig. 4a).

A further complication is the jump in age that would occur in nature between the fresh accreted ice and the oldest meteoric ice. For both SIA and FS models this jump leads to numerical diffusion of

age. In order to follow the age discontinuity and to test independently the isochrone structure obtained by solving the age advection equation (Methods section 2.3, equation (4)), we track particles for the SIA-based model taken along the initial layer structure over time using the modelled velocity field (Fig. 3e,h,k; see Methods sections 2.4-2.5).

- L.67 / L.85: specify the freeze-on rates and area used for the example. Actually, if there is space, you could provide a table that summarizes the key model parameters / boundary conditions used for the result presented. Currently, the info is a bit scattered in the method section.

We decided on adding the information directly into the text instead of providing a table. We added on L67 freeze-on area and rate (new L69):

Following this, we simulate local freeze-on by adding mass using a freeze-on rate $\dot{f} = 0.8 \text{ m a}^{-1}$ - of similar magnitude to the surface accumulation ($\dot{a} = 0.1 \text{ m a}^{-1}$) - over a basal accretion area 6 km along flow and 7 km across flow.

And we changed L84/85 adding the information on freeze-on rate and extent for Figure 3 to the following (new L100):

In order to investigate the effect of ice flow on the plume shape, we apply an equal amount of basal freeze-on, using a freeze-on rate $\dot{f} = 0.42 \text{ m a}^{-1}$ (surface accumulation $\dot{a} = 0.14 \text{ m a}^{-1}$) along an extent of 5 km,- at three different locations along the flow-line.

- This is a bit confusing: are you assuming in all your simulations that the ice moves purely by internal deformation? Or are you using eq. 6 from the methods section, which takes into account both basal sliding and internal deformation? If the former, this could be mentioned earlier on, e.g., as part of a model set-up description.

We add to L64 that we assume no basal sliding in all our calculations. On L94 where 'internal deformation' is introduced this is was already mentioned. L64 (new L64) now reads:

The geometrical setting for a typical plume (Fig. 3a,b) is taken from observations (ice thickness, surface slope and distance to ice divide)^{7,12,13} using simplified topography (uniformly inclined surface and a flat bed) while assuming no basal sliding.

-L. 110-111: You would expect that basal sliding increases for ice flow downstream. Shouldnt this also contribute to the smaller relative height of the plumes, according to Fig. 6? If that is the case, even if it is a secondary effect, you should include it in your interpretation.

This is correct, relative plume height depends on the freeze-on flux F and the flux shape function (horizontal velocity distribution with depth). We add a sentence about the relation to the flux shape function to the sentence L110/111 (new L145) and refer to Figure 6.

Downstream, plume heights generally decrease due to the increasing meteoric ice flux M , which is a consequence of contributing accumulation area and flow convergence. In addition, basal sliding becomes more important downstream, affecting the distribution of horizontal velocity with depth, and thus further decreases the plume height (Fig. 6).

- L113-119: this paragraph seems a bit out of place. It needs better integration (to the following paragraph, presumably?).

Here the point is that we don't know much about the water at the base. However, if we look at PiDMP we know that the slope ratio is important and can amplify the freeze-on rate for any waterflux. We rewrote the paragraph (now two paragraphs) for better transition to the next paragraph, and also added a new subsection title.

Relation to basal freeze-on rates \dot{f}

In general, subglacial hydrology at the base of the ice sheet is poorly constrained as a consequence of the logistic difficulties of achieving sufficient observations. However, recent modelling studies suggest the existence of major meltwater pathways, subglacial lakes and, related to the mechanism described as the pressure-induced depression of the melting-point (PiDMP), widely occurring basal freeze-on^{4,8}. The predicted freeze-on rates are in general very low ($< 1\text{mm yr}^{-1}$) and only locally larger rates (a few centimeters to meters) are obtained⁴. Substantial rates (tens of centimetres per year) have been previously suggested as a result of PiDMP, controlled by the adverse bed slope (α_b), surface slope (α_s) and the water flux (Q_w)¹ (see Methods section 4). The process of PiDMP leads to an increase in freeze-on rate \dot{f} both for increases in water flux Q_w and bed slope α_b , and to a decrease in \dot{f} for an increase in surface slope α_s ^{1,15,16}.

As we do not know the water flux Q_w we use the relationship between the bed and surface slope to quantify the effect of PiDMP on the freeze-on rate \dot{f} . Thus, we define the slope ratio $S = \alpha_b/\alpha_s$, and find that for slope ratios ranging between $-2 \geq S \geq -11$ the freeze-on rate \dot{f} not only increases for increasing Q_w but also for increasing α_s (Fig. 7 and Methods section 4.1, equation (15)). The effect of the slope ratios S on the freeze-on rate \dot{f} , as shown in Figure 7a,b, ranges between a few centimeters (for both small α_s and Q_w) to tens of centimeters (for large Q_w) and even meters (for both large α_s and Q_w). It is readily conceivable that locally conditions arise, which lead to freeze-on rates comparable to the ones used in our experiments.

- L126-132: Quantifying the effects presented in this paragraph would strengthen its statement. Is the feedback mentioned important or not?

From the original Supplementary Figure 7 one can see how freeze-on rates change for a change in water flux for a given surface slope, or how the freeze-on rates change for a change in surface slope for a given water flux. We now changed the Figure so that it shows the specific slopes obtained from the full-system experiment. The changes in slope have a little effect on the freeze-on rate as it cancels each other out. But more important is the effect on the hydraulic gradient. In the text we add the observed changes from the specific FS example. L126-130 now read (new L178):

Our model experiment including full-flow physics and a freely evolving surface results in a reduced (increased) surface slope α_s by 2×10^{-4} upstream (downstream) of the freeze-on area and steadily increasing from $2.7 - 3.3 \times 10^{-3}$ over the freeze-on area (Fig. 4b,c), leading to a drop in slope ratio of nearly $2S$ along the bed slope α_b , and hence leading to freeze-on rates \dot{f} that are slightly larger upstream than downstream assuming a constant water flux Q_w (Fig. 7a,b, black line). The difference in \dot{f} between the onset and end of the freeze-on area is a few percent and negligible over the whole area as they counterbalance each other. However, of significance is the effect of both spatial and temporal surface change⁶ on the hydraulic gradient, possibly redirecting the water upstream or allowing water flow over steeper adverse slopes downstream. This complex feedback mechanism is expected to amplify and extend plume growth over initially unfavourable α_b , while changing the hydraulic gradient. Furthermore, a distributed water system over an adverse slope α_b steep enough for freeze-on to occur is likely to contribute to plume formation over a broader area¹.

- L. 142: similar comment to L. 110-111; what about the effect of increased basal sliding?

Yes the effect is here also important. We changed the sentence on L141/142 and added a further sentence to the paragraph. It now reads (new L213):

In fast flow areas, the large ice flux together with increased basal sliding (leading to a vertical distribution of the horizontal velocity resembling the plug flow case) prevent in general high rising plumes (Fig. 5). An exception occurs for very large water fluxes and along-flow series of freeze-on areas, which leads to stacking of the advected plumes (Fig. 5b). Further along, in the ablation area, where surface melt reduces both the meteoric ice flux M and, as a consequence, the total ice flux Q (hence increasing R) and high availability of surface melt-water reaches the bed^{2,10}, larger plumes are again possible.

- L.145-146: This first sentence isnt clear.

We agree that the sentence is not entirely clear. We try to highlight how the different components, we

looked at so far, influence each other and match the observations. The sentence now reads (new L247):

In summary, our arguments regarding the **interplay between** ice flux and PiDMP through the **freeze-on rate \dot{f}** , which is controlled by the basal water flux Q_w , the **freeze-on index Φ** , and the location of the **observed plumes**, support the hypothesis of basal freeze-on.

- Finally, there is a lack of discussion on how the new findings fit / relate to existing work on the formation of these plumes. I would suggest adding details either in the intro and/or conclusion.

We added details by comparing our results to previous work throughout the result section and also added a new section on viscous contrast which produces structures that were discussed in recent literature. We put our results in context with newly published work.

Figures: Overall, I find the figures to be of good quality. They are however showing a lot of information, and the captions are not sufficiently clear to easily understand what is shown. Below are a few suggestions to improve on this.

-Fig. 1: Panel c, it is hard to see the ice core locations. Consider using a different color, or larger symbol? Caption: The radar flight lines can only be seen on panel c. The mention of "grey thin lines" should be moved to that particular section.

We reorganised the original Figure 1 by adding two new figures (one in main text and other in Supplement) giving spatial information of the various existing data and on the modelled water flow paths. We rewrote the captions for the new figures.

-Fig. 2: Caption: consider grouping the information per panels a bit more. For example, most panels are listed 3 times, which doesn't facilitate following what is happening on each panel. In particular, I find the last three lines confusing (after "f and h"): are the blue / red / and orange lines mentioned corresponding to the vertical profiles? (if so, orange line is on panel I only)? Which solid and dashed-dotted lines are being referred to? Also, I would specify more clearly which is the accretion area: maybe use "red horizontal bar" as there are other red lines in the panels.

We regrouped the information for the panels to make it clearer and followed your suggestions.

-Fig. 3: You could make clearer from the beginning, the fact that the 10ka run is a transient run, and the 200ka run is a steady state run (for example, you could state this in your a and b captions). Furthermore, I suggest removing the references to a and b in your c caption, which I find confusing. Instead, I would replace references to a and b in the panel c legend with 10ka / 200 ka, accordingly (so, the legend for the dashed blue line would read F: transient (10 ka), etc) Similarly, I would remove the reference to

a and b in your d caption, and modify so that it reads something like “, shown for both 10 ka (black dashed-dotted line), and 200 ka (grey solid line).”

We took up your suggestion and changed caption and figure accordingly.

Reviewer #2 (Remarks to the Author):

Review of Basal freeze-on at the origin of complex internal ice-layer stratigraphy.

Title: Might consider clearer title: Extent of basal freeze-on in Northern Greenland

This suggested title focuses on the extent of basal freeze-on, however our focus is on the process of basal freeze-on. Therefore we believe the original title is expressing this aspect accurately.

Overview: This paper works to address systematically a very intriguing question — how much of the ice at the base of the Greenland Ice Sheet is refrozen basal water and how does the refreezing work. This is a nice integration of data analysis and modeling. The authors examine the pressure-induced depression of the melting point (PiDMP). Their major conclusion is that the features may be comprised of freeze-on and that the ice flux is the primary control on their height. The major claims of this paper are novel as several groups have been trying to isolate the source of these structures in the ice sheet. This work will be of interest to the ice rheology, ice core and eventually the ice sheet modeling communities. I have noted my concerns with the paper framing and the figures.

I have concerns about Figure 1 have some suggestions on how to improve it. They should address why their hydrologic pathways differ from other published pathways. Others have indicated these features are along pathways — it curious that theirs differ.

“Thus, our spatial analysis supports the hypothesis of basal freeze-on due to ascending water for Greenland.”

In the original paper we calculated the streamlines along the reversed hydraulic gradient in order to obtain the source region. This is why the hydrological pathway looks different than in other publications. It was not stated clearly in the text. Now we plot the hydrological pathways in following the hydraulic gradient downwards, representing how the water flows from the seed point towards the ocean.

In the summary section the authors note the presence of plumes in the regions of fast flow but do not address the distinct difference between Petermann and NEGIS. In Petermann there are plumes in the onset region while in NEGIS there are not. (line 140).

The difference between Petermann and NEGIS is mainly that for Petermann the boundary between moderate and fast flow is not as sharp as for NEGIS, where the flow speed increases rapidly over a small spatial area. Further, Petermann has one confined zone of ice convergence - where fast flow reaches furthest inland, whereas NEGIS has several converging regions, with ice converging at the inner reaches of fast flow but also along the lateral margins of the fast flow area. Both Petermann and NEGIS show a concentration of plumes along the margin of faster flow, where water pathways converge and water fluxes are expected to be high in regions with increased surface slope. However, the spatial resolution is larger over Petermann and showing more detail, as the radar flight lines are spatially denser over a larger area than over NEGIS. In converging ice flow, existing plumes converge too which leads to complex structures within the ice. The general pattern that can be observed, is that large plumes concentrate in regions of ice flow converging towards fast flow, with prevailing high water fluxes. Large plumes in fast flow regions are quickly advected and can only grow large by being pushed upwards when flowing over regions of local freeze-on. We added the following paragraph on L135 (new L195):

Near the ice divide the observed plumes are usually smaller than $1/10H$, in the range of a few 100 meters. For these relative small plume heights the flux ratio R too is small when assuming flow purely by internal deformation. As the ice flux is low, only low water fluxes are needed to produce the required freeze-on rates (a few millimeters to centimeters). Since the surface slope is nearly flat, Φ is low too ($< 10^{-3}$), commonly leading to bed slopes that are too steep for water to flow uphill, and restricting plume formation. Away from the divide, the first large plumes ($> 1/3H$) start to form in regions where the basal water flux Q_w is increasing, with the total ice flux Q and freeze-on index Φ still being generally small. Further along, towards the margin both Φ and Q_w tend to increase (while Q is still moderate) and allow for larger plume heights. In regions of fast flow (large Q) the available basal water flux and Φ are in general too low to produce freeze-on rates \dot{f} that would sustain the large ice-flux ratio R required for large plumes. These observations fit well with the modelling results by Dow et al.⁴ producing larger freeze-on rates along important water flow paths and towards the margin.

Other Comments

The authors omit mentions one of other primary difference between Antarctica and Greenland the well defined presence of water. In Antarctica there is a clear association with the basal plume and the subglacial water network. In Greenland the ability to map subglacial water is only recently been resolved. Currently, we do not know the distribution of subglacial water in Greenland and its relationship to plumes.

We agree that despite some effort (e.g. Livingston et al 2013; Dow et al, 2018) we do not know the

distribution of subglacial water in Greenland. This is why we assume that we have water everywhere to map possible water paths for the case that there is water. But from ice core drilling we do know that there are regions at the base with water. So we know that there is water but we do not know where.

Figures: Figure 1 is terribly complicated and difficult to follow. I could recommend several revisions. (a) to many lines in 1c. I suggest moving to two separate panels first showing plumes and heat flow anomalies (maybe) on flightlines and ice surface then another panel with plumes on top of hydrology, velocity arrows and contours, (b) to many circles — using x as symbol for the plumes would link to labeling on radar cross sections better and simplify somewhat (c) not clear what the red circles are referring to neither Fahnestock et al (#35) nor Rogozhina et al (#34) seem to have this pattern??? (d) the important new observation to present is the freeze-on index that is almost lost in these plots if should be a bolder color or on top or a more transparent hydrology. Please label with lat, lon somewhere. I know this is not very trendy but at some point someone will try and figure out where these points are and northing and easting just will not be appropriate. Is there an offset between the blue box and the location of the modeled structures ??? I am concerned about the drainage pathways presented. These are very different than the drainage pathways for Greenland published by Livingston et al 2013 Cryosphere or Bell et al 2014 in Nature Geoscience

We agree that the original Figure 1 is very complex and tried to break it down in various figures. We have two separate figures showing panels for the input data (radar flight paths, ice velocity together with surface topography, bed topography, and surface slope; now Fig. 1) and the calculated hydrological pathways (now Supplementary Fig. 2). All are shown together with the position of the mapped plumes, with now changed symbols. For the larger plumes we use filled triangles and filled dots for plume structures smaller than $< 1/3H$. These two overview figures also show the lon/lat grid.

The figure with the freeze-on index shows Φ in bolder colors using a coarser grid (1050 m instead of 600 m) than previously used on top of the flight lines, the plumes and the hydrological paths. We used the same symbols for plumes as described above and they are also used on radar profiles (Supplementary Figure 2). The overview figures with the input data and the hydrological pathways show the plume data together with a lat, lon grid. Areas corresponding to regions of higher melt rates as derived from the Fahnestock et al (2001) paper have been omitted. These areas obtained from the papers by Fahnestock et al (2001) or Rogozhina et al (2016) can now be directly compared with the plume distribution using the lat, lon grid in the new Figure 1. The drainage pathway is calculated by following the seed down-gradient instead of up-gradient as explained in the previous paragraph.

Figure 2 is lovely and captures the essence of this work demonstrating the potential for the process to

modify the age-depth relationship in the ice sheet and to deform the stratigraphy.

Thank you!

Methods

Supplemental Figures

Linkage to supplementary figures weak and non-sequential. Supplementary Figure 2 is first to be mentioned. Sup. Figure 1: similar issues to Figure 1. Magenta arrow confuses me To make the case that there is a relationship between plume and freeze-on index how about a plot of plume height versus freeze-on index to compliment the map???

We reorganised the figures by moving some from the Supplementary section to the main section and adjusted the sequential order of figures.

A plot of plume height versus freeze-on index is a nice idea. But it would not convey the relationship between freeze-on index and plume height as the height depends on various factors. Ice flux, the flux of basal water, the slope ratio, the spatial area as well as the temporal extent of freeze-on influence the height of the plume. In areas of low ice flux large plumes can be obtained with relative little freeze-on and therefore not requiring a large freeze-on index. However, towards the margin, with increasing ice flux, large freeze-on is required and therefore rather a larger freeze-on index. Only for the same condition a larger freeze-on index would lead to a higher plume. As the freeze-on index is obtained using information derived from bed and surface topography (slope, hydropotential) such a plot would not clearly highlight the relationship of freeze-on index and plume height.

Sup. Figure 2: where is this focused region?? Again is it the magenta arrow on previous figure if so why are there a different number of flight lines and circles. For completeness I would like to see the two radar profiles upstream of 2a where there is supposed to be a plume and one without.

The focused region shown in the new Supplementary Figure 1 is now highlighted in the new Fig 2 and Supplementary Fig. 2d. The plume shown in the original Fig.2 (now Figure 3) is highlighted in magenta in new Fig 2 and is the same as shown in Supplementary Figure 1h.

Reviewer #3 (Remarks to the Author):

Review:

Leysinger Vieli et al., "Basal freeze-on at the origin of complex internal ice-sheet layer stratigraphy"

The paper presents results from two ice-flow models, demonstrating that with sufficient material added to the base of an ice sheet it is possible to form structures similar to those observed in radar data from the Greenland Ice Sheet. The authors further argue that this process may influence mass balance estimates and flow properties of the ice sheet.

The results are interesting, in particular the use of a 3D model allowing for different viewing geometries, and the model run with a freely evolving surface. Having said that, I have several reservations outlined below regarding the interpretation and conclusions of the paper. I would like to preface this by saying that I agree that the modelled plumes look very similar to the observed plumes but other studies (Wolowick et al. 2014, Bons et al. 2016) have also obtained similar plume structures invoking different processes. Therefore, similarity with observations is not sufficient to imply causation.

Correlation between freeze-on index and plumes

The correlation in Fig. 1 (and supplementary Fig. 1) between the areas where streamlines cross a high “freeze-on index” and the location of a “plume” is not convincing. There are numerous areas where the streamlines cross a high freeze-on index but no plume is observed in the radar data (e.g. east of Petermann glacier) or where plumes are observed but the freeze-on index is low (the plumes west of the magenta arrow in Fig.1). Other studies (Panton and Karlsson, 2015) have mapped significantly more plumes and if these locations were included, the correlation might be clearer. At least, all plumes mapped for this study should be included in Fig. 1, at the moment it is not clear why plumes smaller than $H/3$ are not shown. Are they expected to result from a different process?

Recommendation: Show that there is a correlation between freeze-on index (or rate) and plumes, and include all plumes on the map.

As stated above in the response to Reviewer #2 it is difficult to show that the plumes correlate with freeze-on index as there are various factors acting together (e.g. ice flux, water flux, area, time span of freeze-on....). And then we only have information where we have a radar transect. However, what we can do is to check upstream along ice flow for each plume if areas with a freeze-on index are crossed (see also answer to Reviewer #1). We find that if we seed within 1.5 km from the observed large plumes about 97% have a freeze-on area 5 km upstream, when calculated on a grid of 150 m. While for the small plumes 93% cover such an area. Doing the same with a random distribution of locations the fit is below 80%. Calculating the same on the larger grid of 1050 m as used in the paper figures we obtain similar results in the sense that the larger plumes give a higher match with freeze-on areas than for smaller plumes. This is most likely due to the fact that it is easier to distinct for large plumes if they rise from the bed or if they are advected. The plumes $< 1/3H$ are also mapped when the RES layer structure

of the meteoric ice shows the typical upward bending seen from the modelling experiment when active basal freeze-on is happening. We added the following table to the Supplementary material:

Number of plumes	159	402	200
Radius and upstream distance	$\geq 1/3H$	$< 1/3H$	random
1.5 km and 5 km	72%	63%	45%
1.5 km and 10 km	86%	74%	54%
3 km and 5 km	86%	84%	68%
3 km and 10 km	96%	90%	70%

and added following paragraph to the methods section (section 5.5):

5.5 Estimating correlation between freeze-on index and plumes. To quantify the relation between plumes and freeze-on areas, we follow streamlines along the reversed (surface) ice-flow gradient starting at each plume, and evaluate if they cross areas of freeze-on index Φ . Further we compare the results between the mapped plume sets ($\geq 1/3H$ and $< 1/3H$) with results for randomly chosen locations. For the reversed streamlines calculated on a 1050 m, as used in this paper, we find the best match between plumes and Φ areas for the large plumes, followed by the small plumes (result reduced by $\approx 10\%$) and the random set, a result by $\approx 30 - 20\%$ lower, respectively (Supplementary Table 1). Calculating reverse streamlines on a 150 m grid¹³ along a 5 km stretch, for seeds within 1.5 km radius from the plume, about 97% and 93% of the large and small plumes, respectively, and less than 80% of the random locations are reached by ice flowing over areas of Φ . The result depends on the grid, the seed area and the streamline distance. Reverse streamlines calculated on the 1050 m grid, seeded over an area of 3 km along 10 km, results in comparable values as for the fine grid (1.5 km and 5 km). Larger RES mapping errors for small plumes (see Methods section 1) partly explains the better result for plumes $> 1/3H$.

We include all plumes on the map, differentiating clearer between large plumes and small plumes by choosing different symbols and sizes. Small plumes are not expected to behave differently, but as mentioned above the mapping is not as obvious as for larger ones. Our reason to leave out the small plumes was that we only had them consistently for 2010 – 2012. We now updated the mapping for all plumes for the years 2013 – 2014.

Freeze-on rates and subglacial water discharge

The freeze-on rates necessary to create the plumes require a significant amount of subglacial meltwater. The authors justify the high melt rates by considering different heat fluxes and sources in the basal environment, and then calculate the subglacial water routes from the hydropotential. This seems like

a simplified approach considering that several hydrological models have been developed for ice-flow models including Elmer/Ice (de Fleurian et al., 2014). Even so, the simple approach does provide an estimate of freeze-on rates.

We agree that our calculation of the subglacial water pathways, assuming that we have water everywhere, is a simplified approach. Using or even developing a hydrological model is beyond the scope of this paper. Especially as there is no subglacial data to validate the models. Therefore we prefer to know that our approach is simplified but gives us some indication where water would likely flow. A recent study (Dow et al, 2018) applied the subglacial hydrology model Glacier Drainage System (GlaDS) to assess the locations and rates of freeze-on by supercooling (freeze-on due to a pressure-induced depression of the melting-point ($PiDMP$)) for both distributed and efficient drainage networks. They found that freeze-on occurred in many areas of the ice-sheet but obtained very low averaged freeze-on rates. However, locally freeze-on rates could reach substantial values (centimeters to meters) with the highest found in channels and the regions of higher freeze-on rates broadly matching the spatial distribution of our mapped plume-like features.

We added to the new paragraph starting on L135 (see comments to Reviewer #2) the following sentence:

These observations fit well with the modelling results by Dow et al.⁴ producing larger freeze-on rates along important water flow paths and towards the margin.

It is not clear why this estimate of freeze-on rates is not shown in Fig. 1 instead of the freeze-on index. Admittedly, the estimate is uncertain but by prescribing a high melt rate for the whole domain, the authors would get an upper boundary for the freeze-on rates. For the reader the freeze-on index is confusing and not easily translatable into freeze-on rate.

We chose by purpose to show the freeze-on index, as a simple multiplication with the water flux, ice density, gravitational acceleration and a division of the whole product by the volumetric latent heat of freezing (see Equation (19) and (20)) leads to the freeze-on rate. By assuming a constant basal melt rate we would not expect the water flux to be constant but rather expect to vary locally. Therefore the freeze-on index is of more general value and can be used with whatever water flux is appropriate for the region of interest.

The catchment area for the plume described and modelled in Fig. 2 is outlined in Fig. 1 as a large area close to the ice divide. In Fig. 1, however, only a few of the streamlines that initiate within the catchment area terminates or passes the plume. This suggests (to the reader) that the catchment area for the plume is in fact substantially smaller and by extension so is the amount of available water and

thereby the freeze-on rates.

We adjusted the outlined catchment area above the plume by restricting it only to the streamlines that feed the plume. It was not clearly stated in the text that the streamlines were calculated to show where the water came from. The streamlines plotted together with our previous outline were calculated by streaming from the plume upwards along the hydrological gradient. Now we seed over the accretion length and 15 km upstream along ice flow using a 1050 m grid and calculating streamlines using steps of 5 km. The path of the streamlines is shown in the new overview Figure (Supplementary Fig. 2c). The previous figure used seeds at 30 km spacing with 3 km steps. As our method is not filling up sinks we need to step over them. The larger step size helps to cross regions where water is ponding and therefore reaches further upstream than a smaller step size. The outlined catchment area with a step of 5 km equals 13422 km² and for a step of 4 km 12471 km². Assuming an area of 10000 km² we now have a relevant basal melt rate of $10000 \times (0.01/1000) = 0.1 \text{ km}^3 \text{ a}^{-1}$ or $3.2 \text{ m}^3 \text{ s}^{-1}$ instead of the previously calculated $4.8 \text{ m}^3 \text{ s}^{-1}$. Leading to a maximum channel width of 30 m and 60 m for a melt rate of 0.02 m/a.

We added following in the methods section 5.2 (new L494):

To obtain the source area we seed at 1050 m along the 6 km accretion length (observed in RES profile) and 15 km plume upstream along ice flow, and calculate streamlines along the reversed (upwards) hydraulic gradient using steps of 5 km (Supplementary Fig. 2d).

Finally, studies have indicated that at least in the western part of North Greenland, the bed is likely to be frozen (MacGregor et al., 2016) thus it seems unlikely that large amounts of meltwater is passing through the subglacial system in this region. In contrast, in the eastern part the bed is likely thawed. This poses a problem for the freeze-on hypothesis since the plumes to the east of the ice divide would then be a result of a different process than the plumes west of the ice divide.

We find that when using the outlines of MacGregor et al (2016) of 'likely thawed', 'uncertain' and 'likely frozen' bed only a minority of plumes are found in a 'likely frozen' area (Supplementary Fig. 2b,c), where most of them are small plumes ($< 1/3H$). Further, the plumes in the 'likely frozen' area are localised in the region where Rogozhina et al (2016) modelled elevated geothermal heat fluxes. We added a new overview figure (Supplementary Fig. 2b) showing streamlines seeded in the areas 'likely thawed' and 'uncertain', which feed some of the plumes in the 'likely frozen' area with water. Streamlines seeded only in the 'likely thawed' area (Supplementary Fig. 2c) do leave a substantial amount of plumes in the Petermann and NEGIS region without water.

We added following in the methods section 5.1 (new L487):

Comparing our mapped plumes with a composition of areas with a temperate bed⁹ we find that most plumes are within areas of a 'likely thawed' or 'uncertain' bed, where water paths seeded from both areas reach most plumes (Supplementary Fig. 2b,c).

Recommendation: Show (an upper bound for) freeze-on rates on the map, show that the outlined area is a catchment area for the plume in Fig. 2, and address the problem regarding the different basal thermal states of the ice sheet.

We addressed these points in our responses above.

Ice-flow models

The authors mention that the rheology of the plumes is different compared to meteoric ice. Unfortunately, this is not investigated further. The Elmer/Ice model does have the capability to resolve ice with different rheological properties and it would be extremely interesting to see the impact on ice-flow when a plume builds up with a different viscosity. Would the rheology contrast in itself be enough to sustain the plume (as claimed in other studies (Bons et al., 2016))? By including different ice rheologies the authors would be able to convincingly refute (or demonstrate) that rheology contrasts are important for the plume formation.

Recommendation: Simulate plume formation and ice-flow invoking different ice rheologies.

The suggestion using the Elmer/Ice model to calculate freeze-on with different ice rheologies goes further than the scope of this paper, as this would open up a whole new detailed investigation. Nevertheless, we defined the accreted ice with a contrasting viscosity to the meteoric ice. We performed two suites of experiments where we create a twofold and tenfold viscosity contrast between the meteoric and the accreted ice. The accreted ice is changed to being two or ten times stiffer (factor of 1/2 or 1/10) than the surrounding (meteoric) ice and changed to being two and ten times softer (factor of 2 or 10) than the meteoric ice. We find, as predicted by the kinematic theory, that over the accretion area the plume with stiffer freeze-on ice rises higher than the plume without viscosity contrast (since the ice can't shear easily it grows higher) and that the plume with softer ice rises less high (as it can shear easily it will move downstream). However, at a distance downstream of the accretion area both plumes with a viscosity contrast rise higher than for the plume without a contrast. The result of the softer ice is not entirely obvious - one way to explain it, is that while the soft ice shears easily it will be blocked by the stiffer ice downstream of the accretion area and has to rise over it. However, as it is a complex fluid dynamical problem it may therefore not have a simple explanation. What this experiment shows is that using a vis-

cosity contrast leads to general higher plumes, therefore requiring smaller freeze-on rates (and therefore smaller water fluxes) to produce the observed plume-like structures. Further the results of the experiment suggests that from the plume shape we can gain information on the viscosity contrast.

We added a figure (Fig. 8) and following paragraph to the main text continuing on L144 (new L219):

Contrast in ice rheology

Plume-ice rheology is expected to differ from meteoric ice rheology², reflecting different ice temperature and fabric, and consequently affecting the relationship between ice-flux ratio R and relative height h (equation (12)) for cases with internal deformation rather than plug flow. Large differences in rheological properties in near basal ice has been observed in Greenland³. A recent study by Wrona et al.¹⁸ describing the geometry and morphology of near basal anomalous layer structures in RES profiles over the Gamburtsev mountains, East Antarctica, concludes that mechanical mixing between meteoric and accreted ice is triggered by rheology contrasts producing complex structures.

To investigate the effect of a rheology contrast between the meteoric and accreted ice on plume growth, we perform a simple experiment, using the FS model. We use the same experiment set up as described earlier, with the exception of a modified accreted-ice viscosity. By changing the rate factor (A) for ice flow^{5,17} (see Methods section 3) by factors of 1/10, 1/2, 2 and 10 in four separate experiments we increase or reduce the stiffness of the accreted ice respectively, in order to compare it with the original case, which uses an uniform viscosity throughout the ice sheet. For the same amount of accreted ice with increased stiffness (factors 1/10 and 1/2) we find that plumes rise as expected higher than in the original case (Fig. 8; red dashed line), since stiffer ice cannot shear so easily. A reduction in accreted-ice stiffness leads however to a complex and not entirely obvious result. Over the accretion area the softer accreted-ice rises not as high as in the original case (Fig. 8a,d), which might be explained by the fact that it shears more easily. However, downstream of the accretion area the softer ice eventually rises higher than the original plume, which is simply advected downstream (Fig. 8b,e). A possible explanation could be that downstream of the accretion area the soft ice has to override the stiffer meteoric ice. Accretion of softer ice over a long time scale leads to instabilities along the advected plume margin (Fig. 8c,f) resembling the ‘fingering’ structures described by Wrona et al.¹⁸. Since a fluid infiltrating a higher viscosity fluid is a complex fluid dynamical problem, it might not have a simple explanation. Nevertheless, we can state that the plume height highly depends on the viscosity contrast, requiring smaller freeze-on rates to achieve the same plume height than stated above. We find that, whatever the rheological contrast (within reason), plume structures can be formed by the accretion process.

Minor comments

Fig. 1 is difficult to interpret. The contrast between orange, magenta and red (and blue) is not big enough. Overall, the figure is too busy. A legend would also help the reader.

We tried to improve this figure by splitting it into several figures and keeping the originally Fig. 1c simpler with stronger colors using a coarser grid (1050 m instead of 600 m). See comments to Reviewer #1.

In Fig. 3d, it is not clear which line belongs to which axis (dotted line is h ?).

The axis are valid for both lines - there is a different grid - so one can read both values along one line.

Added axis information:

d, Ice-flux ratio R (right y-axis) and relative plume height h (left y-axis) for ice flow by internal ice deformation only, shown for both 10 ka (black dashed-dotted line) and 200 ka (grey solid line) of freeze-on.

Line 23: assumed to be isochrones rather than “termed isochrones”

Changed this in text.

Line 72: Distored = distorted?

We corrected this.

Line 109: Missing a space after full stop.

Added space after full stop.

Lines 121 and 122: The use of the ζ_j symbol is not very helpful to the reader and could be phrased differently.

We changed the sentence starting on L120 with (new L169):

The dependence of PiDMP on S has the consequence that water can flow up steeper adverse bed slopes $\alpha_b > 0$ for a steeper inclined surface slope $\alpha_s < 0$ (or vice versa), leading to a larger PiDMP effect owing to larger spatial gradients in overburden and thus melting point.

Line 160: Presumably, “oldest ice” refers to the 1.5 million year old ice-core project but this should be stated more explicitly.

We added to L159/160 with (new L266):

Furthermore, plume growth alters the age-depth relationship, as bottom layers are lifted and compressed, while protected from basal melt, with important implications for ice-core drilling, interpretations, and the search for a site to core at least 1.5-million-year-old ice¹⁴.

References Bons, P. D., D. Jansen, F. Mundel, C. C. Bauer, T. Binder, O. Eisen, M. W. Jessell, M.-G. Llorens, F. Steinbach, D. Steinhage, et al. (2016), *Converging flow and anisotropy cause large-scale folding in Greenlands ice sheet*, *Nature communications*, 7, 11,427. doi:10.1038/ncomms11427

Wolovick, M. J., T. T. Creyts, W. R. Buck, and R. E. Bell (2014), *Traveling slippery patches produce thickness-scale folds in ice sheets*, *Geophys. Res. Lett.*, 41, 88958901, doi:10.1002/2014GL062248.

de Fleurian, B., Gagliardini, O., Zwinger, T., Durand, G., Le Meur, E., Mair, D., and Rback, P. (2014): *A double continuum hydrological model for glacier applications*, *The Cryosphere*, 8, 137-153, <https://doi.org/10.5194/tc-8-137-2014>.

Panton, C. and Karlsson, N. B. (2015), *Automated mapping of near bed radio-echo layer disruptions in the Greenland Ice Sheet*, *In Earth and Planetary Science Letters*, Volume 432. <https://doi.org/10.1016/j.epsl.2015.10.024>.

MacGregor, J. A., et al. (2016), *A synthesis of the basal thermal state of the Greenland Ice Sheet*, *J. Geophys. Res. Earth Surf.*, 121, 13281350, doi:10.1002/2015JF003803.

References

- [1] Alley, R. B., Lawson, D. E., Evenson, E. B., Strasser, J. C., and Larson, G. J. (1998). Glaciohydraulic supercooling: a freeze-on mechanism to create stratified, debris-rich basal ice: II. Theory. *Journal of Glaciology*, 44(148):563–569.
- [2] Bell, R. E., Tinto, K., Das, I., Wolovick, M., Chu, W., Creyts, T. T., Frearson, N., Abdi, A., and Paden, J. D. (2014). Deformation, warming and softening of Greenland’s ice by refreezing meltwater. *Nature Geoscience*, 7:497–502. doi:10.1038/ngeo2179.
- [3] Dahl-Jensen, D. and NEEM community members (2013). Eemian interglacial reconstructed from a Greenland folded ice core. *Nature*, 493:489–494. doi:10.1038/nature11789.
- [4] Dow, C. F., Karlsson, N. B., and Werder, M. A. (2018). Limited impact of subglacial supercooling freeze-on for Greenland Ice Sheet stratigraphy. *Geophysical Research Letters*, 45. doi:10.1002/2017GL076251.
- [5] Glen, J. W. (1955). The creep of polycrystalline ice. *Proceedings of the Royal Society of London, Ser. A*, 228(1175):519–538.

- [6] Gudmundsson, G. H. (2003). Transmission of basal variability to a glacier surface. *Journal of Geophysical Research*, 108(B5).
- [7] Leuschen, C., Gogineni, P., Hale, R., Paden, J., Rodriguez, F., Panzer, B., and Gomez, D. (2014, updated 2016). IceBridge MCoRDS L1B geolocated radar echo strength profiles, version 2. Boulder, Colorado USA: National Snow and Ice Data Center. <http://dx.doi.org/10.5067/90S1XZRBAX5N>.
- [8] Livingstone, S., Clark, C., Woodward, J., and Kingslake, J. (2013). Potential subglacial lake locations and meltwater drainage pathways beneath the Antarctic and Greenland ice sheets. *The Cryosphere*, 7:1721–1740. doi:10.5194/tc-7-1721-2013.
- [9] MacGregor, J. A., Fahnestock, M. A., Catania, G. A., Aschwanden, A., Clow, G. D., Colgan, W. T., Gogineni, S. P., Morlighem, M., Nowicki, S. M., Paden, J. D., Price, S. F., and Seroussi, H. (2016). A synthesis of the basal thermal state of the Greenland Ice Sheet. *JGR*, 121(F003803):1328–1350.
- [10] Machguth, H., MacFerrin, M., van As, D., Box, J. E., Charalampidis, C., Colgan, W., Fausto, R. S., Meijer, H. A. J., Mosley-Thomposn, E., and van de Wal, R. S. W. (2016). Greenland meltwater storage in firn limited by near-surface ice formation. *Nature Climate Change*. doi.10.1038/NCLIMATE2899.
- [11] Morlighem, M., Rignot, E., Mouginot, J., Seroussi, H., and Larour, E. (2014). Deeply incised submarine glacial valleys beneath the Greenland Ice Sheet. *Nature Geoscience*, 7:418–422.
- [12] Morlighem, M., Williams, C., Rignot, E., An, L., Arndt, J. E., Bamber, J., Catania, G., Chauch, N., Dowdeswell, J. A., Dorschel, B., Fenty, I., Hogan, K., Howat, I., Hubbard, A., Jakobsson, M., Jordan, T. M., Kjeldsen, K. K., Millan, R., Mayer, L., Mouginot, J., Nol, B., O’Cofaigh, C., Palmer, S. J., Rysgaard, S., Seroussi, H., Siegert, M. J., Slabon, P., Straneo, F., van den Broeke, M. R., Weinrebe, W., Wood, M., and Zinglensen, K. (2017a). BedMachine v3: Complete bed topography and ocean bathymetry mapping of Greenland from multibeam echo sounding combined with mass conservation. *Geophysical Research Letters*, 44. doi.org/10.1002/2017GL074954.
- [13] Morlighem, M., Williams, C., Rignot, E., An, L., Arndt, J. E., Bamber, J., Catania, G., Chauch, N., Dowdeswell, J. A., Dorschel, B., Fenty, I., Hogan, K., Howat, I., Hubbard, A., Jakobsson, M., Jordan, T. M., Kjeldsen, K. K., Millan, R., Mayer, L., Mouginot, J., Nol, B., O’Cofaigh, C., Palmer, S. J., Rysgaard, S., Seroussi, H., Siegert, M. J., Slabon, P., Straneo, F., van den Broeke, M. R., Weinrebe, W., Wood, M., and Zinglensen, K. (2017b). IceBridge BedMachine Greenland, version 3. Boulder, Colorado USA: NASA DAAC at the National Snow and Ice Data Center.

- [14] Parrenin, F., Cavitte, M. G. P., Blankenship, D. D., Chappellaz, J., Fischer, H., Gagliardini, O., Masson-Delmotte, V., Passalacqua, O., Ritz, C., Roberts, J., Siegert, M. J., and Young, D. A. (2017). Is there 1.5-million-year-old ice near dome c, antarctica? *The Cryosphere*, 11:2427–2347. doi.org/10.5194/tc-11-2427-2017.
- [15] Röthlisberger, H. and Lang, H. (1987). Glacial Hydrology. In *A.M. Gurnell and M.J. Clark (Ed.), Glacio-Fluvial Sediment Transfer - An Alpine Perspective*, pages 207–284. John Wiley and Sons, Chichester, New York, Toronto, Singapore.
- [16] Shreve, R. L. (1972). Movement of water in glaciers. *Journal of Glaciology*, 11(62):205–214.
- [17] Steineman, S. (1958). Experimentelle Untersuchungen zur Plastizität von Eis. Geotechnische Serie Nr. 10, Beiträge zur Geologie der Schweiz., Kommissionsverlag Kümmerli & Frey AG, Geographischer Verlag, Bern.
- [18] Wrona, T., Wolovick, M. J., Ferraccioli, F., Corr, H., Jordan, T., and Siegert, M. J. (2017). Position and variability of complex structures in the central East Antarctic Ice Sheet. In M.J. Siegert, S.S.R. Jamieson, D. W., editor, *Exploration of subsurface Antarctica: uncovering past changes and modern processes*, number 461 in Special Publication, pages 113–129. Geological Society of London. doi:10.1144/SP461.12.

Reviewers' comments:

Reviewer #1 (Remarks to the Author):

The authors have made thorough revisions to the manuscript, which I now find much clearer. Equally, the revisions to the figures are very helpful. Overall, this is a very interesting manuscript, which will no doubt be of interest to the glaciology community.

I have just a few additional comments, which could be addressed by minor adjustment to the text.

- The first comment concerns the relationship between the plume location / elevated PHI / water paths:

In their response, the authors explain that it is difficult to show the correlation between plume location and high PHI / water path, due to various uncertainties in the available datasets. Thus, I would suggest that this is reflected in the text by changing the word "reveals" on line 57 for something less assertive (e.g. "suggests"?). Moreover the new table (supp. Material) and section 5.5 are nice and help supporting the correlation, so I would refer to these along Fig 2e, Supp Fig 2 and section 5.1 (I see that they are cited later on (line 208), but this seems to come a bit late).

- line 108: would suggest using "a uniform amount of basal freeze-on"

- line 109: the values of freeze rate (0.42 m/yr) and accumulation rate (0.14m/yr) seem very specific. Can you explain better what drove this parameter choice?

- Line 169: Define "Adverse slope" when it is first used, i.e., line 169 (currently defined on line 183).

Reviewer #2 (Remarks to the Author):

Review of Revised Manuscript:

Basal freeze-on at the origin of complex internal ice-layer stratigraphy.

In general the paper has been significantly improved in revision. The expanded text makes the work clearer and the figures are much better and easier to understand. My concerns about Figure 1 in particular have been addressed. The authors response to my questions about the difference in the plume distribution between Petermann and NEGIS catchments in northern Greenland clarifies the issue. Their statement that we assume that water is plentiful at the base of the Greenland ice sheet is adequate.

The linkage between the main text and the supplementary figures is improved and the supplementary figures are improved.

I would still suggest a clearer title such as:

Contribution of basal freeze-on to complex ice sheet stratigraphy

My argument would be that the "at the origin" is not the critical contribution of this paper.

The authors have responded to the reviews comments adequately and I would suggest publishing the manuscript.

Reviewer #3 (Remarks to the Author):

This is a very nice modelling study showing simulations of internal stratigraphy development under different conditions. However, I have some reservations about the conclusions of the paper. While I agree that the model results are similar to the observed layer stratigraphy I do not think that is a strong enough argument in itself. Especially since several other studies have successfully reproduced the layer stratigraphy invoking other processes. Thus, while the authors demonstrate that it is possible to grow the plume-like features with basal freeze-on, I am not convinced that it is the most probable process. Below, I outline my main concerns.

#1 Spatial correlation between plumes, freeze-on index and subglacial water paths

Lines 51-53: "Analysing the spatial distribution of internal ice-layer structures with regard to Φ and water flow-path, reveals that large plumes are located along subglacial water paths crossing areas of elevated Φ (Fig. 2 and Supplementary Fig. 2; see Methods section 5.1)."

Lines 191-192: "A detailed spatial analysis for Greenland locates large plumes along possible water-paths crossing areas of high freeze-on indices..."

Suppl. Table 1.

It is not obvious to me that there is a spatial correlation. From Fig. 2, areas of elevated freeze-on index occur in numerous areas. In some areas, they coincide with subglacial water paths and plumes, and in other areas they do not (see also attached figure). The attempt to correlate plume locations and freeze-on index in Suppl. Table 1 does not shed light on this. Considering that the freeze-on index is high in so many parts of the ice sheet, I would expect most streamlines to cross such an area at some point.

Lines 147-149: "In addition, basal sliding becomes more important downstream, affecting the distribution of horizontal velocity with depth, and thus further decreases the plume height (Fig. 6)."

Lines 213-216: "In fast flow areas, the large ice flux together with increased basal sliding (leading to a vertical distribution of the horizontal velocity resembling the plug flow case) prevent in general high rising plumes (Fig. 5). An exception occurs for very large water fluxes and along-flow series of freeze-on areas, which leads to stacking of the advected plumes (Fig. 5b)."

Observations from Petermann basin (e.g. Bell et al., 2014 and Pantou and Karlsson, 2015) indicate that the plumes are increasing in size downstream. In Fig. 2, there is a large area with elevated freeze-on index but the plumes are downstream of this area. This seems to be in direct contrast with the modelling results (e.g. Fig. 5b where the plumes decrease in size downstream). Are the plumes observed in Petermann a result of "very large water fluxes"? And in this case, how large? Can the model reproduce the observations from Petermann?

Lines 199-202: "Away from the divide, the first large plumes ($> 1/3H$) start to form in regions where the basal water flux Q_w is increasing, with the total ice flux Q and freeze-on index Φ still being generally small. Further along, towards the margin both Φ and Q_w tend to increase (while Q is still moderate) and allow for larger plume heights."

Based on the suppl. Fig. 2, I would argue that it is just as likely that the plumes form in areas where there is a transition from frozen to thawed conditions.

#2 Amount of basal meltwater and freeze-on rates

Line 154: Reference no. 27 and 28.

Both references here are modelling studies and do not provide direct evidence of subglacial water or major pathways.

Line 167-168: "It is readily conceivable that locally conditions arise, which lead to freeze-on rates comparable to the ones used in our experiments."

Is it? This statement comes across as unsubstantiated. Is the slope ratio reasonable for the areas where the plumes are observed? Are the values for the water flux reasonable? This needs to be stated explicitly. Also, there is a difference between conditions such as these arising locally, and to them being present on a large scale across the entire ice sheet as evidenced by the extensive presence of the plumes.

Lines 251-252: "...and broadly coincide with modelled freeze-on regions [28]"

This statement is a bit misleading since the freeze-on rates in the study cited (Dow et al.) are at least an order of magnitude smaller than what is invoked here.

Lines 262-263: "Therefore, statements about the mismatch between plume size and freeze-on rates [28] are not valid without considering as well the effect on the relationship between plume height and freeze-on rate both from the flux relationship and rheology contrast."

I agree that looking at the freeze-on rate in itself is not adequate when examining plume heights. However, from the figures it seems that 10*softer ice roughly leads to a doubling in plume height. Since the freeze-on rates are one to two orders of magnitude larger than the rates from Dow et al., using the freeze-on rates from the latter study would never lead to a plume of that height, regardless of whether or not ice-flux and rheology contrasts were included. It should be explicitly stated somewhere that the freeze-on rates from Dow et al., would only lead to X m (or x relative height) plumes. Otherwise the reader might think that the addition of ice-flux and rheology contrast in itself is enough.

#3 Frozen basal conditions

Lines 194-197: "Near the ice divide the observed plumes are usually smaller than $1/10H$, in the range of a few 100 meters. For these relative small plume heights the flux ratio R too is small when assuming flow purely by internal deformation. As the ice flux is low, only low water fluxes are needed to produce the required freeze-on rates (a few millimeters to centimeters)."

Near the ice divide, radar observations indicate that the bed is likely frozen (as shown in Suppl. Fig. 2), thus the subglacial water flux must be zero and therefore also the freeze-on rates. It is a significant weakness in this manuscript that the presence of the plumes in the interior of the ice sheet cannot be explained by the freeze-on process but this weakness is not mentioned or discussed.

Response to the Reviewer's comments

Notes to the following response: 'Lxx' refers to the Lines of the originally submitted manuscript (otherwise noted as 'new Lxx', which refers to the manuscript with mark up), *slanted black text is the reviewers comments*, *our comments are in red*, original text in black, *and in blue is the changed or added text as found in the revised manuscript*.

We made a few general changes in this third revised version:

- (i) We added to the introductory section a final paragraph briefly summarising both the results and conclusions, as required by *Nature Communications*. (new L63-70)

Here, we examine the hypothesis of basal freeze-on and present the consequent isochrone layer architecture obtained using mathematical models of various complexities and spatial dimensions; the results closely resemble the layer architecture observed in RES profiles. Observed plume-like features map well with locations that are favourable in topography for the PiDMP process and match areas where water flow is likely, additionally corroborating the hypothesis. We further establish that low ice-flux and contrasts in ice rheology are conducive to larger plumes. Our findings highlight important implications of the basal freeze-on process for ice dynamics and ice-core drilling.

- (ii) While scrutinising the data we realised that 41 plumes with heights $< H/3$ went missing during the first revision (mainly Petermann and NEGIS region). We updated all the figures showing plumes so as to include the complete data set.

- (iii) For the calculation of Φ we previously used a surface slope threshold set at 10^{-4} to exclude all slopes smaller than this value to avoid calculations with a zero surface. However, we realised that these small slopes are still important in the interior, leading to freeze-on rates in the millimetre per year range for water fluxes as small as $Q_w = 0.01 \text{ m}^3\text{s}^{-1}$ per metre width. We have now only removed areas where the surface is exactly zero and our plot of Φ shows many more areas of small Φ in the central region of the ice sheet. We have updated the relevant figures.

- (iv) In order to clearly visualise the major trend in water paths in Figure 2 we chose seeds on a coarser grid (45 km instead of 12 km) to calculate the streamlines parallel with the hydrostatic gradient vector. We further reduced the content in the figure to make it clearer. The original Figure 2 is now the new Supplementary Figure 2.

- (v) Throughout the text we corrected typos and improved sentences.

Reviewers' comments:

Reviewer #1 (Remarks to the Author):

The authors have made thorough revisions to the manuscript, which I now find much clearer. Equally, the revisions to the figures are very helpful.

Overall, this is a very interesting manuscript, which will no doubt be of interest to the glaciology community.

I have just a few additional comments, which could be addressed by minor adjustment to the text.

- The first comment concerns the relationship between the plume location / elevated PHI / water paths: In their response, the authors explain that it is difficult to show the correlation between plume location and high PHI / water path, due to various uncertainties in the available datasets. Thus, I would suggest that this is reflected in the text by changing the word “reveals” on line 57 for something less assertive (e.g. “suggest”?). Moreover the new table (supp. Material) and section 5.5 are nice and help supporting the correlation, so I would refer to these along Fig 2e, Supp Fig 2 and section 5.1 (I see that they are cited later on (line 208), but this seems to come a bit late).

We changed the sentence (new L78-83) as suggested and further highlight that the general pattern of plume location matches. The sentence reads now:

Analysing the spatial distribution of internal ice-layer structures with regard to Φ and water flow-path, while considering the likelihood of a frozen bed, suggestsreveals that in generallarge plumes are located along the main subglacial water paths crossing areas of elevated Φ (Fig. 2, ~~and~~ Supplementary Figs. 2, 3 and Tables 1, 2; see Methods section 5.1 and 5.5; note, uncertainties are largest in regions of sparse radar data¹⁵ affecting both Φ and the computed water path).

- line 108: would suggest using “a uniform amount of basal freeze-on”

Agreed - changed 'equal' to 'uniform'.

- line 109: the values of freeze rate (0.42 m/yr) and accumulation rate (0.14m/yr) seem very specific. Can you explain better what drove this parameter choice?

We chose the freeze-on rate to be three times larger than the accumulation rate, as to produce large plumes comparable to observed ones that show larger effects in the deflection of the internal layers of the

meteoric ice. It is clearer to highlight the relationship of the freeze-on rate to the surface accumulation instead of the exact values chosen. We replaced this in the text (new L132-135)

In order to investigate the effect of ice flow on the plume shape, we apply an ~~uniform~~ amount of basal freeze-on, using a freeze-on rate ~~three times larger than the surface accumulation~~ $\dot{f} = 0.42 \text{ m a}^{-1}$ (~~surface accumulation~~ $\dot{a} = 0.14 \text{ m a}^{-1}$) along an extent of 5 km, to produce distinctive plumes at three different locations along the flow-line.

- Line 169: Define “Adverse slope” when it is first used, i.e., line 169 (currently defined on line 183).

We leave the definition of ‘adverse slope’ where it is but remove ‘adverse’ in the first sentence. It now reads (new L190-192):

Substantial rates (tens of centimetres per year) have been previously suggested as a result of PiDMP, controlled by the ~~adverse~~ bed slope (α_b), surface slope (α_s), and the water flux (Q_w) ~~when α_b has opposite sign to α_s~~ ¹ (see Methods section 4).

Reviewer #2 (Remarks to the Author):

Review of Revised Manuscript:

Basal freeze-on at the origin of complex internal ice-layer stratigraphy.

In general the paper has been significantly improved in revision. The expanded text makes the work cleared and the figures are much better and easier to understand. My concerns about Figure 1 in particular have been addressed.

The authors response to my questions about the difference in the plume distribution between Petermann and NEGIS catchments in northern Greenland clarifies the issue. Their statement that we assume that water is plentiful at the base of the Greenland ice sheet is adequate.

The linkage between the main text and the supplementary figures is improved and the supplementary figures are improved.

I would still suggest a clearer title such as:

Contribution of basal freeze-on to complex ice sheet stratigraphy My argument would be that the at the origin is not the critical contribution of this paper. The authors have responded to the reviews comments adequately and I would suggest publishing the manuscript.

We see the point of the reviewer and agree that the suggested title is clearer and more accurate than our

previous title. Prior to our study it was not expected that basal freeze-on can produce structures of the observed vertical scale. We change the title to:

Basal freeze-on generates complex ice-sheet stratigraphy.

We adjusted the abstract on new L11-14 too, so that it reads:

Here, we simulate ~~their genesis by the process of~~ basal freeze-on using numerical ice-flow modelling and ~~analyse~~ assess the transient evolution of the emerging ice-plume and the surrounding ice-layer structure as a function of both ~~freeze-on rate and~~ ice flux ~~and freeze-on rates~~.

Reviewer #3 (Remarks to the Author):

This is a very nice modelling study showing simulations of internal stratigraphy development under different conditions. However, I have some reservations about the conclusions of the paper. While I agree that the model results are similar to the observed layer stratigraphy I do not think that is a strong enough argument in itself. Especially since several other studies have successfully reproduced the layer stratigraphy invoking other processes. Thus, while the authors demonstrate that it is possible to grow the plume-like features with basal freeze-on, I am not convinced that it is the most probable process. Below, I outline my main concerns.

We added a paragraph in the introduction referring to the previous work that use numerical modelling to test the hypothesised process in more detail (new L36-46). We agree with the reviewer that layer stratigraphy can look similar even though invoked by different processes, especially when not comparing the structure over the whole ice column. However, we disagree with the reviewer's statement that these previous studies have 'successfully reproduced' the layer stratigraphy. Bons et al. (2016)³ have produced some fold structures (but not reproduced the observed structure) in a 2-D structural model at crystal scale and Wolovick et al. (2014, 2016)^{20,21} produced some overturned folds that resemble, in parts, the observed near-basal fold structure, but not over the entire ice thickness. Furthermore, the RES transect used in Wolovick et al. (2014)²¹ is not oriented along-flow over its entire length. Leysinger Vieli et al. (2007)¹¹ have shown that both the transect flow orientation and the entire vertical profile are of importance, since the layers above are deflected by the process at the base. In our study we compare the whole 3-dimensional layer structure, and observe that the process of basal freeze-on produces a layer structure that resembles the observed structure over the whole ice-sheet depth. The paragraph now reads:

Only two of the suggested processes have been tested by means of numerical modelling^{3,20,21}. However, none of these model results reproduce the observed RES layer architecture over the whole depth

of the ice sheet, concentrating on the dominant near-basal structures. Bons et al. (2016)³ have obtained some fold structure using a 2-D structural model at crystal scale, and Wolovick et al. (2014, 2016)^{20,21} produced overturned folds that resemble in parts the observed near-basal fold structures by applying transient changes in slipperiness. However, as previously shown by Leysinger Vieli et al. (2007)¹¹, the layer structure over the whole ice column is of importance, since a local change in boundary condition or flow mode deflects the isochrones over the column, and leads to patterns that can look distinctively different for transects along or oblique to ice flow. This suggests that in order to evaluate a process by means of layer stratigraphy, both modelled and observed ice stratigraphy need to be compared over the whole ice column along transects of similar angle to ice flow.

#1 Spatial correlation between plumes, freeze-on index and subglacial water paths

Lines 51-53: “Analysing the spatial distribution of internal ice-layer structures with regard to Φ and water flow-path, reveals that large plumes are located along subglacial water paths crossing areas of elevated Φ (Fig. 2 and Supplementary Fig. 2; see Methods section 5.1).” Lines 191-192: “A detailed spatial analysis for Greenland locates large plumes along possible water-paths crossing areas of high freeze-on indices”

Suppl. Table 1.

It is not obvious to me that there is a spatial correlation. From Fig. 2, areas of elevated freeze-on index occur in numerous areas. In some areas, they coincide with subglacial water paths and plumes, and in other areas they do not (see also attached figure). The attempt to correlate plume locations and freeze-on index in Suppl. Table 1 does not shed light on this. Considering that the freeze-on index is high in so many parts of the ice sheet, I would expect most streamlines to cross such an area at some point.

To clarify the issue of spatial correlation; one needs to be aware that Φ only represents the geometrical requirement for freeze-on based on the pressure-induced depression of the melting-point (PiDMP) such as the relationship between the adverse bed, where water can potentially flow uphill, and the surface slope. Hence, it is a purely geometrical quantity. A steeper surface slope allows for a steeper adverse bed slope, which results in a larger freeze-on index Φ . Therefore, the index depends on the bed and surface topography, and it is a property of this index to be found favourable to PiDMP in many places. A further point is that in order to have freeze-on water is needed. For a given water flux Q_w we can calculate the amount of freeze-on \dot{f} for each location by multiplying Φ , which is non-dimensional, with the water flux, the ice density, gravitational acceleration, and finally dividing it by the volumetric latent heat of freezing.

As we do not have direct observations of where and how much water is flowing under the ice sheet, we use Φ in our study to visualise the locations where freeze-on could possibly happen under the right conditions. It is not our aim to use Φ as a predictor for freeze-on plumes.

Next we consider the availability and quantity of water flux, which means we also need to consider the ice flux. In regions of low ice flux, freeze-on rates need not to be big to produce a visible plume while in regions of large ice flux the freeze-on rate needs to be considerable greater to obtain the same plume height.

As water paths have only been modelled and not observed so far, we assume, as an approximation, that water is everywhere and calculate the path it would take. Hence, using this method we are able to visualise the major water transport paths - the regions where we would expect potentially available water to flow. However, in regions where the bed is frozen we do not expect water to flow. To visualise these regions we use the map by MacGregor et al. (2016)¹³ of the likely thermal state of the bed, which can be used to constrain the likelihood of (but not totally exclude) the existence of these water paths in frozen areas. However, we have to be aware that this map too is only a model result, where results from different 3-D thermomechanical models and melt rate estimates obtained from the analysis of dated radiostratigraphy with a 1-D steady state ice-flow model have been compiled.

Furthermore we need to be aware that the accuracy of Φ is related to the uncertainties in the surface and bed dataset. This refers in particular to the bed topography, which shows the largest spatial error as it is interpolated from RES data, where in regions of sparse radar data the vertical error in elevation/thickness exceeds 600 m (see Morlighem et al. (2017) (Supplementary Fig. 3)^{14,15}). In regions of sparse data the error along a single radar transect is between 120 – 200 m and increases away from the transect to 300 – 400 m over a distance between 5 – 10 km. This error does not represent the error in bed slope, which varies much more locally, but gives a general indication where the calculation of Φ is expected to be less accurate. Note that Φ and the water paths are consistent since the same dataset for bed and surface is used, so that with each improved topography dataset the estimates of Φ and water paths become more reliable.

Accounting for the overall error and considering the main water paths, the likely frozen areas (as well as ice flux when looking at the plume distribution) a distinct spatial correlation is visible. Some plumes are visible in areas where the likelihood map suggest a frozen bed but where various studies suggest a thawed bed. Plumes are not observed in regions where there is total agreement for a frozen bed between the studies (e.g. area G3 Response Fig. 1). One could turn the argument around and use the observed plumes as an indication of where water is flowing.

The areas highlighted in green and red circles in the attachment figure of review #3 (see Response Fig. 1)

Response Figure 1: **Labelled attachment to the review of Reviewer #3.** Figure 2 from second revised version marked by Reviewer #3. Labels G1-G5 and R1-R3 have been added by us. The Figure shows calculated freeze-on index Φ (yellow-orange-red) superimposed on the modelled basal-water flow path (blue; seeds on 12 km grid with 3 km step-size along streamline) along the hydrostatic-potential gradient. 2010-2014 radar flight-lines¹⁰ (grey thin lines). Direction and magnitude of ice-surface velocity⁹ on 80 km grid (dark-grey arrows). Flow speed contours at 50, 75 and 100 m a⁻¹ (dark-grey to black). Outlined area (magenta) shown as zoom in Supplementary Fig. 1.

are all located in areas where radar inference by MacGregor et al¹³ suggests negative and positive melt rates, respectively (see Figure 5a¹³). Compared with the eight 3-D thermomechanical models (see Figure 3¹³) almost all models (> 90%) predict a frozen bed at the location of the green circles (except G1 and G5). Whereas for the red circles less, than half (42%) of the 3-D thermomechanical models predict a thawed bed while other sources (e.g. NorthGRIP borehole⁶, radar⁴) observed basal melt in this region.

After we removed the threshold of very low slope from the calculation of Φ , the interior shows many more regions of Φ also within the red circles. This change in coverage leads to a higher match between observed plumes and freeze-on area (see amended Table 1). For the new calculation of the random fit, we selected the points randomly within the area covering the observed plumes and increased the number of points to 10,000.

We undertook the following changes to improve the visualisation of the correlation:

(i) Figure 2 has been amended by:

- using the re-calculated Φ with no artificial threshold as explained above.
- choosing a coarser seed grid (45 km instead of 12 km) using a 5 km step size to visualise the main water path more clearly.
- using the complete data set for plumes (41 small plumes added) as stated above.
- showing the surface-velocity vectors on a 150 km coarse grid.
- plotting the likely frozen area in the background and plotting contour of the balance flux velocity of 4 m a^{-1}

(ii) We moved the previous Figure 2, with all observed plumes and newly calculated Φ , into the supplementary section (new Supplementary Figure 2).

Table 1: **Relative plume mismatch with freeze-on index Φ between observed and random data set.** The data set and results are taken from Table 1 to calculate the difference in percentage for plumes with no freeze-on area relative to the expected value in mismatch obtained from the random set of 10,000 plumes. Note, the approximate error in the random set is $\pm 1\%$.

Compared to random set	Difference in mismatch		Relative difference	
	$\geq 1/3H$	$< 1/3H$	$\geq 1/3H$	$< 1/3H$
1.5 km and 5 km	14%	9%	36%	23%
1.5 km and 10 km	12%	4%	50%	17%
3 km and 5 km	7%	7%	35%	35%
3 km and 10 km	7%	4%	64%	36%

(iii) We updated the table for the changed data set (complete plumes and Φ) and added a table to highlight the difference between the observed plumes and the random plumes with no Φ values

upstream. This highlights that if plumes were randomly distributed with respect to Φ , the expected values in misfit would be larger.

Furthermore, we clarify in the results section that basal freeze-on is a geometrical quantity and make our statement in the sentence beginning on L51 less assertive owing to the uncertainties in the available datasets, and changed the sentence to highlight that the general plume distribution matches the major water paths. The paragraph now reads new L66-78 (new L78-83 shown above for Reviewer #1) now reads:

Although in Greenland the relief of the subglacial topography is less pronounced than in Antarctica, we find that plume-like structures (position of apex in RES profiles; see Methods section 1) are predominantly located on the flanks of the main subglacial basin in regions of moderate ice flow and surface slope (Figs. 1b-d). To verify the relationship with a rising bed we define an index for freeze-on (Φ) as a purely geometrical quantity (see Methods section 4.2 equation (20)) that accounts for the effect of the bed-incline (relative to surface slope) on freeze-on triggered by PiDMP. Analysing the spatial distribution of internal ice-layer structures with regard to Φ and water flow-path, while considering the likelihood of a frozen bed, suggests/reveals that in general large plumes are located along the main subglacial water paths crossing areas of elevated Φ (Fig. 2, and Supplementary Figs. 2, 3 and Tables 1, 2; see Methods section 5.1 and 5.5; note, uncertainties are largest in regions of sparse radar data¹⁵ which affects both Φ and the computed water path). According to theory, PiDMP depends dominantly on basal water-flux and topography, whereas the thermal state of the ice is of secondary importance. Thus, Our spatial analysis thus supports the hypothesis of basal freeze-on due to ascending water for Greenland.

We expanded both methods section 4.2 (new L504-542) and 5.5 (new L597-618) on the uncertainties and clarified the geometrical aspect of the freeze-on index Φ . They now read:

4.2 Basal freeze-on index. In order to spatially compare on an ice-sheet wide scale the potential freeze-on purely related to the PiDMP mechanism, we define the basal freeze-on index Φ as an entirely geometrical quantity, which reflects the basal topography relative to the surface topography. Since the mechanism of PiDMP requires water flux along an adverse bed slope, we only take the heat terms G_p and G_w in equation (15) that depend on the water flux Q_w and the bed-to-surface-slope ratio S to calculate Φ . The heat from the sum of the neglected heat sources $G_g + G_s$ is comparable to the difference of $G_p - G_w$ calculated for small water fluxes (i.e. $Q_w = 0.01$ is $\text{m}^3 \text{s}^{-1}$ per metre width) and ratios $S \geq -5$, even for increased ice flow (i.e. 100 m a^{-1}). For large water fluxes (i.e. $Q_w = 0.11 \text{ m}^3 \text{ s}^{-1}$ per metre width) however, the difference $G_p - G_w$ is one to two orders of magnitude larger (Supplementary Fig. 6a). By neglecting the two heat sources G_g and G_s , we implicitly assume that they are

being balanced by heat conduction towards the cooler ice-sheet surface as well as advection of colder surface-ice along flow.

To obtain the freeze-on index Φ as a non-dimensional quantity we rewrite equation (15) using the terms G_p and G_w only (ignoring G_g and G_s) and introducing equations (18) and (17), respectively, to obtain

$$\begin{aligned} \dot{f} &= \frac{(G_p - G_w) \cos \alpha_b}{L} \\ &= \frac{Q_w \rho_i g}{L} (\tilde{G}_p - \tilde{G}_w) \cos \alpha_b, \end{aligned} \quad (1)$$

and define Φ as

$$\begin{aligned} \Phi &= (\tilde{G}_p - \tilde{G}_w) \cos \alpha_b \\ &= \alpha_s [(1 - S)JC + 1 + (\rho_w/\rho_i - 1)S] \cos \alpha_b, \end{aligned} \quad (2)$$

where Φ now depends largely on bed and surface slope topography and scales with Q_w (Supplementary Figs. 6). The quantity G_w is the heat generated by sub-glacial water flow, while G_p is the heat needed to maintain the water at pressure-melt point. For a given water flux Q_w the freeze-on rate is the result of a simple multiplication between Q_w , the ice density ρ_i , the gravitational acceleration g , the inverse of the volumetric latent heat of freezing $1/L$, and Φ

$$\dot{f} = \frac{Q_w \rho_i g}{L} \Phi. \quad (3)$$

Comparing the non-dimensional heat terms \tilde{G}_w and \tilde{G}_p (Supplementary Fig. 6b; equations (17) and (18), respectively) for a range of $-2 \geq S \geq -11$ shows that the heat taken up by \tilde{G}_p to warm the water increases with decreasing S , while the heat emitted by \tilde{G}_w , due to water flow, decreases towards zero. As the increase in \tilde{G}_p is greater stronger than the decrease in \tilde{G}_w for decreasing S , the heat difference $\tilde{G}_p - \tilde{G}_w$ is increasing, leading to increased freeze-on in orders—as to balance the heat terms heats (Supplementary Fig. 6bb).

The calculation of the freeze-on index Φ depends highly on the bed and surface slope α_b and α_s . However, the interpolation of the RES data to obtain the bed topography leads to large vertical errors in bed elevation exceeding 600 m in regions of sparse radar data (see Supplementary Fig. 3 in Morlighem et al. (2017)¹⁵). A minimum error of 200 m allows us to nearly continuously visualise all used radar transects leading to a minimum transect width of ≈ 4 km in regions of sparse data. Increasing the minimum transect width to 5 and 10 km leads to a maximum error of 225 and 300 m, respectively. Note that Φ and the water path are consistent as the same dataset for bed and surface is used, so that with each improved topography dataset the Φ and water paths become more reliable.

5.5 Estimating correlation between freeze-on index and plumes. To quantify the relation between plumes and areas with freeze-on index-areas, we follow streamlines along the reversed (surface) ice-flow gradient starting at each plume, and determine/evaluate if they cross areas of freeze-on index Φ . Further, we compare the results between the mapped plume sets ($\geq H/3$ and $< H/3$) with the averaged results for 10,000 randomly chosen locations. For the reversed streamlines calculated on a 1050 m grid, as used in this paper, we find the best match between plumes and Φ areas for the large plumes, followed by the small plumes (result reduced by $\approx 10\%$) and the random set, which are reduced by up to 8% and a further 9%—a result by ≈ 30 —20% lower, respectively (see Supplementary Tables 1 and 2). By comparing this difference between the observed and random data set relative to the total random mismatch (the percentage of plumes that are predicted to have no freeze-on area), we find that the observed data results in a distinctively better match than the random data set (Table 2; e.g. the mismatch is reduced by $> 35\%$ and $> 17\%$ for plumes with heights $\geq H/3$ and $< H/3$, respectively). Consequently, we argue that the plume distribution is related to the pattern in Φ , since if in the observational data set the plumes are randomly distributed we would expect a similar mismatch between the data sets.

Calculating reverse streamlines on a 150 m grid¹⁶ along a 5 km stretch, for seeds within 1.5 km radius from the plume, about 98.97% and 94.93% of the large and small plumes, respectively, and less than 85.80% of the random locations are reached by ice flowing over areas of Φ . This result depends on the grid, the seed area and the streamline distance. Reverse streamlines calculated on the 1050 m grid, seeded over an area of 3 km along 10 km, result in comparable values as for the fine grid (1.5 km and 5 km). Larger RES mapping errors for small plumes (see Methods section 1) partly explains the better result for plumes $> H/3$.

Lines 147-149: “In addition, basal sliding becomes more important downstream, affecting the distribution of horizontal velocity with depth, and thus further decreases the plume height (Fig. 6).”

Lines 213-216: “In fast flow areas, the large ice flux together with increased basal sliding (leading to a vertical distribution of the horizontal velocity resembling the plug flow case) prevent in general high rising plumes (Fig. 5). An exception occurs for very large water fluxes and along-flow series of freeze-on areas, which leads to stacking of the advected plumes (Fig. 5b).”

Observations from Petermann basin (e.g. Bell et al., 2014 and Panton and Karlsson, 2015) indicate that the plumes are increasing in size downstream.

Figure 5 in the main text shows how for an equal amount of freeze-on flux F the plumes reach different

heights depending on the total ice flux Q . As the total ice flux Q increases along flow, the plume height decreases for the same value of F . However, any of the three single plume heights increases along their accretion zone. This means that the local increase in plume height along an accretion area happens for any ice flux, but is just less steep for greater ice flux. In figure 5b we also show that if the accreted ice from further upstream is advected over an accretion area downstream then the freshly generated plume pushes the existing accreted ice up. This leads to the stacking of accreted ice and larger plumes are obtained than the case where only the freeze-on flux F over the local accretion area with regard to the total ice flux is considered.

Further, as discussed above, a high ice flux can be compensated by larger Φ and larger water flux, with both increasing towards the ice sheet margin (e.g. Fig. 2 in main text and statement on water flux by Dow et al. (2018)⁷).

Over the Petermann glacier Panton and Karlsson et al. (2015; Figure 6c)¹⁸ show the height and orientation (width and colour of circle) of the 'unit of disrupted radiostratigraphy' (UDR; note, not all UDR's are plumes as described in our paper, which we mapped by their appearance of rising from the bed). In this figure, a general along-flow reduction in the UDR's height is seen, with the biggest circles located upstream and becoming smaller downstream along ice flow. This fits with the Figure 5 presented in our paper, where over accretion areas, despite increasing ice flux, the plume height increases but decreases once the accretion area ends.

The relative height might behave similarly though in absolute height the plumes in the fast flow regions are smaller than in the interior - also visible in Figure 5a,b compared to 5d; also compare Suppl. Fig. 1 with Response Figure 2.

In order to visualise the example mentioned by the Reviewer of Bell et al. (2014)² we plotted the radar transects used in their paper (Response Figure 2a-d), together with a map showing the transects in combination with the likely state of the thermal condition at the bed¹³ and the flow direction⁹. We can see from this figure that ice flow is oblique to the transects and is converging towards the left side of the 'across' profiles. Over the observed area flow velocities are still moderate ($30 - 50 \text{ m a}^{-1}$) but increasing downstream. 'Along' flow transects (Response Figure 2e,f) suggest that plumes extend over a considerable area downstream, where plumes from upstream seem to be pushed up by freeze-on areas downstream (e.g. transect c and f). The general trend is that plume-height decreases downstream. This is a setting similar to our Figure 5b where plumes are stacked on top of each other with the difference that over Petermann we have additional convergence flow.

With ice flux in a converging zone, the ice flux becomes larger with both meteoric ice flux M and ice

flux F originating from freeze-on contributing towards the total ice flux. For the case of uniform freeze-on over the convergence area the plume height is expected to be the same within the convergence as the ice-flux ratio R hasn't changed. However, for a reduction/increase in ice-flux ratio R the converged plumes are expected to fall/rise, respectively.

In summary, our modelling and suggested mechanism of freeze-on is fully consistent with the Petermann

Response Figure 2: **Overview of mapped plume locations over Petermann bei Bell et al. (2014).** Radar transects of Operation Ice Bridge Flight 20110429¹⁰ shown as profiles and transects in combination with the likely state of the thermal condition at the bed¹³ and the flow direction⁹. The section of the transect shown as radar profiles are highlighted in black. **a-d**, 'Across' flow profiles of radar transects 027, 022, 019 and 014. Highlighted in red is the section shown in Bell et al. (2014; Figure 4)². **e,f**, 'Along' flow profiles perpendicular to the sections shown in **a-d**. The sections within the profiles highlighted with the red horizontal bar are highlighted in green on the map. Note that all profiles are oblique to flow and not entirely along or across flow and that the surface is kept at zero and not representing the surface topography..

example put forward by the Reviewer #3.

In Fig. 2, there is a large area with elevated freeze-on index but the plumes are downstream of this area. This seems to be in direct contrast with the modelling results (e.g. Fig. 5b where the plumes decrease in size downstream). Are the plumes observed in Petermann a result of “very large water fluxes”? And in this case, how large? Can the model reproduce the observations from Petermann?

Looking at the area with elevated freeze-on index we realised that we 'lost' some of the small plumes originally displayed in the Supplementary Fig. 1 of the originally submitted paper. This has now been amended so that all plumes from the original dataset (Fig. 1 and Supplementary Fig. 1) and the dataset added during revision 1 are used in the plots.

The relatively wide strip with elevated Φ upstream of Petermann glacier (Figure 2 in main text) is again in an area of a likely frozen bed. The distribution of the water flow paths suggests that only a few main paths (2-3) are crossing this area, coinciding with the occurrence of plumes. This fits nicely with our suggested mechanism of basal freeze-on.

With the previously missing 'small' plumes added it is evident that the plumes start where the freeze-on index is strong, with plumes first being small and increasing further downstream. As the transects are across flow the longitudinal extent of the plume is not visible, but one can see that downstream the observed plumes on the across transects become bigger. This is consistent with the modelling results in Figure 5 (as well as Figure 3) as discussed earlier where plumes increase in height over accretion areas. The water fluxes needed to obtain these plumes are in the range of the fluxes calculated for the example in the paper. A back-of-the-envelope calculation for a plume located at -85 km easting and -1250 km northing that reaches up to 0.25 (600 m) of the ice thickness with an accretion area extending between 8 and 12 km would require freeze-on rates between 0.27 to 0.41 m a^{-1} leading to the freeze-on flux required to obtain the observed maximum plume height ($h = 0.25$ and $R = 0.1218$ with $Q = 27000$ obtained from the product of the balance velocity and ice thickness $11 \text{ m a}^{-1} \times 2400 \text{ m}$). Using a water flux of $Q_w = 0.1$ and $0.15 \text{ m}^3 \text{ s}^{-1}$ per metre width to calculate the freeze-on rate from Φ , leads to the required freeze-on rates over a few grid cells with the lower water flux and averaged over the whole accretion area for the higher water flux.

Lines 199-202: “Away from the divide, the first large plumes ($> 1/3H$) start to form in regions where the basal water flux Q_w is increasing, with the total ice flux Q and freeze-on index Φ still being generally small. Further along, towards the margin both Φ and Q_w tend to increase (while Q is still moderate) and

allow for larger plume heights.”

Based on the suppl. Fig. 2, I would argue that it is just as likely that the plumes form in areas where there is a transition from frozen to thawed conditions.

It is possible that transitional areas between frozen and thawed conditions are a good environment for plume growth (especially $< H/3$, as in such locations a limited amount of water would be expected). However, for plume heights $> H/3$ this argument is not as strong as our observations made in context with Φ , water flux and ice flux, as in this case we would expect large plumes everywhere along this transition. By plotting the data with the main water paths it is generally visible that large plumes are located at the start or along these water systems (new Fig. 2).

We changed the sentence by adding that the regions likely correspond to basal melt water converging, so that it now reads (new L228-230):

Away from the divide, the first large plumes ($> H/3$ $1/3H$) start to form in regions where the basal melt water is likely convergent, leading to an increasing the basal water flux Q_w is increasing, with the total ice flux Q and freeze-on index Φ still being generally small.

#2 Amount of basal meltwater and freeze-on rates

Line 154: Reference no. 27 and 28.

Both references here are modelling studies and do not provide direct evidence of subglacial water or major pathways.

We agree entirely with the Reviewer. The text does highlight this, as we wrote the following in the sentence ending on line 154 (now new L186-189): ‘However, recent modelling studies suggest the existence of major meltwater pathways, subglacial lakes and, related to the mechanism described as the pressure-induced depression of the melting-point (PiDMP), widely occurring basal freeze-on^{7,12}. ‘

Line 167-168: “It is readily conceivable that locally conditions arise, which lead to freeze-on rates comparable to the ones used in our experiments.”

Is it? This statement comes across as unsubstantiated. Is the slope ratio reasonable for the areas where the plumes are observed? Are the values for the water flux reasonable? This needs to be stated explicitly. Also, there is a difference between conditions such as these arising locally, and to them being present on a large scale across the entire ice sheet as evidenced by the extensive presence of the plumes.

Although Φ is observed on a large scale over the ice sheet it is a local phenomena that depends (as

explained earlier) on the bed and surface slopes locally (grid size). The smaller the grid used to calculate the slopes, the larger the range in local slopes (flatter as well as steeper).

We changed the sentence to refer to the grid size (new L201-203):

As the calculated bed and surface slope depend heavily on the used grid size, it is readily conceivable that locally conditions arise in Φ and Q_w that lead to freeze-on rates comparable to the ones used in our experiments (Fig. 7).

Lines 251-252: “and broadly coincide with modelled freeze-on regions [28]”

This statement is a bit misleading since the freeze-on rates in the study cited (Dow et al.) are at least an order of magnitude smaller than what is invoked here.

In our text we state that the regions broadly coincide and not the freeze-on rates. Dow et al. (2018)⁷ state in their discussion that they find subglacial water accumulating in troughs feeding NEGIS and Petermann Glacier (as well as Jakobshavn Isbrae), which allows for more supercooling freeze-on to occur (however with low rates).

The freeze-on rates are expected to differ with our study as the grid chosen by Dow et al. (2018)⁷ is in average 5,700 m compared to our fixed 1,050 m grid. Dow et al.⁷ state in their paper that their results were not changed by the grid resolution - tested by using a minimum grid length of 450 m over NEGIS and compensating with a very rough meshing elsewhere in the domain. As an experiment we looked at our whole modelling domain and at two specific locations covered by Φ (upstream Petermann (centered at -80 m easting and $-1,240$ km northing) and the region with the plume highlighted in our paper (centered at 225 m easting, $-1,360$ km northing; Supplementary Fig. 1)). We used a constant water flux of $Q_w = 0.11$ to multiply with Φ in order to get the freeze-on rates \dot{f} over these two regions for four different grid sizes (150, 300, 1,050 and 4,950 m). The maximum value in freeze-on rate for the entire domain (freeze-on rate calculated for the given Q_w for the maximum Φ found in the domain for each grid size) is 6 times larger for the smallest grid than for the largest grid (see Table 2. However the location is likely not the same. Comparing at two locations we see from the Table 2 that the values vary from site to site for different grid sizes. From this example one can see that the freeze-on rates depend locally very much on the grid size. On a coarser grid, the slopes get averaged out, meaning the big picture is the same but locally the values differ, so that the freeze-on rates calculated for a 5 or 1 km grid are substantially different. Also, the water flux is expected to change locally to with a different grid size - as water paths would change - and affect the freeze-on rates (see Figure 7 in main text).

Moreover, the study by Dow et al. (2018)⁷ refers in general to averaged freeze-on rates, mentioning

Table 2: **Grid size dependence of freeze-on rates.** Calculating the freeze-on rate \dot{f} by multiplying Φ with the water flux Q_w , the ice density ρ_i , gravitational acceleration g , and dividing it by the volumetric latent heat of freezing L for the whole modelling domain using four different grid sizes. The maximum value found for the whole domain as well as the maximum value found in two specific regions with a patch of Φ are shown (upstream Petermann (centered at -80 km easting and -1240 km northing) and the region with the highlighted plume from the paper (Supplementary Fig. 1).

$Q_w = 0.11$	150 m grid	300 m grid	1,050 m grid	4,950 m grid
max. \dot{f} for whole domain	26.5 m a ⁻¹	21.9 m a ⁻¹	10.17 m a ⁻¹	4.3 m a ⁻¹
max. \dot{f} for -80 m E, -1240 m N	1.5 m a ⁻¹	1.2 m a ⁻¹	1.1 m a ⁻¹	0.6 m a ⁻¹
max. \dot{f} for 225 m E, -1360 m N	1.6 m a ⁻¹	1.3 m a ⁻¹	0.95 m a ⁻¹	0.3 m a ⁻¹

some observed peak values (e.g. for the central ice divide region they state an average freeze-on rate of 2.9×10^{-4} m a⁻¹ and a maximum value of 1.5 mm a⁻¹, whereas along a 5 km section of a channel a maximum value of 5.87 m a⁻¹ was found). As freeze-on rates are driven by local conditions, averaged freeze-on rates are not very informative.

We added a sentence to the discussion (new L298-300);

Discrepancies in freeze-on rates between models and reality can be explained by differences in topographic resolution, affecting both the water pathways and the range in bed and surface slopes, with a coarser grid levelling out the extremes in slope.

Lines 262-263: “Therefore, statements about the mismatch between plume size and freeze-on rates [28] are not valid without considering as well the effect on the relationship between plume height and freeze-on rate both from the flux relationship and rheology contrast.”

I agree that looking at the freeze-on rate in itself is not adequate when examining plume heights. However, from the figures it seems that 10*softer ice roughly leads to a doubling in plume height. Since the freeze-on rates are one to two orders of magnitude larger than the rates from Dow et al., using the freeze-on rates from the latter study would never lead to a plume of that height, regardless of whether or not ice-flux and rheology contrasts were included. It should be explicitly stated somewhere that the freeze-on rates from Dow et al., would only lead to X m (or x relative height) plumes. Otherwise the reader might think that the addition of ice-flux and rheology contrast in itself is enough.

The Lines 262-263 outlined above were aimed at the statement made by Dow et al. (2018)⁷ in their discussion where they calculate for the central ice divide region the height of the unit of disturbed ra-

diostratigraphy (UDR) of 15 m with a freeze-on rate of 1.5 mm a^{-1} during 10 ka instead of the observed 115 m.

We added a section in the Methods (new section 3.1) and referred to it in the main text (L141-149 and L196-197)) so that the statement on Lines 262-263 make more sense to the reader.

The paragraph new L175-183 is slightly changed to:

The flux-dependence illustrated has crucial implications for areas of low surface-accumulation rates (e.g. East Antarctica or Northeast Greenland), since where meteoric ice flux M is small, even a low freeze-on flux F is able to produce relatively large plumes (see Methods section 3.1). This effect is amplified in regions with small surface accumulation areas, which further reduce the meteoric ice flux M ; such regions, which exist for instance near ice divides or where diverging flow is divergent regions, typically upstream of above rising beds^{8,17}. Downstream, plume heights generally decrease owing due to the increasing meteoric ice flux M , which is a consequence of both contributing accumulation area and flow-convergence. In addition, basal sliding becomes more important downstream, affecting the distribution of horizontal velocity with depth, and thus further decreasing the plume height (Fig. 6).

And Lines 196-197 are changed to new L232-233:

As the ice flux is low, only low water fluxes are needed to produce the required freeze-on rates (a few millimeters to centimeters per year; see Methods section 3.1).

The new Method section reads (new L439-446):

3.1 Calculating freeze-on plume parameters. For a plume near the ice divide of 115 m height ($H = 3000 \text{ m}$, $h = 0.038$, $\bar{u} = 2 \text{ m a}^{-1}$ with $Q = 6000 \text{ m}^3 \text{ a}^{-1} \text{ m}^{-1}$ using equation (6)), we obtain an ice-flux ratio $R = 0.0035$ (solving $\omega(h)$ in equation (11b); see Fig. 6), and a freeze-on flux $F = 21 \text{ m}^3 \text{ a}^{-1} \text{ m}^{-1}$ which is obtained by the multiplication $R \times Q$ (equation (12)). Dividing F by either the accretion extent or the freeze-on rate \dot{f} , results in \dot{f} or the accretion extent, respectively. Not considering the effect of ice flux Dow et al.⁷ made a simple plume height calculation by multiplying the freeze-on rate of $\dot{f} = 1.5 \text{ mm a}^{-1}$ with time, resulting in a height of only 15 m over 10 ka. Whereas including ice-flux, a plume height of 115 m with a freeze-on rate $\dot{f} = 1.5 \text{ mm a}^{-1}$ requires an accretion extent of 14 km.

#3 Frozen basal conditions

Lines 194-197: “Near the ice divide the observed plumes are usually smaller than $1/10H$, in the range of a few 100 meters. For these relative small plume heights the flux ratio R too is small when assuming flow purely by internal deformation. As the ice flux is low, only low water fluxes are needed to produce the required freeze-on rates (a few millimeters to centimeters).”

Near the ice divide, radar observations indicate that the bed is likely frozen (as shown in Suppl. Fig. 2), thus the subglacial water flux must be zero and therefore also the freeze-on rates. It is a significant weakness in this manuscript that the presence of the plumes in the interior of the ice sheet cannot be explained by the freeze-on process but this weakness is not mentioned or discussed.

Many of the changes made so far to the revised manuscript together with answers given in this letter contribute to the above comment. Nevertheless, we reply to this comment with some more detail:

As mentioned above we improved Figure 2 by adding the likely thermal state for a frozen bed (together with the outline for low ice flux; see answer to comment #1 by Reviewer #3) and moved the original figure 2 to the Supplement (now Supplementary Fig. 2).

However, below follow some more arguments why we cannot exclude the presence of water and therefore plumes near the ice divide.

The map showing the likely thermal state in Fig. 2 and the supplementary Fig. 3 (previously Supplementary Fig. 2) is not purely a product from radar observations - it is derived from several ice-flow modelling results and radar data inversion. The thermal transitions between frozen and thawed can be quite complex on a smaller scale as shown by Chu et al. 2018⁵ using radar bed reflectivity on the example of Petermann glacier, with the general large scale pattern being in good agreement with the likely thermal state map.

The thermal state of the bed (if frozen or not) is only known for certain where boreholes exist. MacGregor et al. (2016)¹³ highlight that their synthesis of thermomechanical models and remote inferences agree in general for the Northeast Greenland Ice stream (thawed) and the west facing slopes of the central ice divides (frozen), but that e.g. the borehole-observed basal thermal state near NGRIP (thawed bed) is rarely represented by either the models or remote inferences and leaving approx. one third of the ice sheet in need of more observations. Therefore, we cannot exclude that some water paths in regions of a likely frozen bed exist, but rather assume that the likelihood of water is smaller. As an example Rogozhina et al. (2016)¹⁹ predict a high geothermal heat flux in the region south of NEEM and to the southeast, so that in these regions where a likely frozen bed is predicted¹³ a thawed bed instead might be likely (Fig. 3a in Rogozhina et al. (2016)¹⁹). This would also fit with our observations of existing plumes. We could argue that if these observed near basal structures are due to freeze-on we can use the plumes to inform on possible water paths beneath the ice sheet and in consequences make inferences about the geothermal flux in particular.

Looking at the two red circles along the divide, highlighted by the Reviewer #3 in the review attachment, we find that they are in regions with very low ice flux and very flat surfaces. The likelihood of thermal

state predicts this region to be a thermal boundary, but from the borehole at NGRIP and from radio echo sounding^{4,13} it is known that in this region there is melting at the ice-sheet bed. Furthermore, the divide region has a very flat surface which easily allows for stronger negative bed-to-surface-slope ratio (S), with freeze-on rates approaching the centimetre per year range even for small water fluxes (see Figure 7a in the main text). Owing to the very low ice flux, these freeze-on rates are enough to lead to observed plume heights between 100 to 300 m.

A back-of-the-envelope calculation for the plume in the center of the southernmost red circle results in a freeze-on flux $F = RQ$ of $33 \text{ m}^3 \text{a}^{-1} \text{m}^{-1}$ for the plume height of 150 m and an ice thickness of 3200 m and accounting for a balance velocity of 2 m a^{-1} (total ice flux $Q = 3200 * 2 = 6400 \text{ m}^3 \text{a}^{-1} \text{m}^{-1}$; the relative height $h = 150/3200 = 0.0469$ resulting in an ice-flux ratio: $R = 0.0052$), which would require freeze-on along $\approx 22 \text{ km}$ for a freeze-on rate of 1.5 mm a^{-1} as used by Dow et al. (2018)⁷ or $\approx 3 \text{ km}$ when assuming $\dot{f} = 1 \text{ cm a}^{-1}$. Not considering the effect of ice flux, Dow et al.⁷ made a simple calculation by multiplying the freeze-on rate with time to obtain the plume height, resulting in a height of only 15 m over 10 ka.

For the plume furthest to the south in the same red circle with a plume height of 230 m, ice thickness of 3200 m and balance velocity of 2.5 m a^{-1} our calculation leads to a total ice flux $Q = 8000 \text{ m}^3 \text{a}^{-1} \text{m}^{-1}$, $h = 0.0719$ and $R = 0.012$, so that $F = 96 \text{ m}^3 \text{a}^{-1} \text{m}^{-1}$ and requiring an accretion length of $\approx .10 \text{ km}$ for $\dot{f} = 1 \text{ cm a}^{-1}$.

The red circled area further to the east is still in an area with low ice flux (balance velocity $\approx 10 \text{ m a}^{-1}$, however with a steeper surface slope (see Figure 1 in main text) and likely more water flowing at the base, both allowing for larger freeze-on rates (larger Φ) compensating for the increased ice flux.

We added a paragraph to the introduction to highlight how poorly constrained the basal temperature of the Greenland ice sheet is (new L57-63).

The basal temperature of the Greenland ice sheet is only poorly constrained, as we do not know the geothermal heat flux or how it varies on a kilometre to 100 km scale. Analysis combining independent data are needed to obtain reconstructions for the basal temperature^{13,19}. Neither do we know the basal topography well enough to evaluate if there are warm-based deeper parts surrounded by frozen areas^{5,15}. In addition does the community's limited understanding of basal hydrology prevent us from predicting its thermodynamics consequences, in particular answering whether it is possible for melt-water channels to advance through freeze-on areas so as to connect isolated melt patches.

References

- [1] Alley, R. B., Lawson, D. E., Evenson, E. B., Strasser, J. C., and Larson, G. J. (1998). Glaciohydraulic supercooling: a freeze-on mechanism to create stratified, debris-rich basal ice: II. Theory. *Journal of Glaciology*, 44(148):563–569.
- [2] Bell, R. E., Tinto, K., Das, I., Wolovick, M., Chu, W., Creyts, T. T., Frearson, N., Abdi, A., and Paden, J. D. (2014). Deformation, warming and softening of Greenland’s ice by refreezing meltwater. *Nature Geoscience*, 7:497–502. doi:10.1038/ngeo2179.
- [3] Bons, P. D., Jansen, D., Mundel, F., Bauer, C. C., Binder, T., Eisen, O., Jessell, M. W., Llorens, M.-G., Steinbach, F., Steinhage, D., and Weikusat, I. (2016). Converging flow and anisotropy cause large-scale folding in Greenland’s ice sheet. *Nature Communications*, 7. doi:10.1038/ncomms11427.
- [4] Buchardt, S. L. and Dahl-Jensen, D. (2007). Estimating the basal melt rate at NorthGRIP using a Monte Carlo technique. *Annals of Glaciology*, 45:137–142.
- [5] Chu, W., Schroeder, D., Seroussi, H., Creyts, T., and Bell, R. (2018). Complex basal thermal transition near the onset of Petermann Glacier, Greenland. *JGR*, 123(F004561):212–241.
- [6] Dahl-Jensen, D., Gundestrup, N., Gogineni, S. P., and Miller, H. (2003). Basal melt at NorthGRIP modeled from borehole, ice-core and radio-echo sounder observations. *Annals of Glaciology*, 37:207–212. doi.org/10.3189/172756403781815492.
- [7] Dow, C. F., Karlsson, N. B., and Werder, M. A. (2018). Limited impact of subglacial supercooling freeze-on for Greenland Ice Sheet stratigraphy. *Geophysical Research Letters*, 45. doi:10.1002/2017GL076251.
- [8] Gudmundsson, G. H. (2003). Transmission of basal variability to a glacier surface. *Journal of Geophysical Research*, 108(B5).
- [9] Joughin, I., Smith, B., Howat, I., Scambos, T., and Moon, T. (2010). Greenland flow variability from ice-sheet-wide velocity mapping. *Journal of Glaciology*, 56:415–430. doi:10.3189/002214310792447734.
- [10] Leuschen, C., Gogineni, P., Hale, R., Paden, J., Rodriguez, F., Panzer, B., and Gomez, D. (2014, updated 2016). IceBridge MCoRDS L1B geolocated radar echo strength profiles, version 2. Boulder, Colorado USA: National Snow and Ice Data Center. <http://dx.doi.org/10.5067/90S1XZRBAX5N>.

- [11] Leysinger Vieli, G. J.-M. C., Hindmarsh, R. C. A., and Siegert, M. J. (2007). Three-dimensional flow influences on radar layer stratigraphy. *Annals of Glaciology*, 46:22–28.
- [12] Livingstone, S., Clark, C., Woodward, J., and Kingslake, J. (2013). Potential subglacial lake locations and meltwater drainage pathways beneath the Antarctic and Greenland ice sheets. *The Cryosphere*, 7:1721–1740. doi:10.5194/tc-7-1721-2013.
- [13] MacGregor, J. A., Fahnestock, M. A., Catania, G. A., Aschwanden, A., Clow, G. D., Colgan, W. T., Gogineni, S. P., Morlighem, M., Nowicki, S. M., Paden, J. D., Price, S. F., and Seroussi, H. (2016). A synthesis of the basal thermal state of the Greenland Ice Sheet. *JGR*, 121(F003803):1328–1350.
- [14] Morlighem, M., Rignot, E., Mouginot, J., Seroussi, H., and Larour, E. (2014). Deeply incised submarine glacial valleys beneath the Greenland Ice Sheet. *Nature Geoscience*, 7:418–422.
- [15] Morlighem, M., Williams, C., Rignot, E., An, L., Arndt, J. E., Bamber, J., Catania, G., Chauch, N., Dowdeswell, J. A., Dorschel, B., Fenty, I., Hogan, K., Howat, I., Hubbard, A., Jakobsson, M., Jordan, T. M., Kjeldsen, K. K., Millan, R., Mayer, L., Mouginot, J., Nol, B., O’Cofaigh, C., Palmer, S. J., Rysgaard, S., Seroussi, H., Siegert, M. J., Slabon, P., Straneo, F., van den Broeke, M. R., Weinrebe, W., Wood, M., and Zinglensen, K. (2017a). BedMachine v3: Complete bed topography and ocean bathymetry mapping of Greenland from multibeam echo sounding combined with mass conservation. *Geophysical Research Letters*, 44. doi.org/10.1002/2017GL074954.
- [16] Morlighem, M., Williams, C., Rignot, E., An, L., Arndt, J. E., Bamber, J., Catania, G., Chauch, N., Dowdeswell, J. A., Dorschel, B., Fenty, I., Hogan, K., Howat, I., Hubbard, A., Jakobsson, M., Jordan, T. M., Kjeldsen, K. K., Millan, R., Mayer, L., Mouginot, J., Nol, B., O’Cofaigh, C., Palmer, S. J., Rysgaard, S., Seroussi, H., Siegert, M. J., Slabon, P., Straneo, F., van den Broeke, M. R., Weinrebe, W., Wood, M., and Zinglensen, K. (2017b). IceBridge BedMachine Greenland, version 3. Boulder, Colorado USA: NASA DAAC at the National Snow and Ice Data Center.
- [17] Ng, F. S. L. (2015). Spatial complexity of ice flow across the antarctic ice sheet. *Nature Geoscience*, 8:847–850. doi:10.1038/ngeo2532.
- [18] Panton, C. and Karlsson, N. (2015). Automated mapping of near bed radio-echo layer disruptions in the greenland ice sheet. *Earth and Planetary Science Letters*, 432:323–331. doi:10.1016/j.epsl.2015.10.024.
- [19] Rogozhina, I., Petrunin, A. G., Vaughan, A. P. M., Steinberger, B., Johnson, J. V., Kaban, M. K.,

Calov, R., Rickers, F., Thomas, M., and Koulakov, I. (2016). Melting at the base of the Greenland ice sheet explained by Iceland hotspot history. *Nature Geoscience*, 9:366–369. doi:10.1038/ngeo2689.

[20] Wolovick, M. J. and Creyts, T. T. (2016). Overturned folds in ice sheets: Insights from a kinematic model of traveling sticky patches and comparisons with observations. *Journal of Geophysical Research*, 121:1065–1083. doi:10.1002/2015JF003698.

[21] Wolovick, M. J., Creyts, T. T., Buck, W. R., and Bell, R. E. (2014). Traveling slippery patches produce thickness-scale folds in ice sheets. *Geophysical Research Letters*, 41. doi:10.1002/2014GL062248.

REVIEWERS' COMMENTS:

Reviewer #1 (Remarks to the Author):

The latest revisions contribute further to over the clarity of the results and manuscript. I support its publication.

In my opinion, the additional comments raised by Reviewer 3's have been adequately addressed. In particular:

- The author can clearly justify how their work differs and improves upon other related studies.
- The amendments brought to the new Figure 2, including the additional plumes and removal of artificial threshold allows the authors to further support the existence of a spatial correlation between plumes, freeze-on index and water paths. It is true that uncertainties exist, and the authors now acknowledge this more explicitly in the text.
- Response Figure 2, supported by comments, reconciles the author's work with observations from Petermann basin
- There are further details on the distribution and amount of freezing, involving mainly their sensitivity to grid resolution.
- The last point concerns the possibility of freeze-on / basal melting in the interior. The main argument brought by the author relates to the overall poor knowledge of the basal conditions in Greenland. They added a new paragraph now discussing this point openly, and as such, I believe have also adequately addressed the last concern.